# Explaining Explainability: Recommendations for Effective Use of Concept Activation Vectors

**Angus Nicolson**                                                *angus.nicolson@eng.ox.ac.uk*
*Institute of Biomedical Engineering*
*University of Oxford*

**Lisa Schut**                                                          *schut@robots.ox.ac.uk*
*OATML, Department of Computer Science*
*University of Oxford*

**Alison J. Noble**                                               *alison.noble@eng.ox.ac.uk*
*Institute of Biomedical Engineering*
*University of Oxford*

**Yarin Gal**                                                          *yarin.gal@cs.ox.ac.uk*
*OATML, Department of Computer Science*
*University of Oxford*

**Reviewed on OpenReview:** *https://openreview.net/forum?id=7CUluLpLxV*

## Abstract

Concept-based explanations translate the internal representations of deep learning models into a language that humans are familiar with: concepts. One popular method for finding concepts is Concept Activation Vectors (CAVs), which are learnt using a probe dataset of concept exemplars. In this work, we investigate three properties of CAVs: (1) inconsistency across layers, (2) entanglement with other concepts, and (3) spatial dependency. Each property provides both challenges and opportunities in interpreting models. We introduce tools designed to detect the presence of these properties, provide insight into how each property can lead to misleading explanations, and provide recommendations to mitigate their impact. To demonstrate practical applications, we apply our recommendations to a melanoma classification task, showing how entanglement can lead to uninterpretable results and that the choice of negative probe set can have a substantial impact on the meaning of a CAV. Further, we show that understanding these properties can be used to our advantage. For example, we introduce spatially dependent CAVs to test if a model is translation invariant with respect to a specific concept and class. Our experiments are performed on natural images (ImageNet), skin lesions (ISIC 2019), and a new synthetic dataset, Elements. Elements is designed to capture a known ground truth relationship between concepts and classes. We release this dataset to facilitate further research in understanding and evaluating interpretability methods.

## 1 Introduction

Deep learning models have become ubiquitous, achieving performance reaching or surpassing human experts across a variety of tasks. However, currently, the inherent complexity of these models obfuscates our ability to explain their decision-making process. As they are applied in a growing number of real-world domains, there is an increasing need to understand how they work (Doshi-Velez & Kim, 2017; Reyes et al., 2020). This transparency allows for easier debugging and better understanding of model limitations.

Model explanations can take many forms, such as input features, prototypes or concepts. Recent work has shown that explainability methods that focus on low-level features can incur problems. For example, saliency methods, which determine the sensitivity of a deep learning model to individual pixels, can suffer from confirmation bias and lack model faithfulness (Adebayo et al., 2018). Even when faithful, saliency maps only show 'where' the model focused in the image, and not 'what' it focused on (Achtibat et al., 2022; Colin et al., 2022).

To address these problems, concept-based methods provide explanations using high-level terms that humans are familiar with. A popular method is concept activation vectors (CAVs): a linear representation of a concept found in the activation space of a specific layer using a probe dataset of concept examples (Kim et al., 2018). However, concept-based methods also face challenges, such as their sensitivity to the specific probe dataset (Ramaswamy et al., 2022a; Soni et al., 2020).

In this paper, we focus on understanding three properties of concept vectors:

1. They cannot be **consistent** across layers,

2. They can be **entangled** with other concepts,

3. They can be **spatially dependent**.

We provide tools to analyse each property and show that they can affect testing with CAVs (TCAV) (§6.1, §6.2 and §6.3) and lead to misleading explanations. To minimise the impact these effects can have, we recommend: creating CAVs for multiple layers, verifying expected dependencies between related concepts, and visualising spatial dependence (§7). These properties do not imply that CAVs should not be used. On the contrary, we may be able to use these properties to better understand model behaviour. For example, we introduce a modified version of CAVs that are spatially dependent and can be used to identify translation invariance in convolutional neural networks (CNNs). Additionally, we do not claim that these properties are necessarily unexpected. Instead, the aim of this paper is to formally define each one, confirm when it can occur, and then clearly examine the consequences of these properties on the interpretation of TCAV scores. In each case, even when expected, these properties can cause misleading explanations.

To provide a concrete example of how the properties can affect experiments, we examine the use-case of Yan et al. (2023) which uses CAVs in the context of skin cancer diagnosis (§ 7). We demonstrate how to use our recommendations to sanity-check TCAV results and, in this specific case, we show that entangled concepts lead to uninterpretable explanations. Our results also indicate that the choice of negative probe dataset can have a substantial impact to the meaning of a CAV. This use-case, combined with our recommendations, can be used by practitioners as a guide for how to effectively use CAVs in practice.

To help explore these properties, we created a configurable synthetic dataset: Elements (§4). This dataset provides control over the ground-truth relationships between concepts and classes in order to understand model behaviour. Using the Elements dataset, researchers can study (1) the faithfulness of a concept-based explanation method and (2) the concept entanglement in a network.

## 2 Background: Concept Activation Vectors

A CAV (Kim et al., 2018) is a vector representation of a concept found in the activation space of a layer of a neural network (NN). Consider a NN which can be decomposed into two functions: $g_l(\boldsymbol{x}) = \boldsymbol{a}_l \in \mathbb{R}^m$ which maps the input $\boldsymbol{x} \in \mathbb{R}^n$ to a vector $\boldsymbol{a}_l$ in the activation space of layer $l$, and $h_l(\boldsymbol{a}_l)$ which maps $\boldsymbol{a}_l$ to the output. To create a CAV for a concept $c$ we need a probe dataset $\mathbb{D}_c$ consisting of positive samples $\mathbb{X}_c^+$ (concept examples), and negative samples $\mathbb{X}_c^-$ (randomly sampled in-distribution images). For the sets $\mathbb{X}_c^-$ and $\mathbb{X}_c^+$, we create a corresponding set of activations in layer $l$:

$$\mathbb{A}_{c,l}^+ = \{g_l(\boldsymbol{x}_i) \quad \forall \boldsymbol{x}_i \in \mathbb{X}_c^+\}, \text{ and } \mathbb{A}_{c,l}^- = \{g_l(\boldsymbol{x}_i) \quad \forall \boldsymbol{x}_i \in \mathbb{X}_c^-\}, \tag{1}$$

We find the CAV $\boldsymbol{v}_{c,l}$ by training a binary linear classifier to distinguish between the sets $\mathbb{A}_{c,l}^+$ and $\mathbb{A}_{c,l}^-$:

$$\boldsymbol{a}_l \cdot \boldsymbol{v}_{c,l} + b_{c,l} > 0 \quad \forall \boldsymbol{a}_l \in \mathbb{A}_{c,l}^+, \text{ and } \boldsymbol{a}_l \cdot \boldsymbol{v}_{c,l} + b_{c,l} \leq 0 \quad \forall \boldsymbol{a}_l \in \mathbb{A}_{c,l}^-, \tag{2}$$

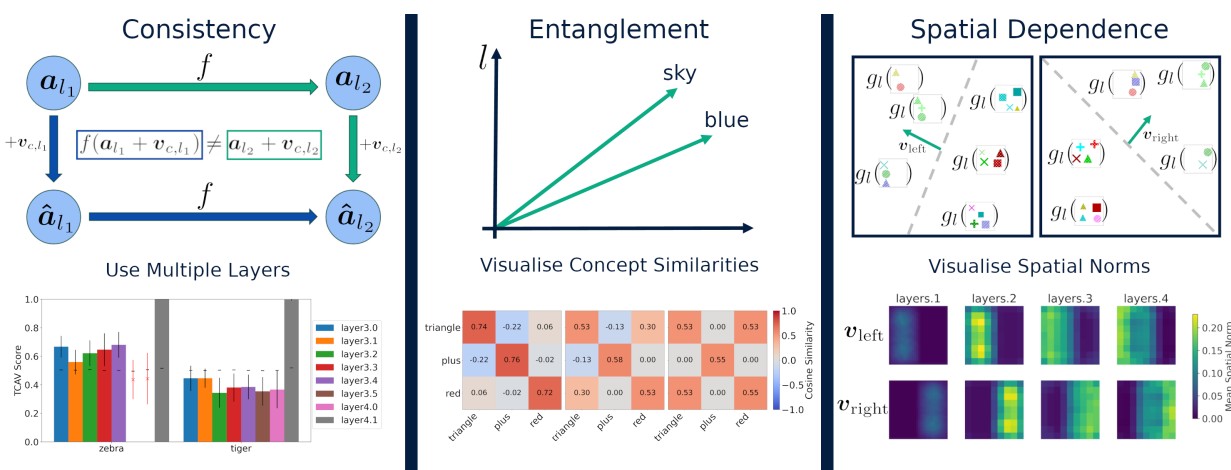

Figure 1: Concept Activation Vectors can be: inconsistent across layers, i.e., we cannot find two concept vectors in different layers that have the same additive effect (left), entangled (middle) and spatially dependent (right). The top panel illustrates each of these different properties. The bottom panels show our recommendations on how to mitigate the impact these effects can have: creating CAVs for multiple layers (left), verifying expected dependencies between related concepts (middle), and visualising spatial dependence (right).

where $\boldsymbol{v}_{c,l}$ is the normal vector of the hyperplane separating the activations $\mathbb{A}_{c,l}^{+}$ and $\mathbb{A}_{c,l}^{-}$, and $b_{c,l}$ is the intercept.[1]

To analyse a model's sensitivity to $\boldsymbol{v}_{c,l}$, Kim et al. (2018) introduce testing with CAVs (TCAV), which determines the model's conceptual sensitivity across an entire class. Let $\mathbb{X}_k$ be a set of inputs belonging to class $k$. The TCAV score is defined as

$$\text{TCAV}_{c,k,l} = \frac{\left|\{\boldsymbol{x} \in \mathbb{X}_k : S_{c,k,l}(\boldsymbol{x}) > 0\}\right|}{|\mathbb{X}_k|}, \tag{3}$$

where the directional derivative of the concept, $S_{c,k,l}$, is defined as

$$S_{c,k,l}(\boldsymbol{x}) = \lim_{\epsilon \to 0} \frac{h_{l,k}\left(g_l(\boldsymbol{x}) + \epsilon\boldsymbol{v}_{c,l}\right) - h_{l,k}\left(g_l(\boldsymbol{x})\right)}{\epsilon} = \nabla h_{l,k}\left(g_l(\boldsymbol{x})\right) \cdot \boldsymbol{v}_{c,l} \tag{4}$$

where $\nabla h_{l,k}$ is the partial derivative of the NN output for class $k$ with respect to the activation. The TCAV score measures the fraction of class $k$ inputs whose activation at layer $l$ is positively influenced by concept $c$. A statistical test comparing the scores of CAVs to random vectors is used to determine the concept's significance (see Appendix 10.1).

## 3 CAV Hypotheses

To use CAV-based explanation methods in practice, it is important to understand how they work. Therefore, we study three properties of CAVs and their effects on TCAV scores. We focus on these hypotheses as they provide insight into network representations and into the meaning encoded by concept vectors.

We formalise each property through a null hypothesis, which we provide evidence to reject later in the paper. In the following text, we use the typesetting `concept` to denote a concept.

### 3.1 Layer Consistency

In general, we want to understand *model* behaviour. However, CAVs explain whether a model is sensitive to a concept in a specific *layer*. In practice, analysing all layers may be computationally infeasible, and it

---

[1]Eq. 2 assumes that the linear classifier has hard boundaries. In practice, the classifiers typically achieve 80-95% accuracy.

is unclear which layers to choose. Therefore, our first hypothesis explores the relationship between CAVs found in different layers. Recall that the TCAV scores depend on the directional derivative: *how the model output changes for an infinitesimal change of the activations in the direction of a CAV*. By perturbing the activations in the direction of a CAV, we explore whether two concept vectors found in different layers can have the same affect on the model output. We refer to this property as *layer consistency* (see Figure 1 for a schematic overview).

**Definition 1 (layer consistency)** *Assume we have a function $f(\cdot)$ that maps the activations from layer $l_1$ into activations in layer $l_2$, where $l_1 < l_2$. Concept vectors, $\boldsymbol{v}_{c,l_1}$ and $\boldsymbol{v}_{c,l_2}$ are consistent across layers iff for every input $\boldsymbol{x}$ and corresponding activations $\boldsymbol{a}_{l_1}$ and $\boldsymbol{a}_{l_2}$, $f(\boldsymbol{a}_{l_1} + \boldsymbol{v}_{c,l_1}) = \boldsymbol{a}_{l_2} + \boldsymbol{v}_{c,l_2}$.*

If two CAVs are consistent across layers then they have the same downstream affect on the model when activations are perturbed in their direction, i.e., even though they are in different layers, they have an equivalent effect on the model output and therefore the model assigns them the same meaning. [2] Our first hypothesis is:

> ***Null Hypothesis 1 (NH1):*** *Concept vector representations are consistent across layers*

In §6.1 we formally explore this hypothesis, and perform empirical evaluations on the Elements and ImageNet (Deng et al., 2009) datasets. We show theoretically the conditions $f$ must meet for layer consistent vectors $\boldsymbol{v}_{c,l_1}$ and $\boldsymbol{v}_{c,l_2}$ to exist.

### 3.2 Entangled concept vectors

Consider the meaning encoded by a concept vector. We label a CAV using the corresponding label of the probe dataset. For example, a CAV may be labelled `striped` or `red`. This implicitly assumes that the label is a complete and accurate description of the information encoded by the vector. In practice, the CAV may represent several concepts – e.g., continuing the example above, the vector may encode both `striped` and `red` simultaneously. We refer to this phenomenon as *concept entanglement*. Mathematically, we formulate this as follows. A concept vector $\boldsymbol{v}_{c,l}$ is more similar to the activations corresponding to images containing the concept than activations for images not containing the concept, i.e. it satisfies

$$\boldsymbol{a}_{c,l}^{+} \cdot \boldsymbol{v}_{c,l} > \boldsymbol{a}_{c,l}^{-} \cdot \boldsymbol{v}_{c,l} \quad \forall \boldsymbol{a}_{c,l}^{+} \in \mathbb{A}_{c,l}^{+}, \boldsymbol{a}_{c,l}^{-} \in \mathbb{A}_{c,l}^{-}. \tag{5}$$

Assume we have concepts $c_1$ and $c_2$, with probe datasets $\mathbb{D}_{c_1}$ and $\mathbb{D}_{c_2}$, respectively. For each probe dataset, we find the activation sets: $\mathbb{A}_{c_1,l} = \{A_{c_1,l}^{+} \cup A_{c_1,l}^{-}\}$ and $\mathbb{A}_{c_2,l} = \{\mathbb{A}_{c_2,l}^{+} \cup \mathbb{A}_{c_2,l}^{-}\}$.

**Definition 2 (entangled concepts)** *A CAV $\boldsymbol{v}_{c_1,l}$ for concept $c_1$ is entangled with concept $c_2$ iff*

$$\boldsymbol{a}_{c_2,l}^{+} \cdot \boldsymbol{v}_{c_1,l} > \boldsymbol{a}_{c_2,l}^{-} \cdot \boldsymbol{v}_{c_1,l} \quad \forall \boldsymbol{a}_{c_2,l}^{+} \in \mathbb{A}_{c_2,l}^{+}, \boldsymbol{a}_{c_2,l}^{-} \in \mathbb{A}_{c_2,l}^{-} \tag{6}$$

Our second hypothesis explores concept entanglement:

> ***Null Hypothesis 2 (NH2):*** *A CAV represents only the concept corresponding to the concept label of its probe dataset*

If concepts are entangled, it is not possible to separate the model's sensitivity to one concept from its sensitivity to related concepts – therefore, if we measure the TCAV score for $c_1$, we will unknowingly incorporate the effect of $c_2$.

In §6.2 we demonstrate that visualising similarity matrices can be used to explore CAV entanglement and discuss how entanglement can affect TCAV.

---

[2] For simplicity we write the concept vectors without a scaling term, but in all experiments we scale by some small $\gamma$ and the mean norm of the activations in that layer – see Appendix 12.1 for details.

### 3.3 Spatial Dependence

Here, we explore the influence of spatial dependence on concepts. Let $\mathbb{D}_{c,\mu_1}$ and $\mathbb{D}_{c,\mu_2}$ denote two datasets containing the same concept but in different locations $\mu_1 \neq \mu_2$. By location we mean the location of the concept relative to the frame of the image. The exact form of this will depend on the specific dataset and concept in question. For example, in the Elements dataset we use binary labels such as left/right or top/bottom but more complex representations could be used (e.g. a segmentation map of which pixels contain a concept). For example, $\mathbb{D}_{c,\mu_1}$ may contain exemplars of `striped on the left` of the image, and $\mathbb{D}_{c,\mu_2}$ exemplars of the `striped on the right` of the image – an example is shown in fig. 2. As before, we construct latent representations $\mathbb{A}_{c,l,\mu_1}$ and $\mathbb{A}_{c,l,\mu_2}$ for datasets $\mathbb{D}_{c,\mu_1}$ and $\mathbb{D}_{c,\mu_2}$, respectively. Let $\boldsymbol{v}_{c,l}$ be the concept vector found using probe dataset $\mathbb{D}_{c,\mu_1}$.

**Definition 3 (activation spatial dependence)** *Let $\boldsymbol{a}_{l,i}$ be the activations corresponding to input $\boldsymbol{x}_i$ in layer $l$, and let $\mu_{c,i}$ be the location of concept $c$ in $\boldsymbol{x}_i$. A layer has a spatially dependent representation of a concept iff there exists some function $\phi$ which maps $\boldsymbol{a}_{l,i}$ to $\mu_{c,i}$ for all inputs $\boldsymbol{x}_i$:*

$$\exists \phi : \forall \boldsymbol{x}_i \in \mathbb{X}_c^+, \phi(\boldsymbol{a}_{l,i}) = \mu_{c,i} \tag{7}$$

Activation spatial dependence in a NN may be due to architecture design, training procedure and/or the training dataset. In CNNs, it is the natural consequence of the receptive field of convolutional filters containing different regions of the input. If the NN has spatially dependent activations and the probe dataset has a spatial dependence, it may be possible to create a concept vector with spatial dependence.

**Definition 4 (concept vector spatial dependence)** *A concept vector $\boldsymbol{v}_{c,l}$ is spatially dependent with respect to the locations $\mu_1$ and $\mu_2$ iff*

$$\boldsymbol{a}_{c,l,\mu_1}^+ \cdot \boldsymbol{v}_{c,l} > \boldsymbol{a}_{c,l,\mu_2}^+ \cdot \boldsymbol{v}_{c,l} \quad \forall \boldsymbol{a}_{c,l,\mu_1}^+ \in \mathbb{A}_{c,l,\mu_1}^+, \boldsymbol{a}_{c,l,\mu_2}^+ \in \mathbb{A}_{c,l,\mu_2}^+. \tag{8}$$

If a CAV is spatially dependent then, by the definition above, it is more similar to the activations from images containing the concept in a specific location. This means the CAV represents not only the concept label, but the concept label at a specific location, e.g. striped objects on the right of the image, rather than striped objects in general. As done for the other two properties, we propose a hypothesis and aim to reject it later in the paper:

> ***Null Hypothesis 3 (NH3):*** *Concept activation vectors cannot be spatially dependent*

We reject this hypothesis in §6.3 by analysing how the concept location in the probe dataset influences the spatial dependence of concept vectors. Rejecting NH3 motivates the introduction of *spatially dependent CAVs* (§ 6.3), which can be used to test if a model is translation invariant with respect to a specific concept and class.

## 4 Elements: A configurable synthetic dataset

To explore these hypotheses, we introduce a new synthetic dataset: Elements. In this dataset we can control: (1) the training dataset and class definitions, allowing us to influence model properties, such as concept correlation in the embedding space, and (2) the probe dataset, allowing us to test concept vector properties, such as concept vector spatial dependence. We further elaborate on these advantages in Appendix 11.

Figure 2 shows examples of images in the Elements datasets. Each image contains $n$ elements, where an element is defined by seven properties: colour, brightness, size, shape, texture, texture shift, and coordinates within the image. The dataset can be configured by varying the allowed combination of properties for each element. The ranges and configurations used for each property is given in Appendix 11.

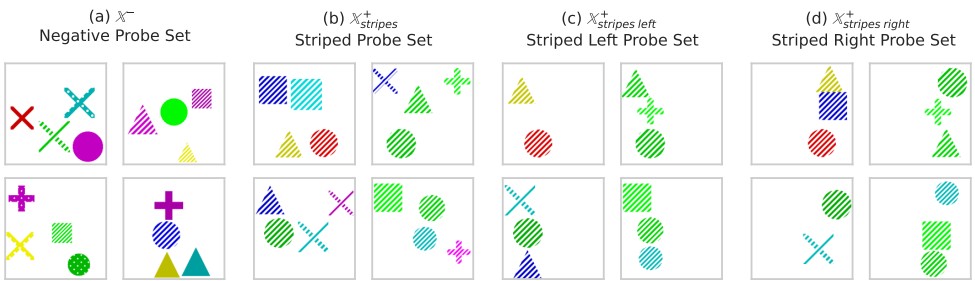

Figure 2: Example images from Elements probe datasets. (a) Negative probe set. A random selection of images – equivalent to images found in the model training set. (b) Positive probe set for `stripes`. (c) Positive probe set for `stripes on the left`. (d) Positive probe set for `stripes on the right`.

## 5 Related Work

**Concept Correlation and Entanglement**   Chen et al. (2020) discuss how concept vectors can be correlated, making it challenging to create a vector that solely represents one concept. While their work focuses on de-correlating concepts *during training*, we focus on analysing the impact of correlated concepts *after training* and show how they can lead to misleading explanations (§6.2). Fong & Vedaldi (2018) use cosine similarity to demonstrate that the similarity between concepts varies based on the vector creation method. In our work, we also use cosine similarity to compare concept vectors. The distinction lies in our focus on CAVs and the insights they provide into the dataset and model. There have been several works analysing correlated concepts in interpretable-by-design networks (Heidemann et al., 2023; Zarlenga et al., 2023). Our work complements these works by studying standard neural network architectures and the post-hoc explanations of TCAV. Raman et al. (2024) examine the effect of inter-concept relationships in CAVs and provide several useful metrics for measuring how well these relationships are represented, but they do not explore its effect on TCAV scores.

**Spatial Dependence**   Biscione & Bowers (2021) describe how CNNs are not inherently translation invariant but can learn to be (under certain conditions on the dataset). This finding challenges the common assumption that CNNs possess inherent translation invariance. Through *spatially dependent CAVs*, we demonstrate translation invariance with respect to a specific concept and class, rather than in general, providing more detailed information about a model. Raman et al. (2023) examine the locality of concept bottleneck models (CBMs) and find that, in some cases, CBMs make concept predictions using information far from the object of interest. In this work we examine post-hoc concept-based explanations, rather than predictions of an interpretable-by-design model. Additionally, our definition of spatial dependence is more related to whether we can learn CAVs which mean `stripes on the right of the image`, rather than whether a CAV is using information near/far from where the concept is located in the image.

**What concept representations does our analysis apply to?**   Most concept-based interpretability methods represent concepts as *vectors* in the activation space of a trained neural network (Kim et al., 2018; Fong & Vedaldi, 2018; Zhou et al., 2018; Ghorbani et al., 2019; Zhang et al., 2020; Ramaswamy et al., 2022b; Fel et al., 2023). However, some concept-based methods use different representations: individual neurons (Bau et al., 2017), regions of activation space (Crabbé & van der Schaar, 2022) or non-linear concepts (Bai et al., 2022; Li et al., 2023). Our work focuses on the properties of concept *vectors*.

**How is our work relevant in practice?**   To give insight into when the various properties may be relevant, we performed a review of computer vision papers which use CAVs in (1) the high-stakes applications of medical imaging (including skin cancer, skin lesions, breast cancer, and histology (Yan et al., 2023; Fürböck et al., 2022; Pfau et al., 2020)), and (2) computer vision research on models trained with well-known datasets (Krizhevsky, 2009; Lin et al., 2014; Wah et al., 2011; Zhou et al., 2017; Sagawa et al., 2020; Deng et al., 2009). A summary table can be found in Appendix 15. We found that the following papers could have benefited

from evaluating: consistency (Yan et al., 2023; Ramaswamy et al., 2022a; Fürböck et al., 2022; Yuksekgonul et al., 2023; Ghosh et al., 2023; Lucieri et al., 2020), entanglement (Yan et al., 2023; Ramaswamy et al., 2022a; Fürböck et al., 2022; Yuksekgonul et al., 2023; Ghosh et al., 2023; Graziani et al., 2020; McGrath et al., 2022; Lucieri et al., 2020; Pfau et al., 2020), and spatial dependence (Yan et al., 2023; Ramaswamy et al., 2022a; Fürböck et al., 2022; Yuksekgonul et al., 2023; Ghosh et al., 2023; McGrath et al., 2022; Lucieri et al., 2020; Pfau et al., 2020). We provide a detailed example, using the application of skin cancer diagnosis (Yan et al., 2023), in § 7.

**Datasets**  While several datasets have been introduced for evaluating interpretability methods, they differ from ours in a few key ways. There are three questions we need our dataset to help answer:

1. Is the concept represented in the network?

2. Is the concept used for the network's prediction?

3. How does the network represent correlated concepts?

Existing datasets either do not allow insight into all three, or they have other practical reasons for being unsuitable. The Benchmarking Interpretability Method (BIM) (Yang & Kim, 2019) inserts objects into scene images. While it benefits from utilizing real images and complex concepts (dog or bedroom), it also presents challenges. One drawback is that relying on real images makes it challenging to establish the ground truth relationship between concepts and class predictions or to know the similarities between concepts. As such, it does not give us insight into (2) or (3). The CLEVR dataset (Johnson et al., 2016) could give insight into all 3, but because it renders 3D shapes it is too slow for our purposes. Elements generates images in 0.004s compared to CLEVR's 4s. This translates to a significant time saving when more data is required – 4s for 1000 images with Elements versus 1h for CLEVR. Analyzing CAV properties is our core focus. Elements, with its speed and flexibility, allows us to create the many different dataset versions required for experiments. The Navon and Trifeature datasets, used by Hermann & Lampinen (2020) to study feature representations, could also give insight into the three questions, with associated concepts of shape, color and texture relating to each image. However, there is only one large object in each image so our experiments on spatial dependence would not have been possible. The synthetic dataset in Yeh et al. (2020) is similar to our dataset but it was designed for concept discovery, featuring images where each object corresponds to a single concept (shape). In our dataset, each object contains multiple concepts, allowing us to create associations between them. We focus on explanation faithfulness by ensuring that the concepts must be used correctly by the model to achieve a high accuracy. So, for an accurate model, we have a ground truth understanding of how each concept is used. dSprites Matthey et al. (2017) and 3D Shapes Burgess & Kim (2018) are probably the most similar datasets to ours but Elements is a far larger dataset. For example, even for the simple version of the Elements dataset, there are approximately $10^{10}$ different images after just a single object has been placed in the image. This is considerably larger than the $737,280$ and $480,000$ total images in the dSprites and 3D Shapes datasets, respectively. The additional complexity of Elements means an NN needs to approximate a more complex function and having multiple objects per image ensures the model has to learn object/location-based representations of the concepts, rather than image-based, e.g., to answer whether a striped triangle is present it is not sufficient to simply determine if stripes occur anywhere in the image. An extended literature review can be found in Appendix 15.

## 6 Results: Exploring Concept Vector Properties

We explore the hypotheses on consistency (NH1), entanglement (NH2) and spatial dependency (NH3) in § 6.1, § 6.2 and § 6.3, respectively. We perform experiments using CAVs on the Elements and ImageNet datasets. Implementation details can be found in Appendix 10.

### 6.1 Consistent CAVs

**Theory**  We begin investigating NH1, which states that CAVs are consistent across layers, i.e. $f(\boldsymbol{a}_{l_1} + \boldsymbol{v}_{c,l_2}) = \boldsymbol{a}_{l_2} + \boldsymbol{v}_{c,l_2}$. Let $\hat{\boldsymbol{a}}_{l_1}$ and $\hat{\boldsymbol{a}}_{l_2}$ be linear perturbations to the activations in layers $l_1$ and $l_2$, respectively:

$$\hat{\boldsymbol{a}}_{l_1} = \boldsymbol{a}_{l_1} + \boldsymbol{v}_{c,l_1} \tag{9}$$

$$\hat{\boldsymbol{a}}_{l_2} = \boldsymbol{a}_{l_2} + \boldsymbol{v}_{c,l_2} = f(\boldsymbol{a}_{l_1}) + \boldsymbol{v}_{c,l_2} \tag{10}$$

We want to investigate if $\boldsymbol{v}_{c,l_1}$ and $\boldsymbol{v}_{c,l_2}$ have the same effect on the activations (and hence the model), i.e. if:

$$f(\hat{\boldsymbol{a}}_{l_1}) = \hat{\boldsymbol{a}}_{l_2} \tag{11}$$

$$f(\boldsymbol{a}_{l_1} + \boldsymbol{v}_{c,l_1}) = f(\boldsymbol{a}_{l_1}) + \boldsymbol{v}_{c,l_2}. \tag{12}$$

Assuming $f$ is continuous and differentiable, in Appendix 9 we prove the result that Eq. 12 can hold if and only if $f$ is equal to

$$f(\boldsymbol{a}_{l_1}) = g(\boldsymbol{a}_{l_1}) + M\boldsymbol{a}_{l_1} + \boldsymbol{b}. \tag{13}$$

Where $g(\cdot)$ is a periodic function with period $\boldsymbol{v}_{c,l_1}$ and $M \in \mathbb{R}^{m_{l_2} \times m_{l_1}}$ is a non-zero linear term with constant $\boldsymbol{b} \in \mathbb{R}^{m_{l_2}}$. More intuitively, we can obtain layer consistent vectors $\boldsymbol{v}_{c,l_1}$ and $\boldsymbol{v}_{c,l_2}$ if and only if $f$ is composed of a periodic function with period $\boldsymbol{v}_{c,l_1}$ and a non-zero linear term $M$. In principal, $f$ could approximate a function of this form since neural networks are universal approximators (Sonoda & Murata, 2017). However, it seems reasonably unlikely that even if the model was modeling a periodic function it would have a period of exactly the same direction as a CAV. Throughout the rest of this section we provide empirical evidence that, in practice, layer consistent CAVs are not found.

**Experiments**   Our goal is to investigate the question *are the concept vectors found using TCAV consistent across layers?* We measure the consistency of two perturbations using the consistency error:

$$\epsilon_{consistency} = ||f(\hat{\boldsymbol{a}}_{l_1}) - \hat{\boldsymbol{a}}_{l_2}|| = ||f(\boldsymbol{a}_{l_1} + \boldsymbol{v}_{c,l_1}) - (\boldsymbol{a}_{l_2} + \boldsymbol{v}_{c,l_2})|| \tag{14}$$

In our experiments, we use a scaling term to reduce the size of $\boldsymbol{v}_{c,l_1}$ and $\boldsymbol{v}_{c,l_2}$ to ensure the perturbed activation remains in distribution – see Appendix 12.1 for details. If two perturbations have a consistency error of 0, then they have the same effect on the model. We include the following benchmarks:

*Optimised CAV* (lower bound): TCAV may not find a $\boldsymbol{v}_{c,l_2}$ that has a consistency error of 0 with $\boldsymbol{v}_{c,l_1}$. Therefore, we use gradient decent on $\boldsymbol{v}_{c,l_2}$ to minimise the consistency error, which acts as a lower bound.

*Projected CAV*: the error between $f(\boldsymbol{v}_{c,l_1})$ and $\boldsymbol{v}_{c,l_2}$, which measures how consistent the vectors are when projected into the next layer. If $f(\cdot)$ conserves vector addition, the projected CAVs would have 0 error.

*Random* (upper bound): We include two benchmarks. Random CAVs found using probe datasets containing random images, and a Random Direction vector: $\boldsymbol{v}_{c,l_2} \sim \text{Uniform}(-1,1)$. If the consistency error is similar to random, it suggests that the CAVs between layers are as similar to each other as random directions.

Figure 3 shows the $\epsilon_{consistency}$ for different $\boldsymbol{v}_{c,l_2}$ across different training runs (see Appendix 12 for details). The concept CAV obtains a nonzero consistency error, suggesting that CAVs across different layers are not consistent. When we compare it with the benchmarks, we find:

- The consistency error for the optimised CAVs is lower, implying that the standard approach to find CAVs does not find optimally layer consistent CAVs. However, the nonzero error for optimised CAVs suggests it is not possible to find consistent vectors across these layers.

- As expected, the projected CAVs have a nonzero error, indicating that vector addition is not preserved.

- The random CAVs have a higher error, suggesting the concept CAVs are more similar than random vectors.

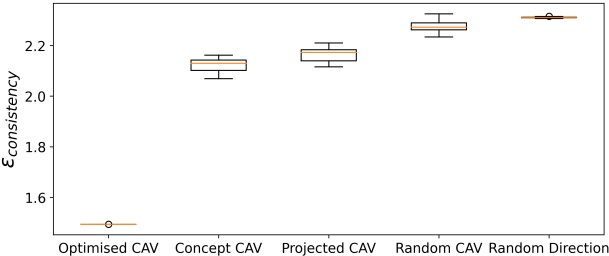

Figure 3: Empirical evidence for inconsistent CAVs across layers. The consistency error for different $\boldsymbol{v}_{c,l_2}$ for `striped` in the penultimate convolutional layer of a ResNet-50 trained on ImageNet. The optimised CAV acts as lower bound, whereas the random CAV and Direction act as baselines that provide an intuitive upper bounds. Concept CAV: `striped` CAVs, trained as normal. Projected CAV: `striped` CAVs from layer $l_1$ projected into layer $l_2$, $f(\boldsymbol{v}_{c,l_1})$.

The inability to find consistent concept vectors across layers suggests that the directions encoded by CAVs in different layers are not equivalent; instead we speculate that they represent different components of the same concept. This result in unsurprising given that previous works have demonstrated that model representations are more complex later in the NN (Mordvintsev et al., 2015; Olah et al., 2017; Bau et al., 2017), therefore it is unlikely that the same aspects of a concept are represented in different layers (discussed further in Appendix 12.3). Consequentially, TCAV scores across layers can vary as they perform different tests – they measure the class sensitivity to a different version of the concept.

Figure 5c shows that concept vectors found in different layers of a model can give contradictory TCAV scores (further examples available in Appendix 12.4). In the Elements dataset, shape concepts are encoded in each layer as the test accuracy for each layer is above 93%. Therefore, we expect to be able to use TCAV on each of these layers. However, the TCAV scores for `cross` in the Elements dataset contradict each other across 'layers.3' and 'layers.4', suggesting a positive and negative influence, respectively. This contradiction makes it difficult to draw a conclusion about the model's class sensitivity to `cross`.

On the right of fig. 5c, we show the TCAV scores for `striped` for various classes in a ResNet-50 model trained on ImageNet. The accuracy for the `striped` vectors in ImageNet is above 96% for all layers tested, suggesting that the concept is encoded by the model in each of the layers. As in Elements, we do not observe consistent TCAV scores across layers. Instead, we observe a large change in the TCAV scores for `striped` in the penultimate layer, compared to earlier layers. 'layer4.1' suggests `striped` positively influences the likelihood of the classes tiger and leopard. However, earlier layers suggest that the class is not is not sensitive to the concept. This shows how, depending on the layers that are tested, different conclusions can be drawn.

In order to determine how often TCAV scores are inconsistent across layers, we introduce a new metric, the TCAV layer consistency score. It is a measure of how well the significant TCAV scores for some concept, $c$, and class, $k$, agree with each other across different layers $l_i, i \in 1 \ldots L$. It is based on how many layers are on the same side of the null TCAV score. The null TCAV score is the mean TCAV score for a set of random CAVs (see Appendix 10.1 for more detail). Mathematically the metric, $S_{\text{consistent}}$, is defined as

$$S_{\text{consistent}} = |2((\frac{1}{L}\sum_i \text{TCAV}_{c,k,l_i} > \text{TCAV}_{r,k,l_i}) - 0.5)| \tag{15}$$

where $r$ indicates the TCAV score is calculated using random CAVs. It is difficult to make confident statements about the sensitivity of the model when the layer consistency score is close to 0 as it indicates the TCAV scores from different layers tend to disagree on the direction of the sensitivity.

We obtain a mean TCAV consistency score of 0.841 across all concepts/classes/layers for Elements and a mean TCAV consistency score of 0.868 for a selection of 6 classes and 16 concepts for a ResNet-50 trained on ImageNet (See Appendix 12.5 for further details). The scores indicate that although complete disagreement across layers is not common (5.5% and 2.1% of scores were equal to 0 for Elements and ImageNet,

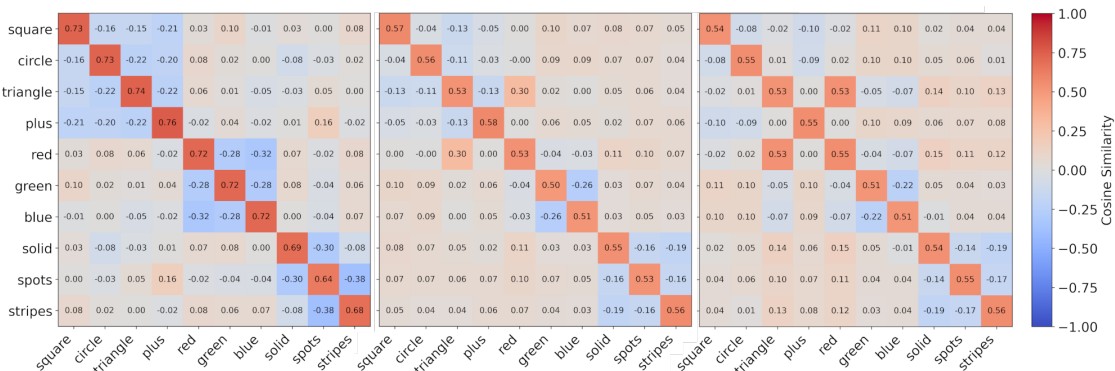

Figure 4: Cosine similarities demonstrating entangled concepts. Mean pairwise cosine similarities for all concepts from different versions of the simple Elements dataset, with an increasing association between `red` and `triangle` from left to right: $\mathbb{E}_1$, $\mathbb{E}_2$ and $\mathbb{E}_3$.

respectively), it can occur, and a not insubstantial number of concepts/classes have a reasonable amount of disagreement across layers (23% and 14% of scores were $\leq 0.5$ for Elements and ImageNet, respectively).

## 6.2 Entanglement

Different concepts may be associated with each other. For example, consider `blue` and the `sky` – a fundamental aspect of the sky is that it is often blue. These concepts are inherently linked and should not be treated as independent. This section will discuss how to discover these associations using CAVs and the implications for TCAV.

To explore entanglement, we quantify and visualize concept associations by computing average pairwise cosine similarities between CAVs (we compute multiple CAVs for each concept). We investigate three models trained on different versions of the Elements dataset. Each dataset is identical aside from the association between `red` and `triangle`:

$\mathbb{E}_1$: each combination of colour, shape and texture is equally likely,

$\mathbb{E}_2$: the only shape that is red is triangles,

$\mathbb{E}_3$: the concepts of red and triangle only ever co-occur.

In fig. 4 we show one plot for each dataset. For $\mathbb{E}_1$, we observe no positive association between the concepts. In $\mathbb{E}_2$, we observe a small positive association between the triangle and red concepts. Lastly, in $\mathbb{E}_3$, the cosine similarity between the `red` and `triangle` CAVs approaches the similarity of the concept with itself. The trend between $\mathbb{E}_1$, $\mathbb{E}_2$ and $\mathbb{E}_3$ is likely due to the underlying association between the `red` and `triangle` increasing. We perform similar analyses on ImageNet in Appendix 13.

Interestingly, we often observe a negative cosine similarity between mutually exclusive concepts. The model has encoded concepts that cannot co-occur (e.g., each element can only have a single colour) in directions negatively correlated with each other. The presence of the `red` diminishes the likelihood of the `blue` or `green` being present, and by having these concepts negatively associated with each other the model builds in this reasoning. This means that the `red` CAV does not solely signify `red`, it also encapsulates `not blue` and `not green`.

Next, we investigate the effect of entangled concept vectors on TCAV scores. We analyse the TCAV scores for the 'striped triangles' class in $\mathbb{E}_1$ and $\mathbb{E}_2$. The class label depends solely on the presence `stripes` and `triangle`. Therefore, we expect all other concepts to obtain low TCAV scores (indicating negative

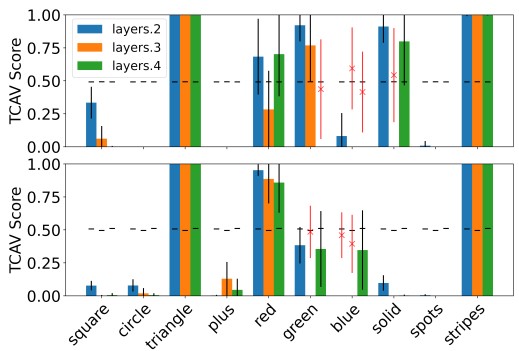 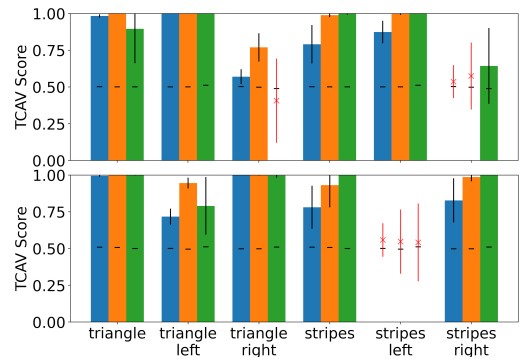

(a) Entangled CAVs increase `red` TCAV scores for $\mathbb{E}_2$. TCAV scores for all concepts in $\mathbb{E}_1$ (top) and $\mathbb{E}_2$ (bot) for the class of striped triangles.

(b) TCAV scores are spatially dependent for the 'striped triangles on the left' (top) and 'striped triangles on the right' (bot) classes in Elements.

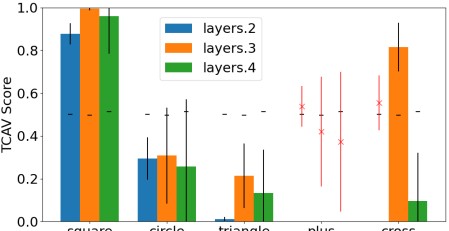 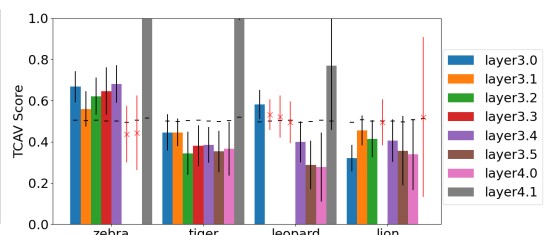

(c) Inconsistent TCAV scores across layers. Left: TCAV scores for shape concepts for the 'solid red squares' class in Elements. Right: TCAV scores for `striped` for a subset of ImageNet classes.

Figure 5: Consistency, entanglement, spatial dependence can affect TCAV scores. The standard deviation is black or red for significant and insignificant results, respectively. The null for each layer is shown as a horizontal black line.

sensitivity), as their presence makes the class less likely, or insignificant TCAV scores, if the concept is uninformative.[3]

The results for $\mathbb{E}_1$ and $\mathbb{E}_2$ are shown on the top and bottom of fig. 5a, respectively. For $\mathbb{E}_1$ (the unaltered dataset), we find that only the `stripes` and `triangle` vectors have a high TCAV score across multiple layers. For $\mathbb{E}_2$ (the altered dataset), however, the model appears to be sensitive to `red`, `triangle` and `stripes`, with high TCAV scores for each. This is due to the association between the `red` and `triangle` CAVs. $2,374/5,000$ images in the test dataset contain striped triangles. None of these are incorrectly classified, so it is unlikely that the model uses the red concept for its prediction. Instead, the association between CAVs causes a misleadingly high TCAV score for the red concept. In conclusion, associated CAVs can lead to misleading explanations.

### 6.3 Spatial Dependence

Finally, we investigate NH3: *are CAVs spatially dependent?* In the case of a convolutional based neural network, where the activations are of shape $H \times W \times D$, we can reshape the CAVs back into the original shape of the activations, and compute the channel-wise norm as follows:

$$\mathbf{S}_{c,l} = \|\text{reshape}(\boldsymbol{v}_{c,l}, (H, W, D))\|_2, \tag{16}$$

where $\mathbf{S}_{c,l} \in \mathbb{R}^{H \times W}$, and $\| \cdot \|_2$ is the $L_2$ norm across the channel dimension. We refer to this array as the spatial norms of the CAV.

---

[3]assuming that the model uses each concept correctly

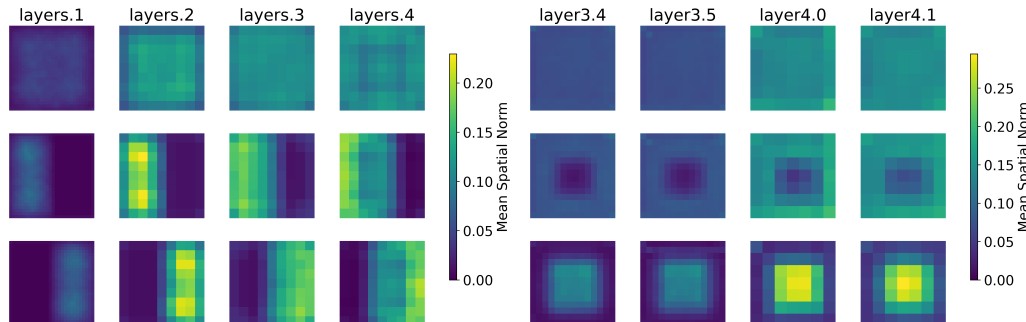

Figure 6: Spatial norms reflect the spatial dependence of the probe dataset. Left: Mean spatial norms for `red` (top), `red left` (middle) and `red right` (bottom) for Elements. Right: Mean spatial norms across for `striped` (top), `striped edges` (middle) and `striped middle` (bottom) for ImageNet.

If a CAV's spatial norm varies substantially across the $(H, W)$ dimensions, it indicates that the CAV is spatially dependent (see Appendix 14.2 for an explanation). Visualising a CAV's spatial norms shows us which regions contribute most to the directional derivative and, consequently, to the TCAV score.

To create spatially dependent CAVs, we constructed spatially dependent probe datasets for Elements and ImageNet where we either restricted the location of the concepts or greyed out parts of the image – see fig. 2 for examples and Appendix 14.1 for further details.

When a spatially independent probe dataset is used to create CAVs, as in the top row of fig. 6, the spatial norms are uniform, suggesting the CAVs are not spatially dependent[4]. However, when the probe dataset exhibits spatial dependence, so do the resulting CAVs. The regions of near-zero norm indicate that the corresponding spatial regions of the gradients do not contribute to the directional derivative and, consequently, to the TCAV score.

Next, we investigate the question *does the model have a different conceptual sensitivity depending on the concept's location in the input image?* As CAVs operate in the activation space of a specific layer, we can show that a model is not translation invariant if:

1. The model has activation spatial dependence, i.e. pixels in different locations affect the activations differently.

2. Each depth-wise slice of the activations, of shape $(1, 1, D)$, affects the logit output differently.

Both of these components affect the TCAV score. (1) influences $\boldsymbol{v}_{c,l}$ and (2) influences $\nabla h_{l,k}\left(g_l(\boldsymbol{x})\right)$. For (1), fig. 6 demonstrates that the model has activation spatial dependence as the locations with the highest spatial norms approximately correspond to the location of the concept in the image space.

To address (2), we compute the TCAV scores for different sets of spatially dependent CAVs to determine if the sensitivity of the model changes depending on the concepts location. To investigate this, we created spatially dependent classes in the Elements dataset, where the class depends on what concepts are present *and* on where they are in the image, such as 'striped triangles on the left'. We use spatially dependent CAVs to show that a model is not translation invariant with respect to `striped` or `triangle` in fig. 5b. Here, we discuss the results for the class of 'striped triangles on the left'. The TCAV scores for `striped`, `triangle`, `striped left` and `triangle left` are high, indicating a positive influence of these concepts on the class. However, the `striped right` and `triangle right` TCAV scores often do not differ significantly from the null scores, providing no evidence to suggest the model is sensitive to these concepts. The difference between the left and right biased TCAV scores indicates that the model is not translation invariant with respect to these concepts as the model's sensitivity depends on where the concept is present in the image input space.

---

[4]the individual CAVs may still be spatially dependent, but this cancels out across training runs. See Appendix 14.3 for details.

Overall, this suggests that we can use CAVs to detect model translation invariance. See Appendix 14.6 for examples on ImageNet and in Appendix 14.7, even though we cannot use spatial norms to visualise it, some preliminary evidence that spatially dependent CAVs can exist for transformer-based architectures.

# 7 Practitioner Recommendations

Our results have shown that failure to appropriately consider consistency, entanglement, and spatial dependence may result in drawing incorrect conclusions when using TCAV. Therefore, we recommend the following:

- *Consistency*: creating CAVs for multiple layers, rather than a single one;

- *Entanglement*: (1) verifying expected dependencies between related concepts, and (2) being mindful that a positive TCAV score may be due to concept entanglement;

- *Spatial Dependence*: visualising concept vector spatial dependence using spatial norms.

In § 5, we provided example papers do not consider these properties but may be influenced by them. To provide a concrete example, we examine the use-case of Yan et al. (2023) which uses CAVs in the context of skin cancer diagnosis. Below we demonstrate how our recommendations could have been used and how the analysis impacts the conclusions drawn.

**Consistency**  The authors use CAVs on a single layer. As discussed in § 6.1 and § 12, different layers can represent different aspects of the same concept. To have a better understanding of the overall effect of the concept on the model, CAVs should be created for multiple layers.

**Entanglement**  There are multiple concepts which have opposed meanings, for example `regular streaks` and `irregular streaks`, or `regular vascular structures` and `irregular vascular structures`. As such, we expect the cosine similarities between the CAVs to confirm that these concepts are negatively correlated (or less similar to each other than to other concepts).

**Spatial Dependence**  Some of the concepts have expected spatial dependencies, for example, `dark borders` and `dark corners`. Spatial norms could be used to confirm these spatial dependencies exist. Equally, for concepts such as the presence of a `ruler`, the spatial norms could confirm the CAVs have no overall spatial dependence.

## 7.1 Experiment Setup

To further this example, we run illustrative experiments on a similar dataset to Yan et al. (2023). The dataset used in Yan et al. (2023) is not publicly available.

**Model**  We finetune a ResNet50 (He et al., 2016) pretrained on ImageNet (Deng et al., 2009) on the ISIC 2019 dataset (Tschandl et al., 2018; Codella et al., 2017; Combalia et al., 2019) for the binary classification of melanoma. We use a binary cross entropy loss and the Adam optimiser (Kingma & Ba, 2015), training until convergence of validation loss to achieve an area under the receiver operating characteristic curve (AUC) of 0.91 on the validation split.

**CAVs**  For the CAVs, as in Yan et al. (2023), we use the derm7pt dataset (Kawahara et al., 2019). There are 12 clinical concepts which have been expertly labelled within the dataset. We hand labelled three additional concepts, which were used in Yan et al. (2023), of `dark corners`, `dark borders` and `ruler` which are possible confounders for the model. We defined `dark corners` as any image with a circular aperture which left the corners of the image black, `dark borders` as any image containing rectangles of blacked out areas and `ruler` as any image containing a ruler. For each of the medical concepts, there are three labels: typical/regular, atypical/irregular, and absent. When training CAVs for these concepts, we either used random images or

images with the label of 'absent' as the negative probe dataset. We used the 'absent' labels because using random images as the negative set gave low-quality CAVs, with accuracies of 50-65% (see fig. 7). It is unclear from Yan et al. (2023) what they used for negative sets. For training the CAVs we used 70 images per concept and used 30 different random seeds for the random negative probe set to get 30 CAVs per concept. Yan et al. (2023) do not use TCAV, so this setup is different from the original paper.

## 7.2 Results

As shown in fig. 7, initial results with random CAVs gave poor performance for the medical concepts, so we used the 'absent' category for each class as the negative set. This gave better performances with accuracy between 60-75% for the medical concepts but this is still far lower than for the confounders at 80-90%. We hypothesise that this is due to the simplicity of the confounding concepts. This is supported by the accuracy reported by Yan et al. (2023), where they also obtained a lower CAV accuracy for the medical concepts. The accuracy for the confounding concepts of `dark borders` and `dark corners` drops in later layers. This is likely due to the spatial nature of the concepts – an idea further supported by fig. 10 where the CAVs in later layers have reduced spatial distinction.

Below, we examine how the three CAV properties analysed in this paper affect the TCAV scores and the conclusions you can draw from them.

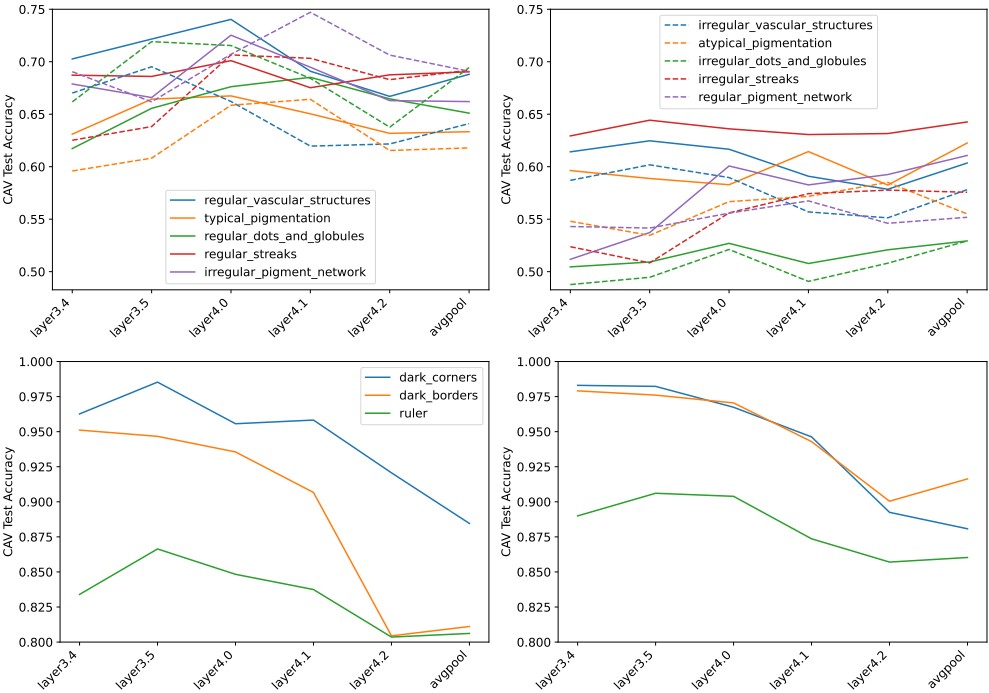

Figure 7: Mean CAV test accuracies for the melanoma use-case. Top: Medical concepts where random images (right) or images where the concept is labelled as absent (left) are used in the negative probe dataset. Bottom: Potential confounders where CAVs were trained with (right) and without (left) a flip augmentation.

**Consistency** The TCAV scores for many of the medical concepts (fig. 8a) are consistent across layers, irrespective of if random images or 'absent' images were used for the negative probe set. The consistent scores provide more confidence in using them to explain the model's behaviour as repeated significance tests are performed indicating the model has the same sensitivity to the concept. In terms of understanding the model, the scores provide some evidence that it operates similar to human experts, as the TCAV scores for the atypical/irregular medical concepts are high for the malignant class, as expected, and the confounding concepts are often not significant (prior to using a flip augmentation - see the spatial dependence section below for discussion), providing little evidence that the model is sensitive to the confounders.

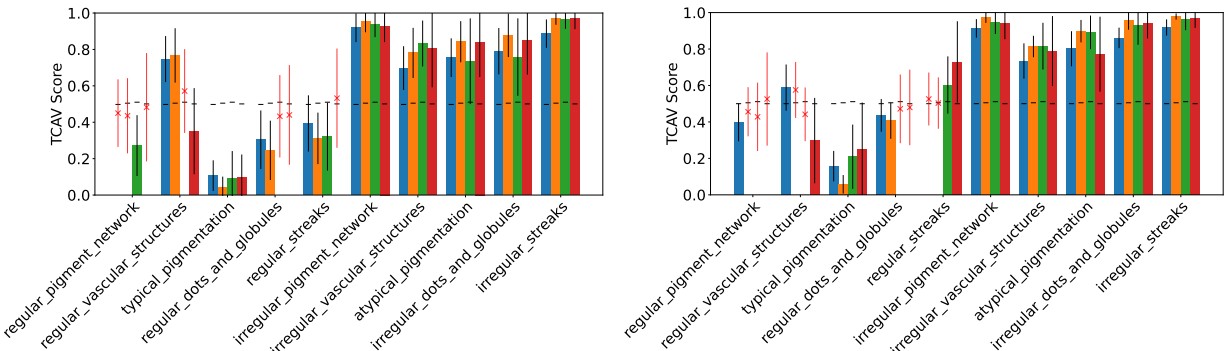

(a) TCAV scores are not qualitatively different for CAVs of differing accuracy. TCAV scores for medical concepts where random images (left) or images where the concept is labelled as absent (right) are used in the negative probe dataset.

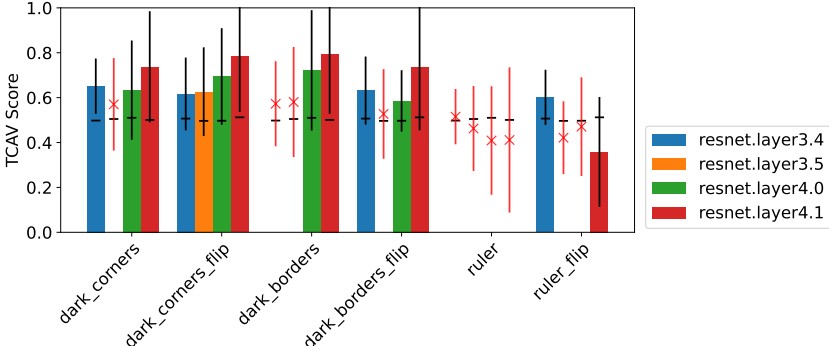

(b) CAVs trained with an augmentation were significant more often. TCAV scores for potential confounders where CAVs were trained with and without a flip augmentation.

Figure 8: TCAV scores for the melanoma use-case. The standard deviation is black or red for significant and insignificant results, respectively. The null for each layer is shown as a horizontal black line.

**Entanglement**  Interestingly, for the CAVs with 'absent' labelled images in their negative probe dataset (the right of fig. 9), the CAVs with opposed meanings often appear to be the most similar to each other. For example, `regular vascular structures` has a negative or zero similarity with all concepts except `irregular vascular structures` (with a similarity of 0.11). We hypothesise that this is because the two concepts share the same negative set. This hypothesis is supported by Ramaswamy et al. (2022a) where, while they did not discuss changing solely the negative set, they did show that CAVs are sensitive to the choice of probe dataset. In addition, if compared to the similarities between CAVs trained using random images as the negative set (the left of fig. 9), we see that this pattern disappears. The higher similarity between concepts which have opposite meanings suggests that the CAVs do not represent the concepts they are labelled for. Therefore, in this case, we do not believe the TCAV scores can be interpreted as we do not have confidence the CAVs represent their desired concept. This example highlights a more general problem that the negative probe set can have a substantial affect on the resulting CAV, even though it is the positive probe set that is designed to represent the concept.

**Spatial Dependence**  The spatial norms in fig. 10 show a clear spatial dependence in the center for each of the medical concepts across all layers. This aligns with expectations, as the dataset requires the skin lesion to be centered in the image and each concept is related to the appearance of the lesion. The `dark corners` and `dark borders` concepts, however, show deviations to this pattern in the earlier layers, with `dark corners` having high spatial norms in the corners and `dark borders` high spatial norms in the top. This is desirable, as these confounding concepts depend on features at the edge of the image, away from the lesion. Along with the accuracies in fig. 7, this suggests that the CAVs in earlier layers better represent these

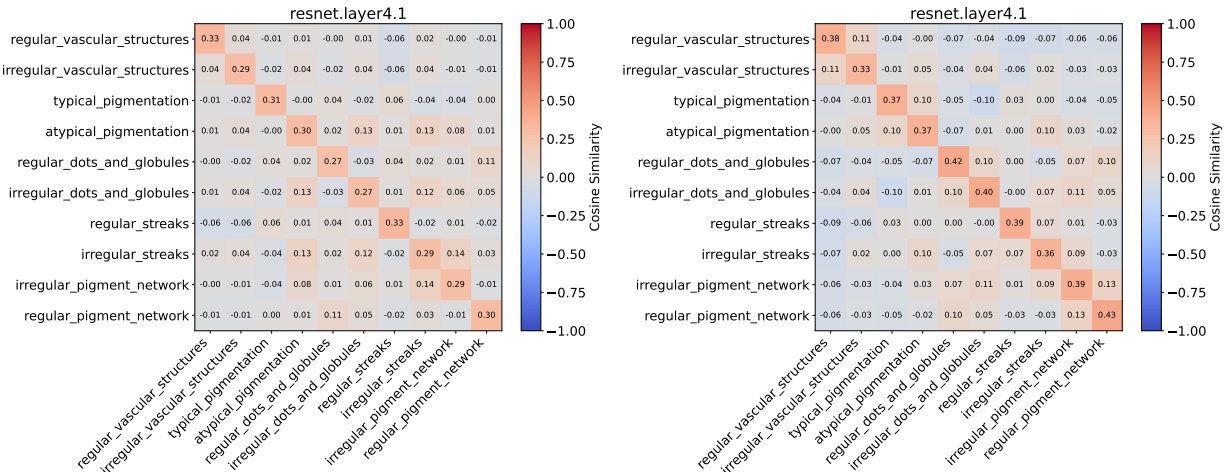

Figure 9: Cosine similarity matrix for CAVs of different concepts from derm7pt when random images (left) or images where the concept is labelled as absent (right) are used in the negative probe dataset.

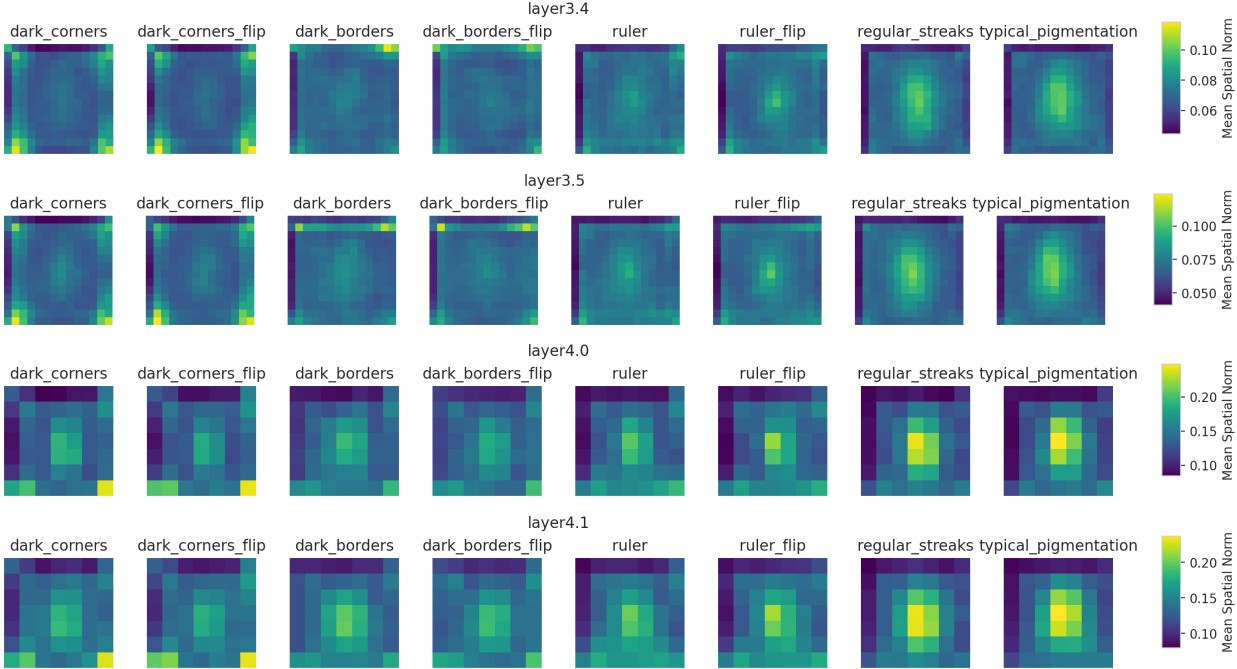

Figure 10: Mean CAV spatial norms for a selection of CAVs from the melanoma use-case.

spatially dependent concepts. For `dark borders`, however, we hypothesise that the high spatial norms in just a single direction suggest that the CAVs are not a good representation of dark borders in locations other than the top. Therefore, we retrained CAVs for each of the confounding concepts but with an augmentation applied to the probe dataset to randomly flip the images in the horizontal and/or vertical direction. This removes any bias that there may be in the probe dataset for the concept to be in one particular direction. Figure 10 shows that the CAVs trained for `dark borders` with an augmentation (`dark borders flip`) had only minor improvements for their spatial norms, with some layers being slightly less uni-directional. Figure 7, however, shows that the accuracy for each of the CAVs improved. This suggests that there was an issue with bias in the probe dataset and by using a flip augmentation we create CAVs which better represent the

concepts, but, surprisingly, the activations of the model encode the `dark borders` concept in a spatially unsymmetrical manner.

The TCAV scores for CAVs trained with the augmentation (fig. 8b) obtain significance in more layers but the results are still fairly inconclusive. For example, the TCAV scores for `ruler` in layer3.4 and layer 4.2 are above/below the null, respectively. This means that layer3.4 suggests a positive influence, whereas layer4.2 suggests a negative influence. With less than a 1% difference in accuracy between the CAVs of the two layers it is not clear which layer we might trust more and so no conclusive statements can be said about the influence of `ruler` on malignancy predictions.

### 7.3   Summary

The poor accuracy of medical concepts when using random images in the negative probe set required the use of the 'absent' category instead. However, the similarity matrices in fig. 9 suggest that the CAVs trained using the 'absent' category do not represent the desired concepts. Therefore, we do not think either set of medical CAVs can produce meaningful TCAV scores.

For the confounders, however, the CAVs had high accuracy and the spatial norms indicated spatial dependence in concepts we expect spatial dependence, providing evidence that these CAVs are suitable to use. However, the spatial dependence seemed directional, and so an augmentation was added to flip the probe images horizontally/vertically. Although this did not substantially change the spatial norms, it improved CAV accuracy and increased the number of layers for which we had significant TCAV scores. From these scores, it appears the model is sensitive to `dark corners` although the evidence is weak with large standard deviations in TCAV score and for `ruler` we found inconclusive results with inconsistent scores across layers.

This use-case demonstrates the importance of analysing consistency, entanglement and spatial dependence of CAVs, alongside more typical evaluations such as CAV accuracy and statistical significance, in order to understand CAV-based explanations and the conclusions you can draw from them. Our experiments provide an example for practitioners to follow in their own experiments with CAVs.

## 8   Conclusion and Future Work

In this work, we explore three key properties that influence concept activation vectors (CAVs): consistency, entanglement and spatial dependence. First, we derive conditions under which CAVs in different layers are not consistent and substantiate our findings with empirical evidence. This sheds light on why CAV-based explanations methods can give conflicting conclusions across layers. Next, we introduce visualisations designed to facilitate the exploration of associations between concepts within a dataset and model. Lastly, we show that spatial dependence impacts CAVs, and introduce a method that can be used to detect spatial dependence within models. We provided clear recommendations on how to mitigate the impact of these properties on CAV-based explanations and demonstrated how to use those recommendations for a medical imaging use-case. The CAV properties were explored using a synthetic dataset, Elements, where custom probe datasets can easily be created to analyse properties of interest. We release this dataset to help further explore this problem space.

In the introduction, we cited several interpretability methods that employ vector representations to convey semantically meaningful concepts. Our study has illuminated certain properties and consequential outcomes arising from these vector-based approaches. In future research, the characteristics inherent in alternative forms of representation, such as clusters within activation space (Crabbé & van der Schaar, 2022), should be investigated and the relative merits assessed.

### Acknowledgments

We appreciate both the members of OATML and the Noble group for your support and discussions during the project, in particular Andrew Jesson. We are also grateful to Been Kim for your thoughts and feedback on our work. Thank you to Ben Eastwood for our discussions on mathematical proofs. We also extend our thanks to the reviewers and the action editor of TMLR for their invaluable feedback and constructive

comments which have enhanced the quality of the resulting paper. A. Nicolson is supported by the EPSRC Centre for Doctoral Training in Health Data Science (EP/S02428X/1). L.S. is supported by the EPSRC Centre for Doctoral Training in Autonomous Intelligent Machines and Systems grant (EP/S024050/1) and DeepMind. J.A. Noble acknowledges EPSRC grants EP/X040186/1 and EP/T028572/1.

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

**Explaining Explainability: Recommendations for Effective Use of Concept Activation Vectors**

Supplementary Material

## 9 Consistency Proof

Let $\boldsymbol{a}_{l,i}$ be the activation vector in layer $l$ for the input $\boldsymbol{x}_i \in \mathbb{X}$. Function $f$ maps the activations in layer $l_1$ to layer $l_2$, where $l_1 < l_2$, i.e. $f(\boldsymbol{a}_{l_1,i}) = \boldsymbol{a}_{l_2,i}$. We assume that $f$ is continuous and differentiable.

Let $\hat{\boldsymbol{a}}_{l_1}$ and $\hat{\boldsymbol{a}}_{l_2}$ be linearly perturbed activations in each of these layers:

$$\hat{\boldsymbol{a}}_{l_1} = \boldsymbol{a}_{l_1,i} + \boldsymbol{u} \tag{17}$$

$$\hat{\boldsymbol{a}}_{l_2} = \boldsymbol{a}_{l_2,i} + \boldsymbol{v} = f(\boldsymbol{a}_{l_1,i}) + \boldsymbol{v}, \tag{18}$$

where $\boldsymbol{u} \in \mathbb{R}^{m_{l_1}}$ and $\boldsymbol{v} \in \mathbb{R}^{m_{l_2}}$ are vectors of non-zero norm (since CAVs are directions in activation space) and $m_{l_1}$ and $m_{l_2}$ are the dimensions of layer $l_1$ and $l_2$, respectively.

For the two perturbations to have the same effect on the activations (and hence the model) it must hold that:

$$f(\hat{\boldsymbol{a}}_{l_1}) = \hat{\boldsymbol{a}}_{l_2}$$
$$f(\boldsymbol{a}_{l_1,i} + \boldsymbol{u}) = f(\boldsymbol{a}_{l_1,i}) + \boldsymbol{v} \tag{19}$$

where we have substituted in Eq. 17 and Eq. 18. For $\boldsymbol{u}$ and $\boldsymbol{v}$ to be consistent for all possible activations they must be constant with respect to $\boldsymbol{a}_{l_1,i}$, i.e. we can assume that $\boldsymbol{u}$ and $\boldsymbol{v}$ are not functions of $\boldsymbol{a}_{l_1,i}$. If we rearrange Eq. 19 to obtain an equation for $\boldsymbol{v}$, we obtain

$$\boldsymbol{v} = f(\boldsymbol{a} + \boldsymbol{u}) - f(\boldsymbol{a}). \tag{20}$$

where we have simplified the notation by writing $\boldsymbol{a}_{l_1,i}$ as $\boldsymbol{a}$. Let us differentiate Eq. 20

$$\frac{d}{d\boldsymbol{a}}\boldsymbol{v} = \frac{d}{d\boldsymbol{a}}(f(\boldsymbol{a} + \boldsymbol{u}) - f(\boldsymbol{a})) \tag{21}$$

$$\boldsymbol{0} = \frac{d}{d\boldsymbol{a}}f(\boldsymbol{a} + \boldsymbol{u}) - \frac{d}{d\boldsymbol{a}}f(\boldsymbol{a}) \tag{22}$$

$$\boldsymbol{0} = f'(\boldsymbol{a} + \boldsymbol{u}) - f'(\boldsymbol{a}) \tag{23}$$

$$f'(\boldsymbol{a} + \boldsymbol{u}) = f'(\boldsymbol{a}) \tag{24}$$

which implies that the derivative of $f$, $f'$, is periodic with period $\boldsymbol{u}$. This is a strong restriction on the form that $f'$ can take. Let's integrate to find out what implications it has on the form of $f$. First, let's split the periodic function $f'$ into its mean value and its oscillatory part:

$$f'(\boldsymbol{a}) = g'(\boldsymbol{a}) + M \tag{25}$$

where $M \in \mathbb{R}^{m_{l_2} \times m_{l_1}}$ is the mean value across one interval for each component of $f'(\boldsymbol{a})$ and $g'(\boldsymbol{a})$ is a periodic function with zero integral across a single period, i.e.

$$\int_{\boldsymbol{0}}^{\boldsymbol{u}} g'(\boldsymbol{a})d\boldsymbol{a} = \boldsymbol{0}. \tag{26}$$

If we integrate $g'(\boldsymbol{a})$ from $\boldsymbol{a}$ to $\boldsymbol{a} + \boldsymbol{u}$ (using a change of variables with $\boldsymbol{t} \in \mathbb{R}^{m_{l_1}}$) we find

$$\int_{\boldsymbol{a}}^{\boldsymbol{a+u}} g'(\boldsymbol{t})d\boldsymbol{t} = g(\boldsymbol{a}+\boldsymbol{u}) - g(\boldsymbol{a}) \tag{27}$$

and by using Eq. 26 we find

$$g(\boldsymbol{a}+\boldsymbol{u}) - g(\boldsymbol{a}) = 0. \tag{28}$$

Therefore $g$ is periodic with period $\boldsymbol{u}$. If we now take the integral of $f'(\boldsymbol{a})$, we find

$$f(\boldsymbol{a}) = \int f'(\boldsymbol{a})d\boldsymbol{a} \tag{29}$$

$$f(\boldsymbol{a}) = \int g'(\boldsymbol{a})d\boldsymbol{a} + \int M d\boldsymbol{a} \tag{30}$$

$$f(\boldsymbol{a}) = g(\boldsymbol{a}) + M\boldsymbol{a} + \boldsymbol{b}. \tag{31}$$

Hence, $f$ satisfies Eq. 24 (and therefore Eq. 20) if and only if it is composed of a periodic function with period $\boldsymbol{u}$ and a linear term. However, there are further restrictions upon $f$. Let $M = \boldsymbol{0}$ so that $f$ is simply a periodic function. In this case

$$\boldsymbol{v} = f(\boldsymbol{a}+\boldsymbol{u}) - f(\boldsymbol{a}) \tag{32}$$

$$\boldsymbol{v} = f(\boldsymbol{a}) - f(\boldsymbol{a}) \tag{33}$$

$$\boldsymbol{v} = \boldsymbol{0} \tag{34}$$

which contradicts our non-zero assumption on the norm of $\boldsymbol{v}$. Hence we can obtain layer consistent vectors $\boldsymbol{v}$ and $\boldsymbol{u}$ if and only if $f$ is composed of a periodic function with period $\boldsymbol{u}$ and a non-zero linear term $M$. We provide no proof as to whether this form of $f$ can occur in practice in a neural network, however our empirical results in the main paper and § 12 suggest that it does not. Our proof holds generally, however, in the next three sections, we go into more detail for a linear function as it is a special case of Eq. 31 and for the ReLU and sigmoid functions as they are common activation functions used in neural networks.

### 9.1 Special Case: Linear Function

One counter-example that at first look seems to contradict Eq. 31 is a linear function. If $f$ is a linear function, i.e. it conserves vector addition, then Eq. 20 trivially holds:

$$\boldsymbol{v} = f(\boldsymbol{a}+\boldsymbol{u}) - f(\boldsymbol{a}) \tag{35}$$

$$\boldsymbol{v} = f(\boldsymbol{a}) + f(\boldsymbol{u}) - f(\boldsymbol{a}) \tag{36}$$

$$\boldsymbol{v} = f(\boldsymbol{u}). \tag{37}$$

The general result (Eq. 31) requires that $f$ be a combination of a periodic function, $g(\boldsymbol{a})$, and a linear function, $M\boldsymbol{a} + \boldsymbol{b}$, but we have just shown that $f = M\boldsymbol{a} + \boldsymbol{b}$ would also hold. This is because a function that outputs some constant value is a special case of a periodic function, where there is no minimal period as all periods are valid. So, for the case where $f$ is linear $g(\boldsymbol{a}) = \boldsymbol{c}$, where $\boldsymbol{c} \in R^{m_{l_2}}$. Or for a more specific example, in the case of $f = M\boldsymbol{a} + \boldsymbol{b}$, $\boldsymbol{c} = \boldsymbol{0}$. Hence, the result in Eq. 31 generally holds and, for the case where $f$ is linear, the only layer consistent vector $\boldsymbol{v}$ in layer $l_2$ for some vector $\boldsymbol{u}$ in layer $l_1$ is that same vector projected into layer $l_2$ by $f$, i.e. $f(\boldsymbol{u})$.

### 9.2 Example: ReLU Function

In a neural network, $f$ often involves a rectified linear unit (ReLU), so below we find $\boldsymbol{v}$ when $f = \text{ReLU}$. Let $a_{l_1,i,j}$, $v_j$ and $u_j$ refer to the individual elements of $\boldsymbol{a}_{l_1,i}$, $\boldsymbol{v}$ and $\boldsymbol{u}$, respectively. By the definition of a ReLU activation:

$$f(a_{l_1,i,j}) = \max(0, a_{l_1,i,j}) = \begin{cases} a_{l_1,i,j} & a_{l_1,i,j} > 0 \\ 0 & a_{l_1,i,j} \leq 0 \end{cases} \tag{38}$$

So, Eq. 20 becomes:

$$
\begin{aligned}
v_j &= \max(0, a_{l_1,i,j} + u_j) - \max(0, a_{l_1,i,j}) \\
&= \begin{cases} a_{l_1,i,j} + u_j - a_{l_1,i,j} & a_{l_1,i,j} + u_j > 0, a_{l_1,i,j} > 0 \\ a_{l_1,i,j} + u_j + 0 & a_{l_1,i,j} + u_j > 0, a_{l_1,i,j} \leq 0 \\ 0 - a_{l_1,i,j} & a_{l_1,i,j} + u_j \leq 0, a_{l_1,i,j} > 0 \\ 0 - 0 & a_{l_1,i,j} + u_j \leq 0, a_{l_1,i,j} \leq 0 \end{cases} \\
&= \begin{cases} u_j & a_{l_1,i,j} + u_j > 0, a_{l_1,i,j} > 0 \\ a_{l_1,i,j} + u_j & a_{l_1,i,j} + u_j > 0, a_{l_1,i,j} \leq 0 \\ a_{l_1,i,j} & a_{l_1,i,j} + u_j \leq 0, a_{l_1,i,j} > 0 \\ 0 & a_{l_1,i,j} + u_j \leq 0, a_{l_1,i,j} \leq 0. \end{cases}
\end{aligned}
\tag{39}
$$

If $a_{l_1,i,j} + u_j > 0, a_{l_1,i,j} \leq 0$ or $a_{l_1,i,j} + u_j \leq 0, a_{l_1,i,j} > 0$ for any element $j$ then there does not exist a $\boldsymbol{v}$ such that Eq. 19 is true for all $i$, i.e., when either of these statements are true, you cannot have two vectors which have the same effect on the activations across layers for all possible inputs. And if we assume that the elements of $\boldsymbol{a}$ can take any value in practice then there exists no two layer consistent vectors across a ReLU function.

### 9.3 Example: Sigmoid Function

In this section, we consider the sigmoid activation: $f(x) = \frac{1}{1+\exp(-x)}$. For ease of notation, we drop $i$ and $j$ as they do not change, but $a_{l_1}$ and $a_{l_2}$ refer to $a_{l_1,i,j}$ and $a_{l_2,i,j}$, respectively. From Eq. 19, the concept vectors are consistent iff

$$f(a_{l_1} + u) = f(a_{l_1}) + v \tag{40}$$

$$\frac{1}{1+\exp(-a_{l_1} - u)} = \frac{1}{1+\exp(-a_{l_1})} + v \tag{41}$$

Simplifying Eq. 41 for $v$, we get:

$$
\begin{aligned}
v &= \frac{1}{1+\exp(-a_{l_1} - u)} - \frac{1}{1+\exp(-a_{l_1})} \\
v &= \frac{(1+\exp(-a_{l_1})) - (1+\exp(-a_{l_1} - u))}{(1+\exp(-a_{l_1}))(1+\exp(-a_{l_1} - u))} \\
v &= \frac{\exp(-a_{l_1}) - \exp(-a_{l_1} - u)}{(1+\exp(-a_{l_1}))(1+\exp(-a_{l_1} - u))} \\
v &= \frac{1 - \exp(-u)}{(\exp(a_{l_1}) + 1)(\exp(a_{l_1}) + \exp(-u))}
\end{aligned}
$$

This can be simplified further with partial fractions:

$$v = \frac{1 - \exp(-u)}{(\exp(a_{l_1}) + 1)(\exp(a_{l_1}) + \exp(-u))} \tag{42}$$

$$= \frac{\exp(-a_{l_1})}{(\exp(a_{l_1}) + 1)} - \frac{\exp(-a_{l_1} - u)}{(\exp(a_{l_1}) + \exp(-u))} \tag{43}$$

$$= \frac{\exp(-a_{l_1})}{(\exp(a_{l_1}) + 1)} - \frac{\exp(-a_{l_1})}{(\exp(a_{l_1} + u) + \exp(-u))} \tag{44}$$

For a single $v$ to exist which is consistent for all $a_{l_1}$ it cannot depend on $a_{l_1}$. Since the left half of Eq. 44 depends on $a_{l_1}$, the only way that $v$ does not depend on $a_{l_1}$ is if the right hand side cancels out the left. This only occurs when $\boldsymbol{u} = \boldsymbol{v} = \boldsymbol{0}$. Since $\boldsymbol{u}$ and $\boldsymbol{v}$ are directions in activation space, and hence have a non-zero norm, this is a contradiction. Therefore, for the sigmoid function, under no conditions does there exist layer consistent vectors.

## 10 Implementation Details

In this section, we provide general implementation details applicable to the whole paper. For details relating to individual experiments and additional results, see Sections 12, 13 and 14.

### 10.1 Concept Activation Vectors

**Background**  In (Kim et al., 2018), a statistical test, TCAV, determines whether the model's sensitivity to a concept is significant. The test compares a set of CAV scores found using a concept dataset with CAV scores found using random data. To do this, we must find multiple CAVs for each concept. In practice, each of these CAVs is trained with the same positive set, $\mathbb{X}_c^+$, but a different random set, $\mathbb{X}_c^{r-}$, where $r \in 1, 2 \ldots R$ denotes the random index. A CAV corresponding to a specific random index is labelled $\boldsymbol{v}_{c,l}^r$.

**Implementation Details**  In this work, we create multiple CAVs per training run (30 unless otherwise stated), each using the same positive probe dataset but a different random set. We label a CAV trained with a specific random set as $v_{c,l}^r$, where $r \in 1, 2 \ldots R$ denotes the random index. Random CAVs are generated from pairwise combinations of random data sets, and we conduct a two-sided Welch's t-test to test whether the means of concept and random TCAV scores are equal. If a set of CAVs passes this test with a $p$ value less than $0.01^5$, we consider the concept meaningful. We refer to the mean TCAV score of the random CAVs as the null; it acts as the TCAV score all other CAVs should be compared against to understand their sensitivity to the model. The null is often very close to 0.5, simplifying the interpretation of the TCAV score to the concept having positive sensitivity when greater than 0.5 and negative when less.

### 10.2 Elements

**Classification Model**  The model architecture is a simple convolutional neural network with six layers: each layer contains a convolution, batch norm and ReLU, followed by an average pooling and fully connected layer to give the logit outputs. The first three convolutional layers utilise a max-pooling operation to reduce dimensionality. We train the model using Adam (Kingma & Ba, 2015) with a learning rate of 1e-3 until the training accuracy is greater than 99.99%, giving a validation accuracy of 99.98% for the standard dataset. We use a different number of channels for the models trained on different datasets. This allows us to provide more model capacity when needed. The number of channels per layer for each model/dataset is summarised in Table 1.

The models for datasets $\mathbb{E}_2$ and $\mathbb{E}_3$ in section 6.2 are the same architecture as for the simple dataset ($\mathbb{E}_1$).

---

[5]we use a threshold of 0.01 to help reduce the false discovery rate

Table 1: The number of each channels for the models trained on the simple, standard and spatial versions of the Elements dataset.

| Layer | Model | | |
|---|---|---|---|
| | Simple | Standard | Spatial |
| layers.0 | 64 | 64 | 64 |
| layers.1 | 64 | 64 | 64 |
| layers.2 | 64 | 64 | 128 |
| layers.3 | 64 | 128 | 256 |
| layers.4 | 64 | 128 | 256 |
| layers.5 | 64 | 128 | 256 |

**Probe Dataset**  For Elements, the probe datasets are generated so that the positive examples for a concept contain only objects with that concept, so, for example, a red concept image will contain four objects with random shapes and textures that occur within the dataset, but all of them will be red. The negative set consists of random samples from the dataset.

## 10.3  ImageNet

ImageNet is used to demonstrate the experiments on a real-world application.

**Classification Model**  We use the default weights for a ResNet-50 (He et al., 2016) in the TorchVision package in PyTorch, which used a variety of data augmentation techniques including Mixup (Zhang et al., 2018), Cutmix (Yun et al., 2019), TrivialAugment (Müller & Hutter, 2021), and Batch Augmentation (Hoffer et al., 2020).

**Probe Dataset**  Most probe datasets used to train CAVs were collated from the Broden dataset (Bau et al., 2017), particularly focusing on textures such as `striped`, `meshed` or `dotted`, or objects such as `car`, `sea` or `person`. Some concepts were manually curated, such as the `anemone` concept, which was collected from test images of the 'anemone fish' class from ImageNet that were not used elsewhere in the experiments. Examples of some of these concepts are available in Figure 11.

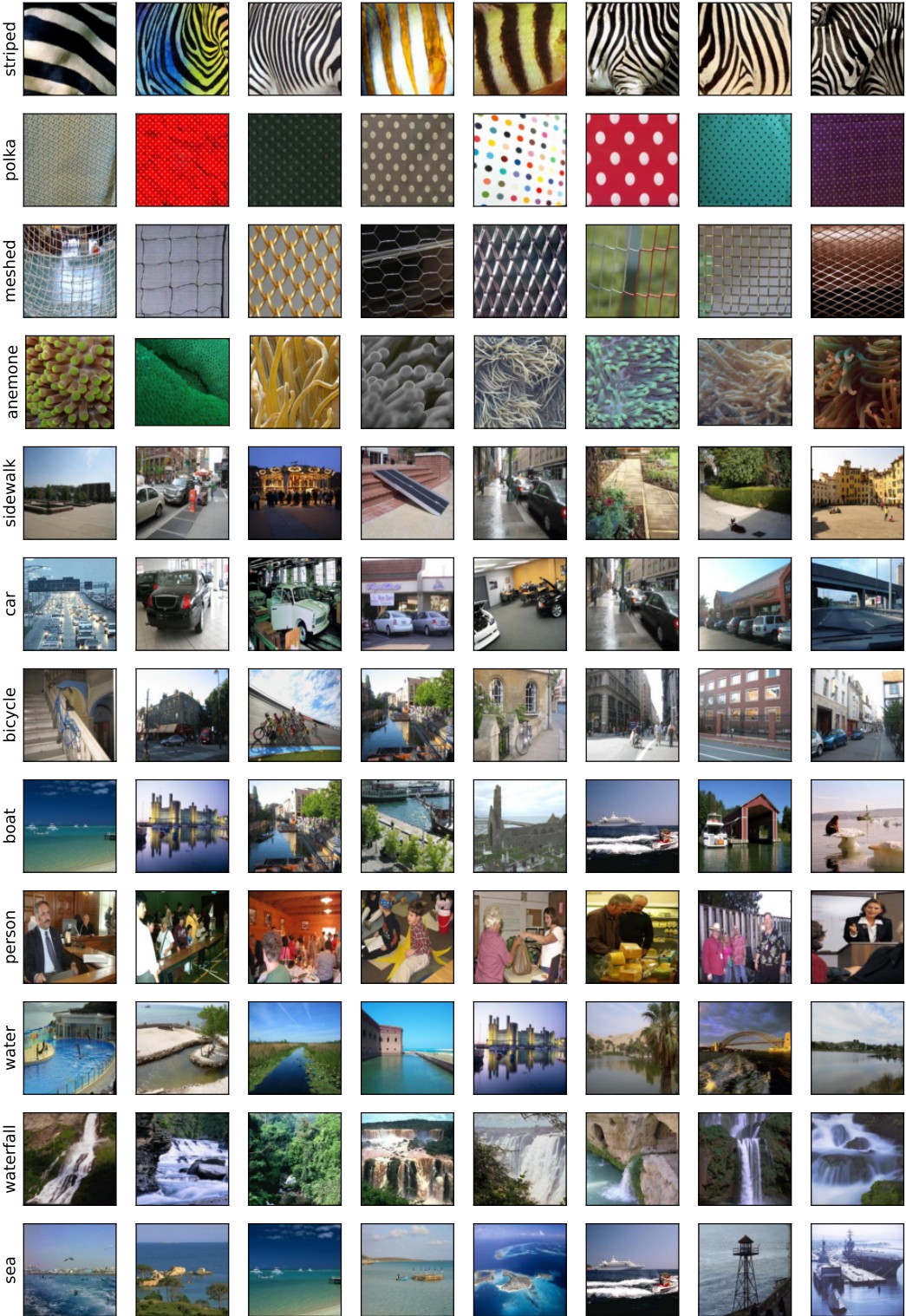

Figure 11: Example positive probe datasets for different concepts for ImageNet.

### 10.4 Layer Selection

When creating CAVs, we need to choose a model layer. For the small CNN used for Elements, we can create CAVs for all layers, but for larger models, such as the ResNet-50 for ImageNet, it is computationally infeasible. In this paper, we focus on layers near the end of the model. The justification for this is twofold. (1) From an information theory perspective, the activations earlier in the network may contain more irrelevant information, suggesting the activations closer to the output may be more relevant to the prediction (Xu et al., 2020; McGrath et al., 2022). We aim to use TCAV to explain the model output, therefore later layers may be more desirable. (2) The model representations may be more complex in later layers. This allows us to create CAVs for more complex concepts. We find that the empirical evidence supports these hypotheses. Figures 12 and 13 show the accuracy of the linear classifiers used to create the CAVs on a held-out test set for each probe dataset. The accuracy for each concept tends to increase in later layers. This suggests the CAVs better capture the model representations in later layers. However, we observe variation across the concepts. For example, the colours in Elements are easily classified in all layers, whereas the shapes/textures have lower performance in layers.1. As our goal is to understand the behaviour of concept vectors (when the concept is represented), we focus on CAVs that obtain at least 90% test accuracy. Therefore, we omit layers.1 in our analysis.

We do not create CAVs for the final convolutional layer in either the simple CNNs for Elements (layers.5) or the ResNet-50 for ImageNet (layer4.2) due to the gradient behaviour in these layers. In both cases, the network has no non-linearities after the layer. Therefore, the gradient of the logit with respect to the activations solely depends on the model weights, not the activations. TCAV relies on having a distribution of directional derivatives, which are then thresholded and averaged over different data points. For these layers, the gradient is the same for all inputs, and hence so is the directional derivative. This means the TCAV score for an individual CAV in these layers will be exactly 1 or 0. As such, we do not perform TCAV on layers after which there are no non-linearities.

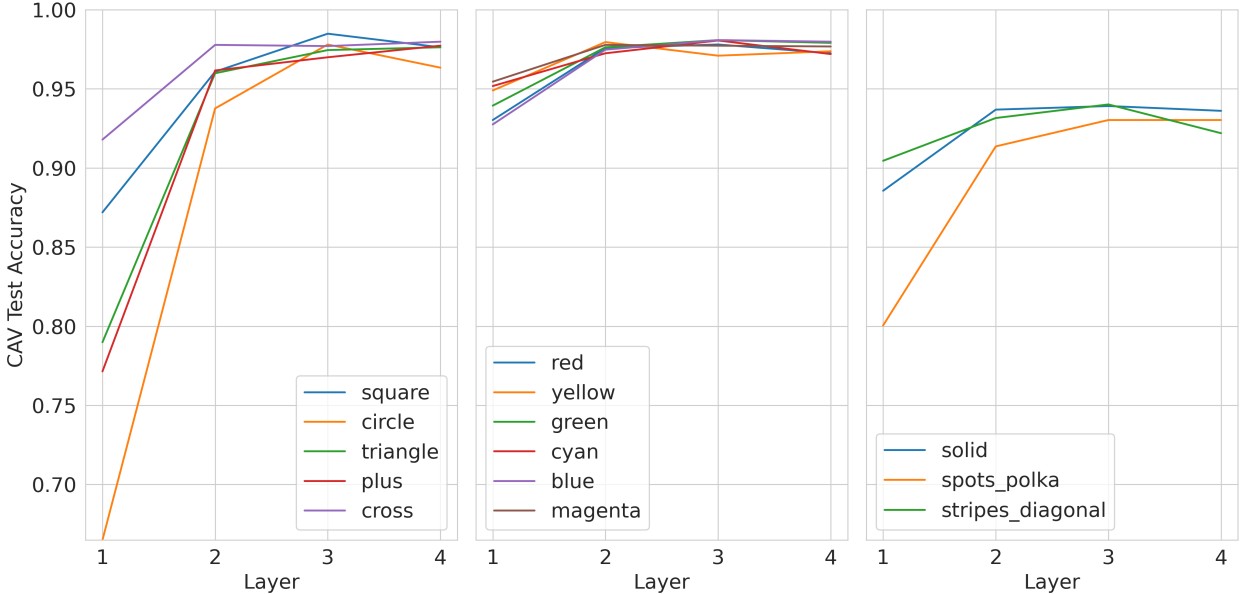

Figure 12: Mean test accuracy for the linear classifiers from which the CAVs are generated for all concepts in the standard Elements dataset (split by concept type).

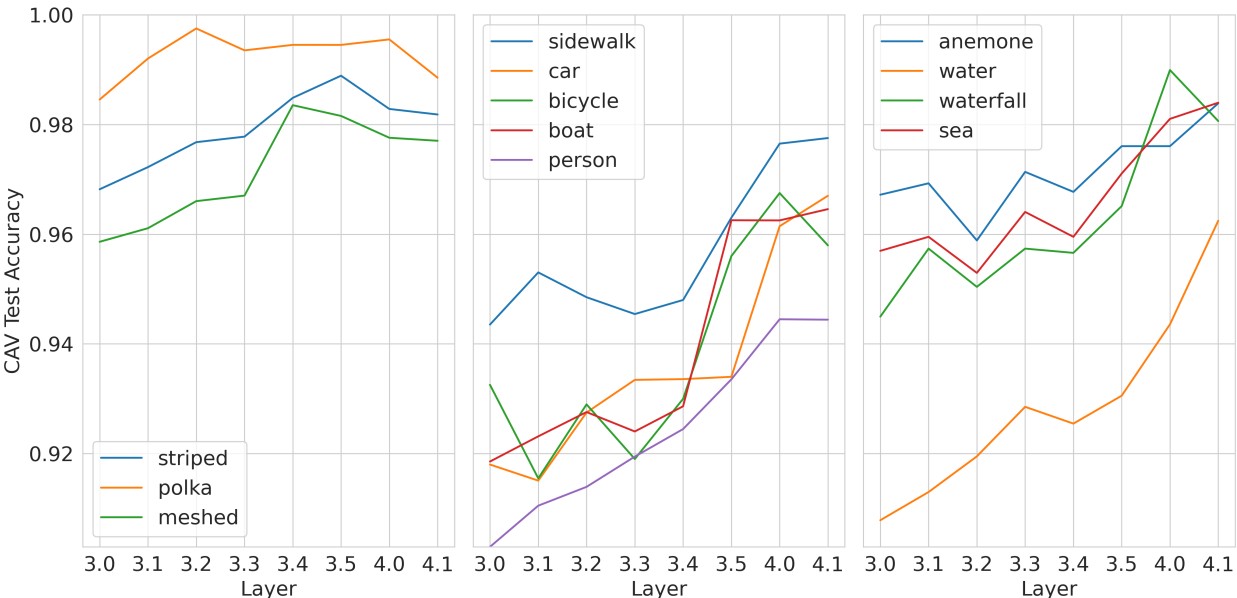

Figure 13: Mean test accuracy for the linear classifiers from which the CAVs are generated for a selection of concepts in ImageNet.

# 11 Elements Dataset

## 11.1 Benefits of the Elements Dataset

**Configurable datasets**  The configurable nature of the dataset allow us to explore different properties of the model and of CAVs. For example, we can introduce an association between the red and striped concepts in the training set by requiring that all red elements are striped; or we can create a probe dataset of red elements on the right of the image to explore CAV spatial dependence.

**Ground truth model behaviour**  The classes are configured as combinations of the elements' shape, colour and texture. Therefore, by construction, we have the ground truth relationship between each concept and class. As these relationships are within the dataset, knowing the ground truth relationship between each concept and class does not allow us to explore model faithfulness. For that, we need the ground truth influence of each concept on model predictions. By having a class for each possible combination concepts, the model must learn linearly separable representations of the concepts in the representation space (before the final linear layer) of the NN to achieve a high accuracy. Therefore, we have the ground-truth relationship of how each concept influences the model's predictions and can explore the faithfulness of concept-based explanation methods.

## 11.2 Elements Configuration

In the Elements dataset, there are various attributes we can vary. These attributes come in two types – image attributes and element attributes. The image attributes are: the number of elements per image and the size of the image. The element attributes are: colour, brightness, size, shape, texture, texture shift, and x and y coordinates within the image. Most are self-explanatory from their name, but texture shift requires more explanation. It is a small change in how the texture is applied to the object so that, for example, spots are not always in the same location with respect to the edge of the object. In this section, we describe the values that each of these attributes can take for the different versions of Elements that are used in the paper.

**Standard**   The default version contains four objects in each image and the allowed concepts are the primary colours and their pairwise combinations (red, green, blue, yellow, cyan, magenta), five shapes (square, circle, triangle, plus, cross) and three textures (solid, spots and diagonal stripes).

**Simple**   In some experiments, when stated, we use a simpler version which contains fewer shapes and colours. This is to reduce the complexity of the figures. The standard and simple dataset configurations are in Table 2.

**Spatially Dependent**   For some experiments in section 6.3 we use a spatially dependent version of the standard Elements dataset. This has the same configuration but introduces spatially dependent classes. As in the standard dataset, there is a class for all combinations of two and three concepts, 'striped squares' or 'spotted cyan triangles'. However, there are additional classes which depend on where the element is present in the image. For all classes involving triangles, we add two new classes which depend on if the object is in the top or bottom half of the image. For example, the class 'blue triangles' will now have two additional classes related to it of 'blue triangles on the top' and 'blue triangles on the bottom'. Similarly, we introduce two new classes for all classes involving squares, but for the left/right halves of the image rather than the top/bottom.

**Entangled**   We use two alternative versions of the simple dataset in the Entanglement experiments: $\mathbb{E}_2$ and $\mathbb{E}_3$. As described in the main text, these are identical to the simple dataset, apart from the association between the red and triangle concepts. In each case, we restrict some of the allowed combinations of concepts that an element can take. This also removes some classes from the dataset which we reflect in any trained models. $\mathbb{E}_2$ does not allow any shape apart from triangles to be red. This removes classes like 'red circles' or 'spotted red squares'. $\mathbb{E}_3$ has this restriction and then places a further restriction that triangles have to red. This removes classes such as 'blue triangles' or 'striped green triangles'.

Table 2: The configurations for the standard and simple versions of the elements dataset. Element size and image size are in pixels. Brightness is the value of the pixels in the element.

| Property | Dataset | |
|---|---|---|
| | Standard | Simple |
| Colours | red, green, blue, yellow, cyan, magenta | red, green, blue |
| Shapes | square, circle, triangle, plus, cross | square, circle, triangle, plus |
| Textures | spots, stripes, solid | spots, stripes, solid |
| Brightness | 153-255 | 153-255 |
| Element Size / pixels | 48-80 | 48-80 |
| No. Elements per image | 4 | 4 |
| Image Size / pixels | 256 | 256 |

## 11.3 Examples

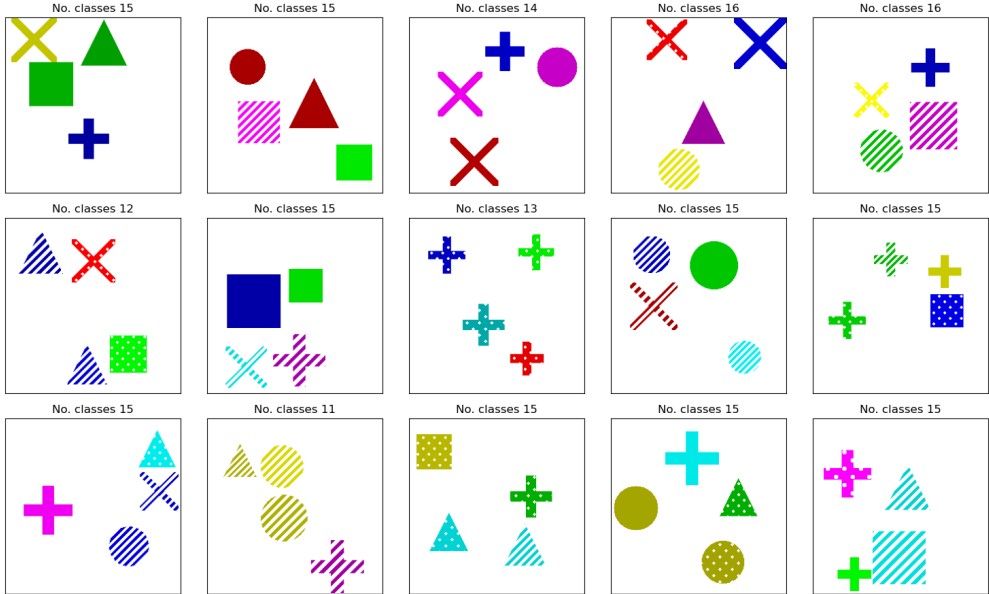

Figure 14: Example images from the standard elements dataset. The number of classes each image belongs to is displayed above it.

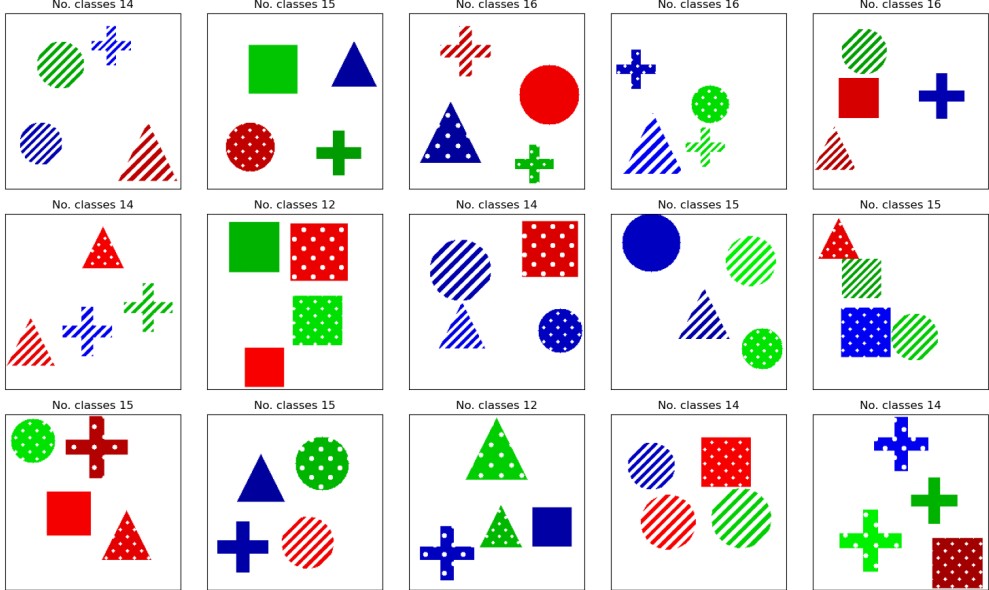

Figure 15: Example images from the simple elements dataset. The number of classes each image belongs to is displayed above it.

## 12 Consistency Experiment Details

Figure 3 shows the consistency error for various types of CAV. In this section, we describe how we find the different types of CAV. In each case, $v_{c,l_1}$ is a CAV trained as normal using a probe dataset, whereas the creation method for $v_{c,l_2}$ varies for each experiment and is described below:

**Optimised CAV** We use gradient descent to optimise $\boldsymbol{v}_{c,l_2}$ to minimise the consistency error:

$$\arg\min_{\boldsymbol{v}_{c,l_2}} ||f(\boldsymbol{a}_{l_1} + \boldsymbol{v}_{c,l_1}) - (\boldsymbol{a}_{l_2} + \boldsymbol{v}_{c,l_2})|| \tag{45}$$

The starting point for each optimisation process is a CAV in layer $l_2$, $\boldsymbol{v}_{c,l_2}^{r_2}$, trained on a different random probe dataset. The nearly identical errors for each optimisation process support the likelihood of a global minimum being reached.

**Concept CAV** These are simply normal CAVs trained as described in Sections 2 and 10, , the CAVs are trained using a probe dataset containing $\mathbb{X}_c^+$ and $\mathbb{X}_c^-$. The distribution is over $r_2$ for different random probe datasets $\mathbb{X}_{c,r_2}$ for $\boldsymbol{v}_{c,l_2}$, where $r_2 \neq r_1$ denotes the random set.

**Projected CAV** CAVs in layer $l_1$ projected into layer $l_2$ using $f$: $f(\boldsymbol{v}_{c,l_1})$. The distribution is over different CAVs in layer $l_1$, $\boldsymbol{v}_{c,l_1}^r$.

**Random CAV** CAVs trained using a random probe dataset for both the positive and negative sets:

$$\begin{aligned} \mathbb{X}_c^- &= \mathbb{X}_{c,r_1}^- \\ \mathbb{X}_c^+ &= \mathbb{X}_{c,r_2}^-, \quad r_2 \neq r_1 \end{aligned} \tag{46}$$

**Random Direction** Each element of $\boldsymbol{v}_{c,l_2}$ is drawn from a uniform distribution between [-0.5, 0.5], and rescaled to be a unit vector. The distribution is over different random seeds for the random number generator.

## 12.1 Scaling perturbations

To ensure that $\boldsymbol{a}_l + \boldsymbol{v}_{c,l}$ stays in-distribution, we scale the perturbations as follows:

$$\hat{\boldsymbol{a}}_l = \boldsymbol{a}_l + \gamma \boldsymbol{v}_{c,l} \frac{\overline{||\boldsymbol{a}_l||}}{||\boldsymbol{v}_{c,l}||}, \tag{47}$$

where $\gamma$ is a hyperparameter used for perturbation size (typically set to 0.01), $||\cdot||$ the $L_2$ norm of a vector, and $\overline{||\boldsymbol{a}_l||}$ the average norm of $\boldsymbol{a}_l$. We scale the perturbation by the mean activation norm to account for the difference in the norms between the activation and concept vector to have consistently sized perturbations across layers.

We performed experiments to explore the sensitivity of consistency error to the size of the perturbation, $\gamma$, for various layers and concepts for the ImageNet and Elements datasets. In Figures 16 and 17, we show the results for a variety of concepts for both the ImageNet and Elements datasets. Similar patterns were observed across experiments. As we increase $|\gamma|$, the consistency error scales linearly, as a larger perturbation causes a larger difference between $f(\hat{\boldsymbol{a}}_{l_1})$ and $\hat{\boldsymbol{a}}_{l_2}$. The scale of the y axis on the left of Figures 16 and 17 is not particularly meaningful without context, so we scale it by the norm of the perturbation in layer $l_2$, $||\gamma \boldsymbol{v}_{c,l_2} \frac{\overline{||\boldsymbol{a}_{l_2}||}}{||\boldsymbol{v}_{c,l_2}||}||$, on the right of the same figures. Values greater than one mean that the difference between $f(\hat{\boldsymbol{a}}_{l_1})$ and $\hat{\boldsymbol{a}}_{l_2}$ are larger than the perturbation made in layer $l_2$.

In addition to scaling $\gamma$ for both perturbations (in layer $l_1$ and $l_2$), we explored fixing the size of the perturbation in layer $l_1$ and varying $\gamma$ in $l_2$. We fixed $\gamma_{l_1}$ to be 0.01 and then varied $\gamma_{l_2}$ from 0 to 0.02 and measured the consistency error of the two perturbations. In Figure 18, we show these results for `striped` CAVs from a ResNet-50 trained on ImageNet. We repeated the experiment on two types of CAVs: (1) standard CAVs (2) optimised CAVs (as described in the previous section). In both cases having $\gamma$ fixed at 0.01 gives a larger error, but the minimum error is not substantially different. We observe a larger difference in error between $\gamma_{l_2} = 0.01$ and the minimum error for the standard CAVs than for the optimised CAVs. This is to be expected, as the directions of the standard CAVs are less aligned than the optimised CAVs.

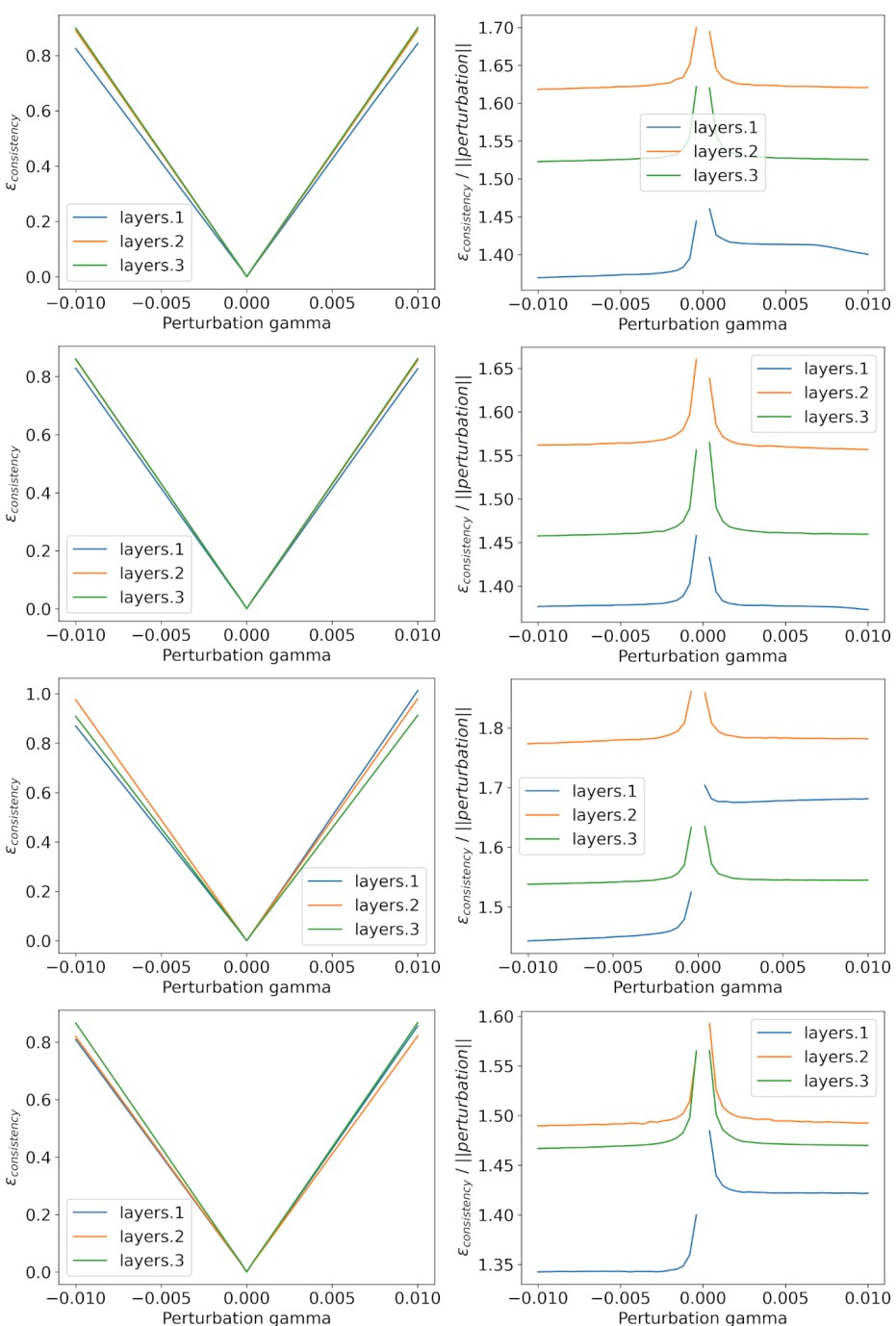

Figure 16: The mean consistency error for (from top to bottom) `red`, `blue`, `triangle` and `striped` CAVs across layers (left) scaled by the size of the perturbation (right) for the Elements dataset.

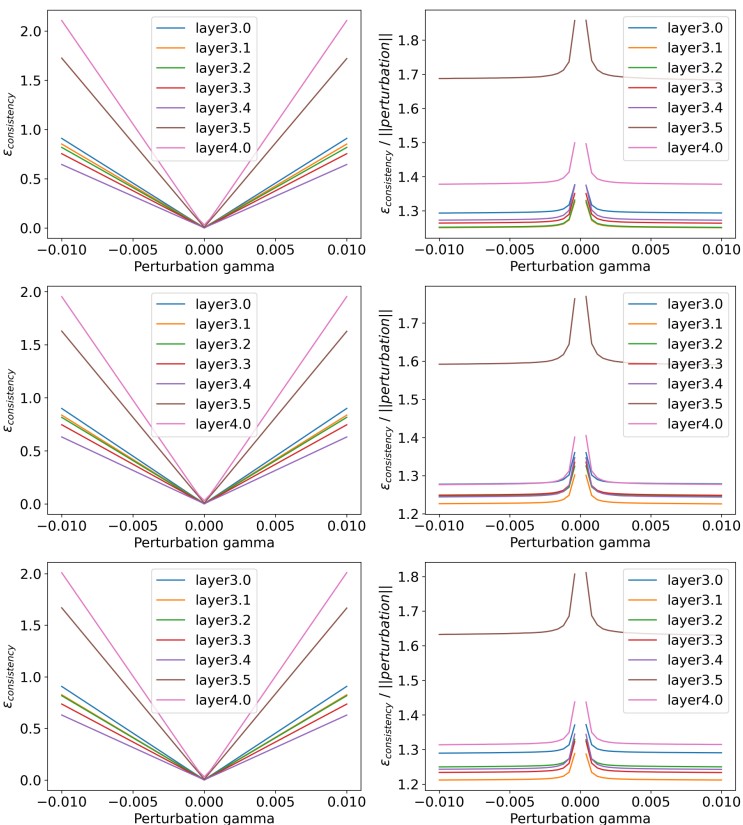

Figure 17: The mean consistency error across 10 CAVs for `striped` (top), `lined` (middle) and `dotted` (bottom) CAVs across layers (left) scaled by the size of the perturbation (right) for a ResNet-50 trained on ImageNet.

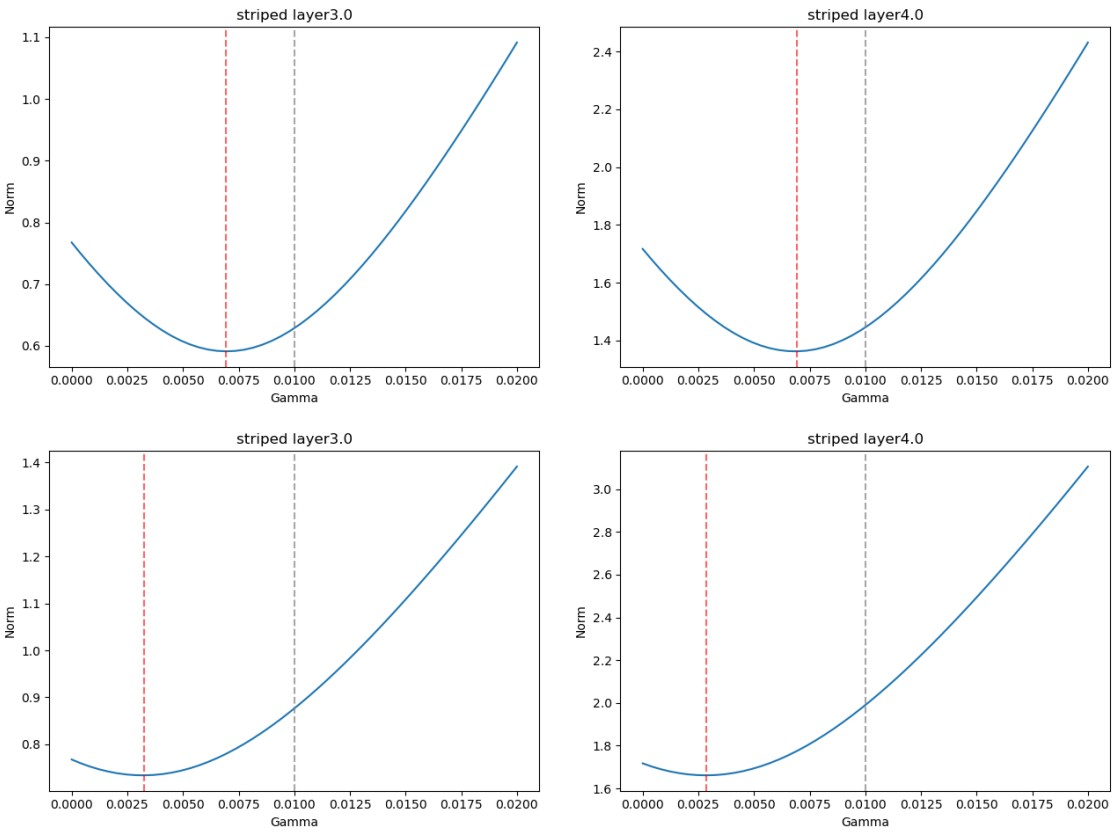

Figure 18: The sensitivity of the mean consistency error to scaling $\gamma_{l_2}$ for optimised CAVs (top) and standard CAVs (bottom) where $l_1$ =layer3.0 (left) and $l_1$ =layer4.0 (right) of a ResNet-50 trained on ImageNet. $\gamma = 0.01$ is the standard perturbation made in our experiments and is marked by a grey dashed line and the $\gamma$ which gives a minimal consistency error is marked by a red dashed line.

## 12.2   Additional results

In this section we provide additional results for the different consistency experiments as in fig. 3. We include the results for multiple concepts and layers for both the ImageNet and Elements datasets. To allow for better comparison between layers, we normalise the consistency errors in some of the figures. For each layer, the consistency error is divided by the mean error for the optimised CAVs. For the normalised plots, a value of one can be seen as the lowest error possible for that layer. The relative ordering of layers is approximately consistent across the different types of CAV. For example, in fig. 19, there is a downward trend between layer3.0 to layer3.5 and then an increase for layer4.0.

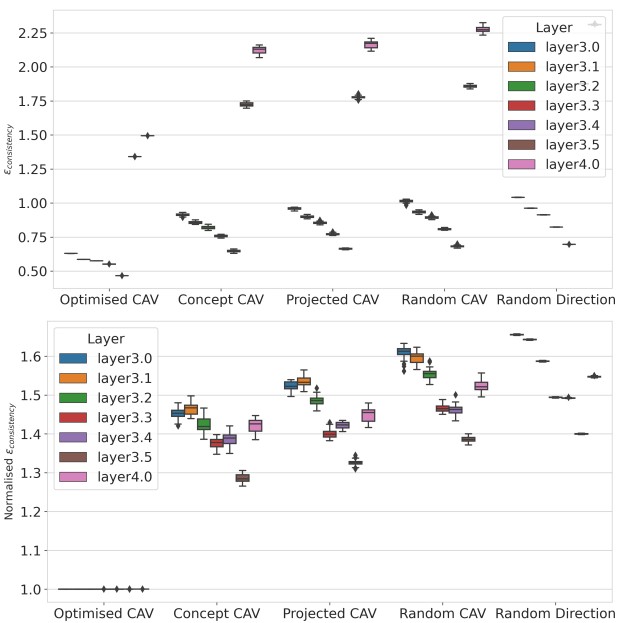

Figure 19: The distribution of consistency errors (top) and normalised consistency errors (bottom) for different $\boldsymbol{v}_{c,l_2}$ for `striped` in a selection of layers from a ResNet-50 trained on ImageNet. Optimised CAV: The lower bound – a vector optimised to have the minimum error. Concept CAV: `striped` CAVs, trained as normal. Projected CAV: striped CAVs from layer $l_1$ projected into layer $l_2$, $f(\boldsymbol{v}_{c,l_1})$. Random CAV: CAVs with random images for the probe dataset. Random Direction: Random vectors drawn from a uniform distribution.

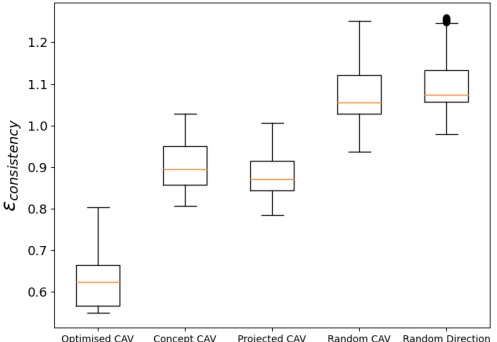

Figure 20: The distribution of consistency errors for different $\boldsymbol{v}_{c,l_2}$ for `square`, `triangle`, `red`, `green`, `solid` and `stripes` CAVs for 'layers.1', 'layers.2' and 'layers.3' of a CNN trained on the Elements dataset. Optimised CAV: The lower bound – a vector optimised to have the minimum error. Concept CAV: CAVs, trained as normal. Projected CAV: striped CAVs from layer $l_1$ projected into layer $l_2$, $f(\boldsymbol{v}_{c,l_1})$. Random CAV: CAVs with random images for the probe dataset. Random Direction: Random vectors drawn from a uniform distribution.

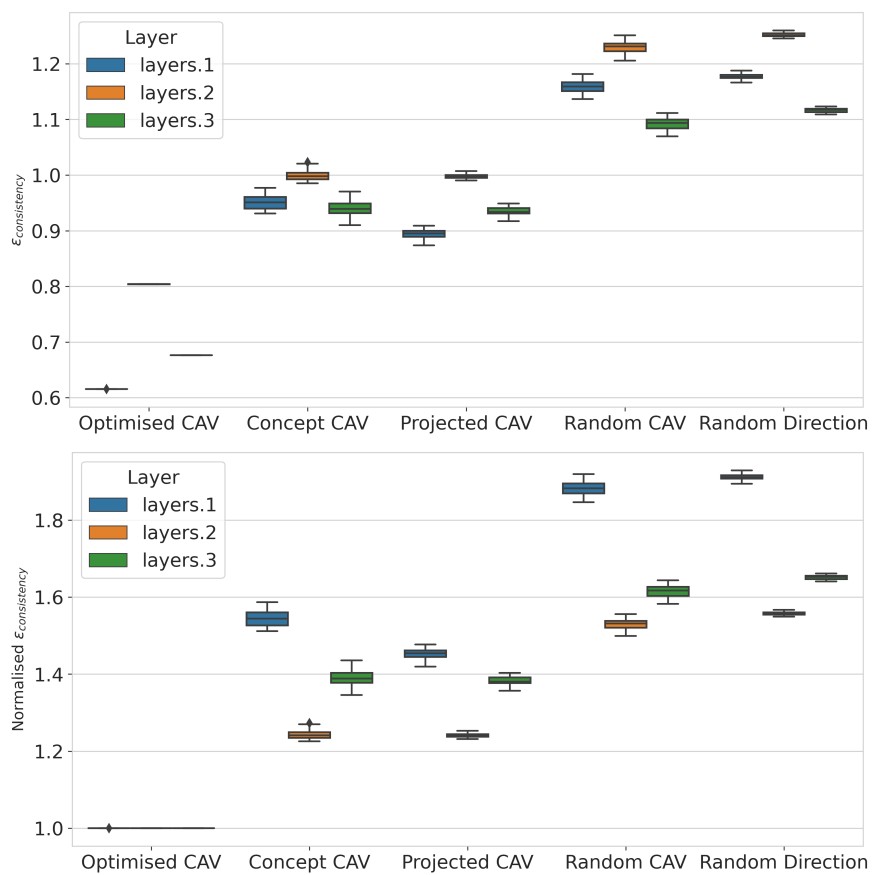

Figure 21: The distribution of consistency errors (top) and normalised consistency errors (bottom) for different $\boldsymbol{v}_{c,l_2}$ for `square` CAVs for a variety of layers for the Elements dataset. Optimised CAV: The lower bound – a vector optimised to have the minimum error. Concept CAV: CAVs, trained as normal. Projected CAV: striped CAVs from layer $l_1$ projected into layer $l_2$, $f(\boldsymbol{v}_{c,l_1})$. Random CAV: CAVs with random images for the probe dataset. Random Direction: Random vectors drawn from a uniform distribution.

## 12.3 DeepDream

DeepDream (Mordvintsev et al., 2015) is a feature visualisation tool. It starts from an image, generated by sampling noise from a random uniform distribution and then iteratively updates the input image to maximise the L2 norm of activations of a particular layer. We use a similar approach but instead maximise the dot product between the activations and a CAV: $\boldsymbol{v}_{c,l} \cdot \boldsymbol{a}_l$. In Figures 22 and 23, we show these visualisations for a selection of ImageNet CAVs for successive layers in a ResNet-50.

These visualisations offer qualitative evidence that the CAVs represent different components of the same concept in different layers. For example, the `car` concept in Figure 22 consists of many square box-like objects in earlier layers, but nothing recognisable as a car, whereas, in later layers, whole car sections can be seen in the visualisations.

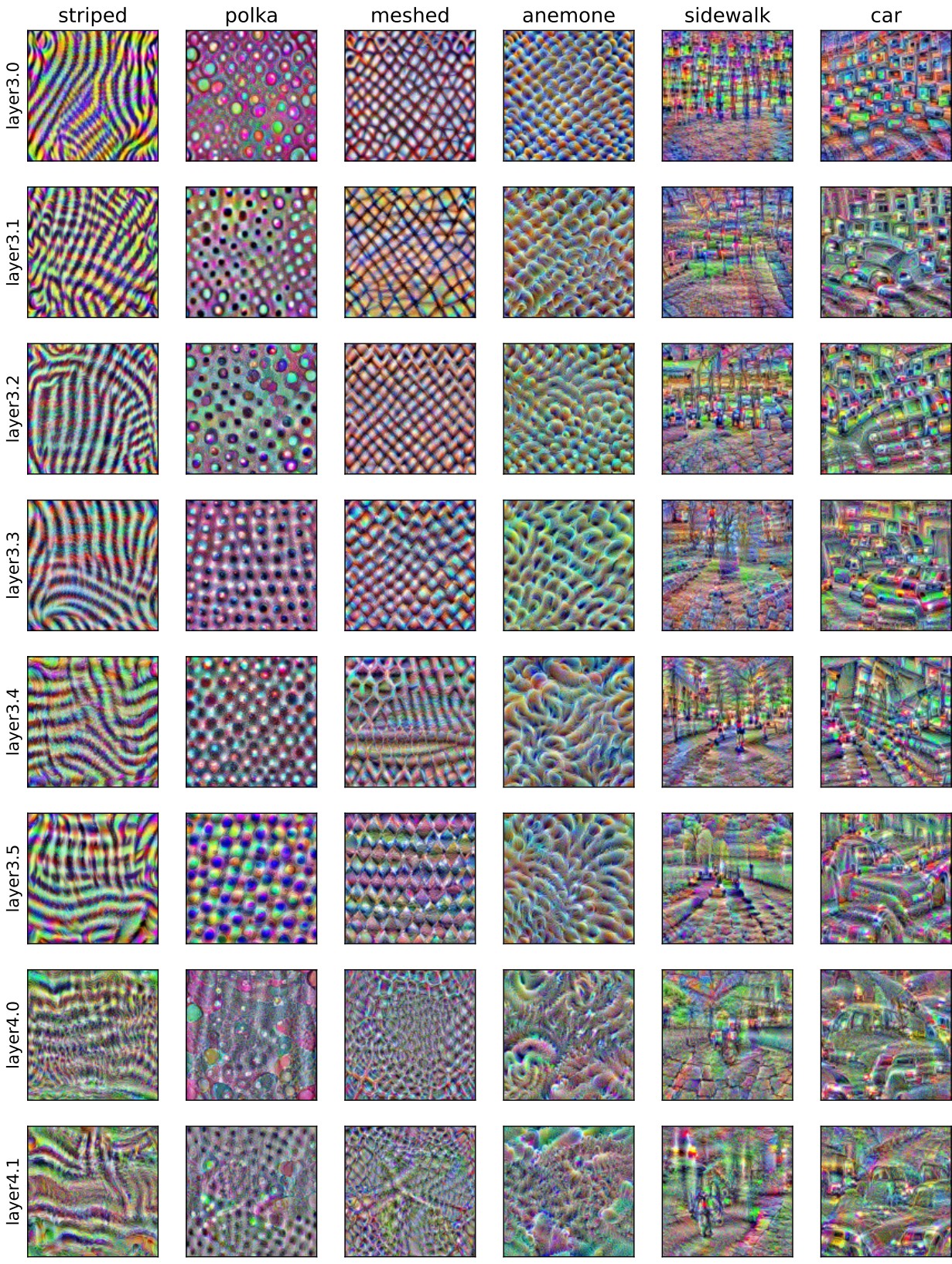

Figure 22: CAV visualisations using DeepDream for a selection of concepts from ImageNet. Each row corresponds to a layer of a ResNet-50 and each column a different concept.

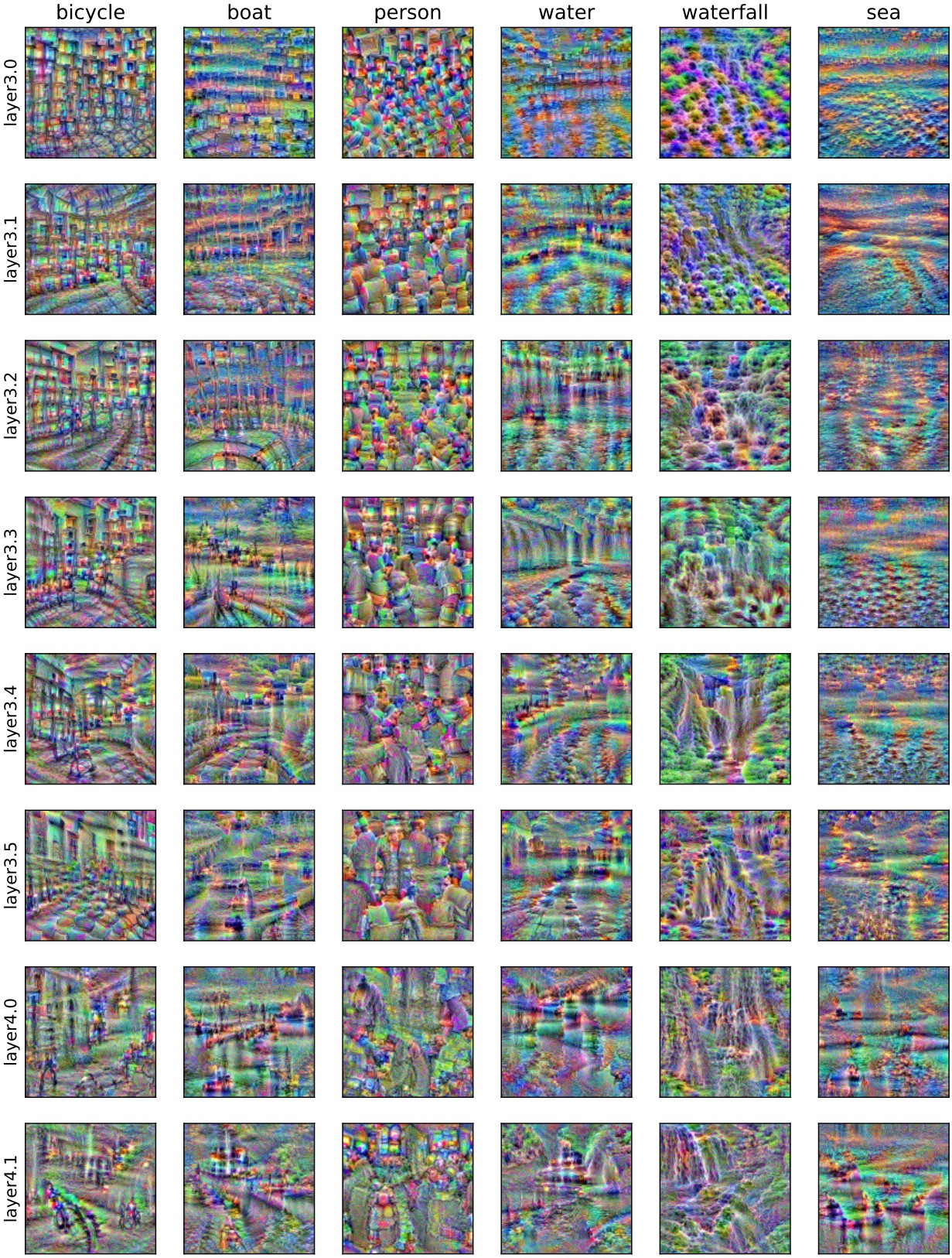

Figure 23: CAV visualisations using DeepDream for a selection of concepts from ImageNet. Each row corresponds to a layer of a ResNet-50 and each column a different concept.

## 12.4 Inconsistent TCAV Scores

In this section, we display additional examples of inconsistent TCAV scores across layers for ImageNet classes. In Figure 24 each subfigure contains at least one concept that has inconsistent TCAV scores. We include example images from those classes in fig. 25.

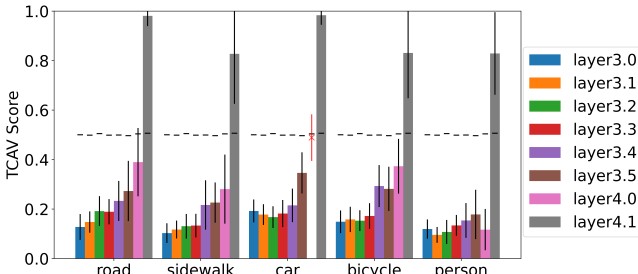

(a) TCAV scores for a selection of concepts for the 'car wheel' class in ImageNet.

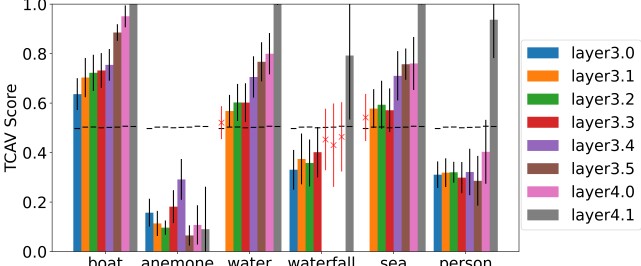

(b) TCAV scores for a selection of concepts for the 'dock' class in ImageNet.

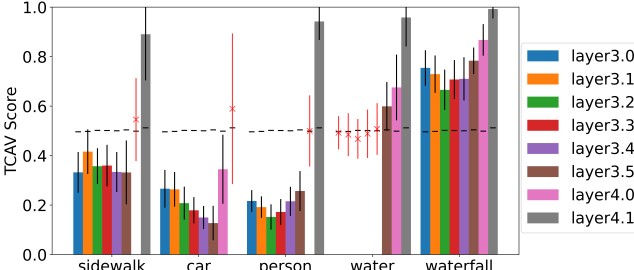

(c) TCAV scores for a selection of concepts for the 'sidewalk' class in ImageNet.

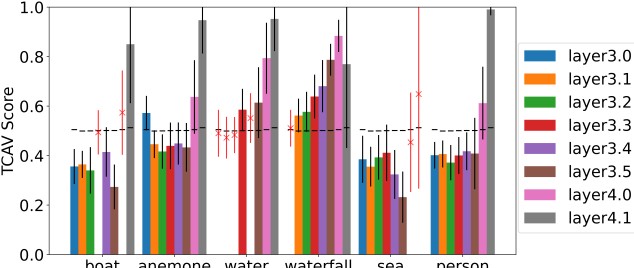

(d) TCAV scores for a selection of concepts for the 'lionfish' class in ImageNet.

Figure 24: Inconsistent TCAV scores for a selection of concepts and classes in ImageNet. The standard deviation is shown in black for significant results and red for insignificant results. The mean TCAV score for random CAVs are shown as horizontal black lines.

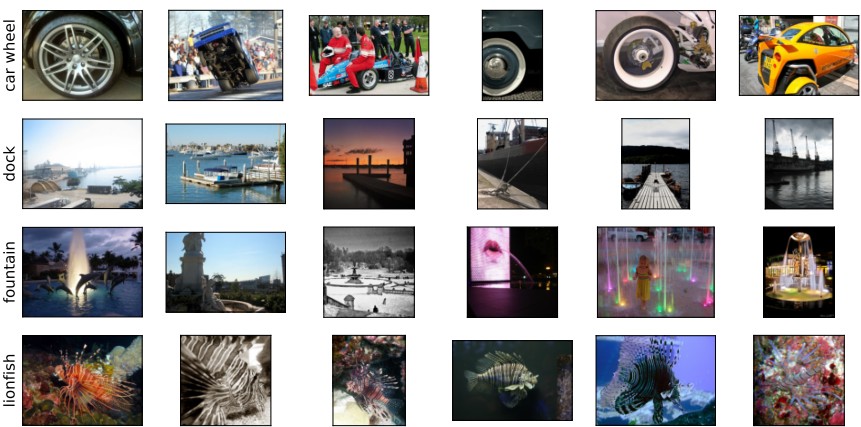

Figure 25: Example images from a selection of ImageNet classes.

## 12.5 TCAV Layer Consistency Score

At the end of § 6.1 we defined the TCAV layer consistency score as a measure of how often significant TCAV scores disagree on the direction of sensitivity for a specific concept and class across a set of layers. Here, we provide some example results for both the ImageNet and Elements datasets. In each case we train 30 CAVs for each concept/layer pair to calculate the TCAV score across. For Elements, when analysing all 69 classes from the simple version of the dataset, we obtain a mean TCAV consistency score of 0.841 across all concepts and all layers of the simple NN. 38/690 sets of TCAV scores had a TCAV layer consistency score of less than 0.2 and 158/690 less than 0.5. For ImageNet, we obtain a mean TCAV consistency score of 0.868 across the last 8 convolutional layers (from layer 3.0 to layer 4.1) for a range of concepts (striped, polka, meshed, road, sidewalk, car, bicycle, boat, person, anemone, water, waterfall, sea, sky, snow, tree) and classes (zebra, leopard, tiger, car wheel, acoustic guitar, academic gown). 7/96 sets of TCAV scores had a TCAV layer consistency score of less than 0.2 and 13/96 had a score less than 0.5. See Figure 26 for the full distributions.

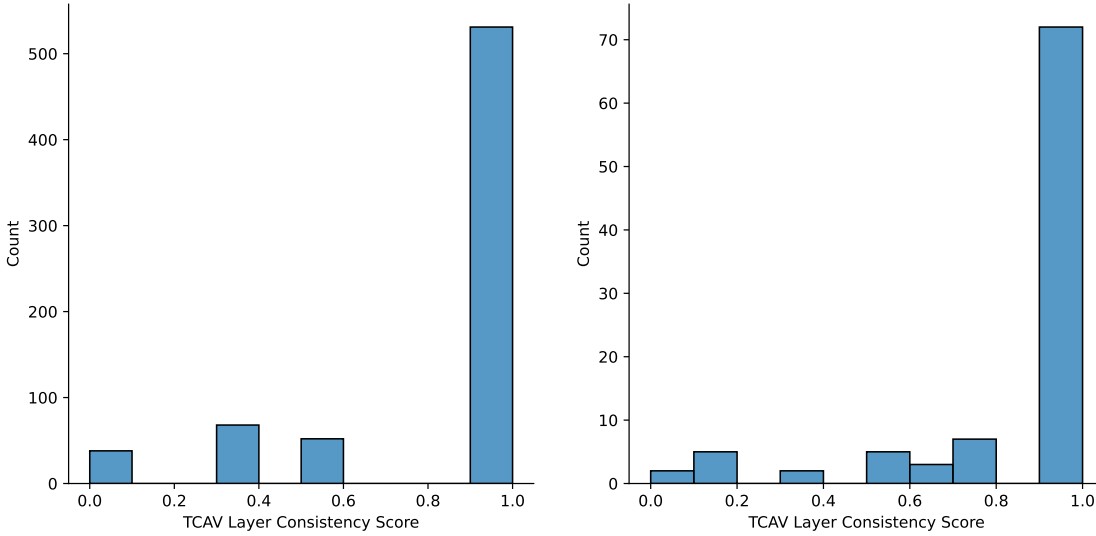

Figure 26: Distribution of TCAV layer consistency score for the simple Elements dataset (left) and a selection of classes/concepts for a ResNet-50 trained on ImageNet (right).

## 13 Entanglement Experiment Details

We use cosine distance to measure how similar two CAVs are. Assuming the CAVs, $v_{c_1,l}$ and $v_{c_2,l}$, are unit vectors this simplifies to the dot product of the two:

$$\text{Cosine Similarity} = \frac{\boldsymbol{v}_{c_1,l} \cdot \boldsymbol{v}_{c_2,l}}{\|\boldsymbol{v}_{c_1,l}\| + \|\boldsymbol{v}_{c_2,l}\|}$$
$$= \boldsymbol{v}_{c_1,l} \cdot \boldsymbol{v}_{c_2,l} \tag{48}$$

In our visualisations (fig. 4 and section 13.1) we compare multiple CAVs for each concept, each with a different random probe dataset, denoted by $r$. This allows us to see how similar CAVs for the same concept are on repeat training runs. Each value in the visualisation is the mean cosine similarity between the concepts on its corresponding x and y axis labels between all CAVs which do not have the same random probe dataset, i.e.:

$$\sum_{r_1}^{R} \sum_{r_2!=r_1}^{R} \frac{\boldsymbol{v}_{c_1,l}^{r_1} \cdot \boldsymbol{v}_{c_2,l}^{r_2}}{R(R-1)} \tag{49}$$

where $\boldsymbol{v}_{c_1,l}^{r_1}$ is the CAV corresponding to concept $c_1$, layer $l$ and random probe dataset $\mathbb{X}_{c,r_1}$.

### 13.1 Additional Results

#### 13.1.1 Elements

In fig. 27, we show the cosine similarities for all concepts in the standard Elements dataset. The conclusions are similar to the visualisation for $\mathbb{E}_1$ in fig. 4, but the negative associations between the mutually exclusive concepts are weaker. We hypothesise that this is because there are more concepts within each group. This is empirically justified as the average cosine similarity of each group approximately corresponds to the number of concepts in the group. If this hypothesis is true, it makes it unlikely that we will find similar groupings in real datasets containing natural images as concepts are rarely partitioned as neatly or in as few possible combinations.

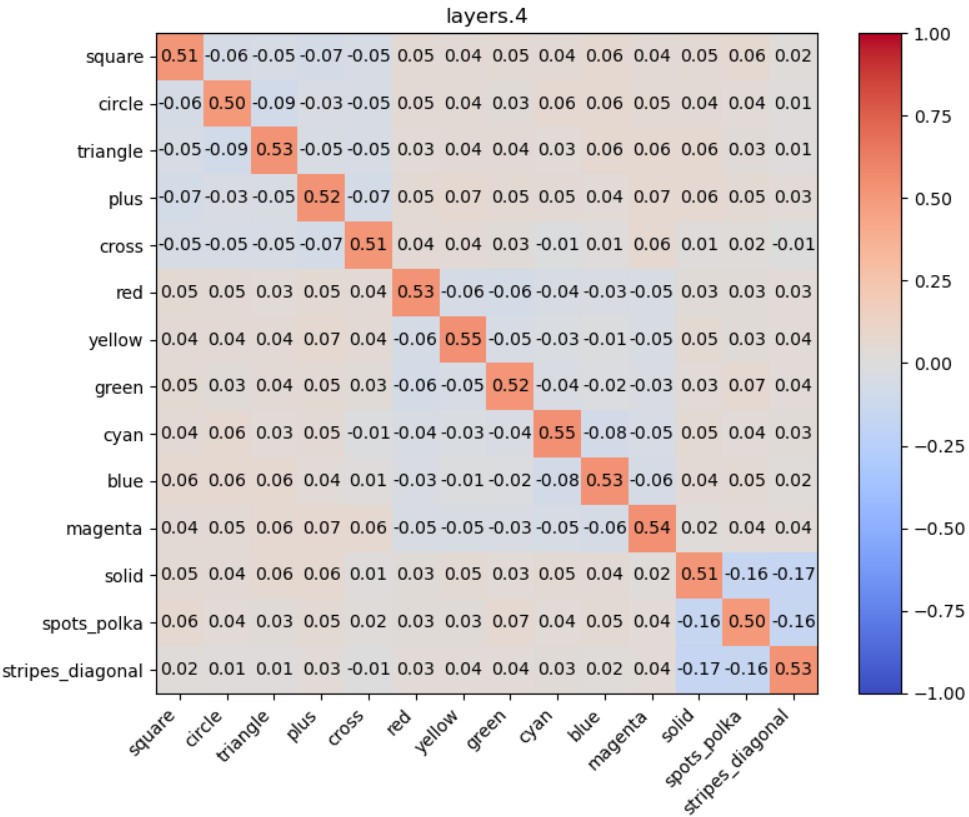

Figure 27: Mean pairwise cosine similarities between 30 CAVs for different concepts from the standard Element dataset.

### 13.1.2  ImageNet

In Figure 28, we show the pairwise cosine similarities for a selection of concepts for a ResNet-50 trained on ImageNet. The associations between concepts are less clear-cut than for Elements, but qualitatively they make intuitive sense. For example, the concepts most similar to `field` are `grass` and `earth`, in the top of Figure 28, and the concepts most similar to `sidewalk` are `bicycle`, `road` and `hedge`. The latter makes sense as many of the `hedge` exemplars in the probe dataset are next to a path or road. This emphasises the importance of designing your probe dataset to match the concept you want the CAV to represent.

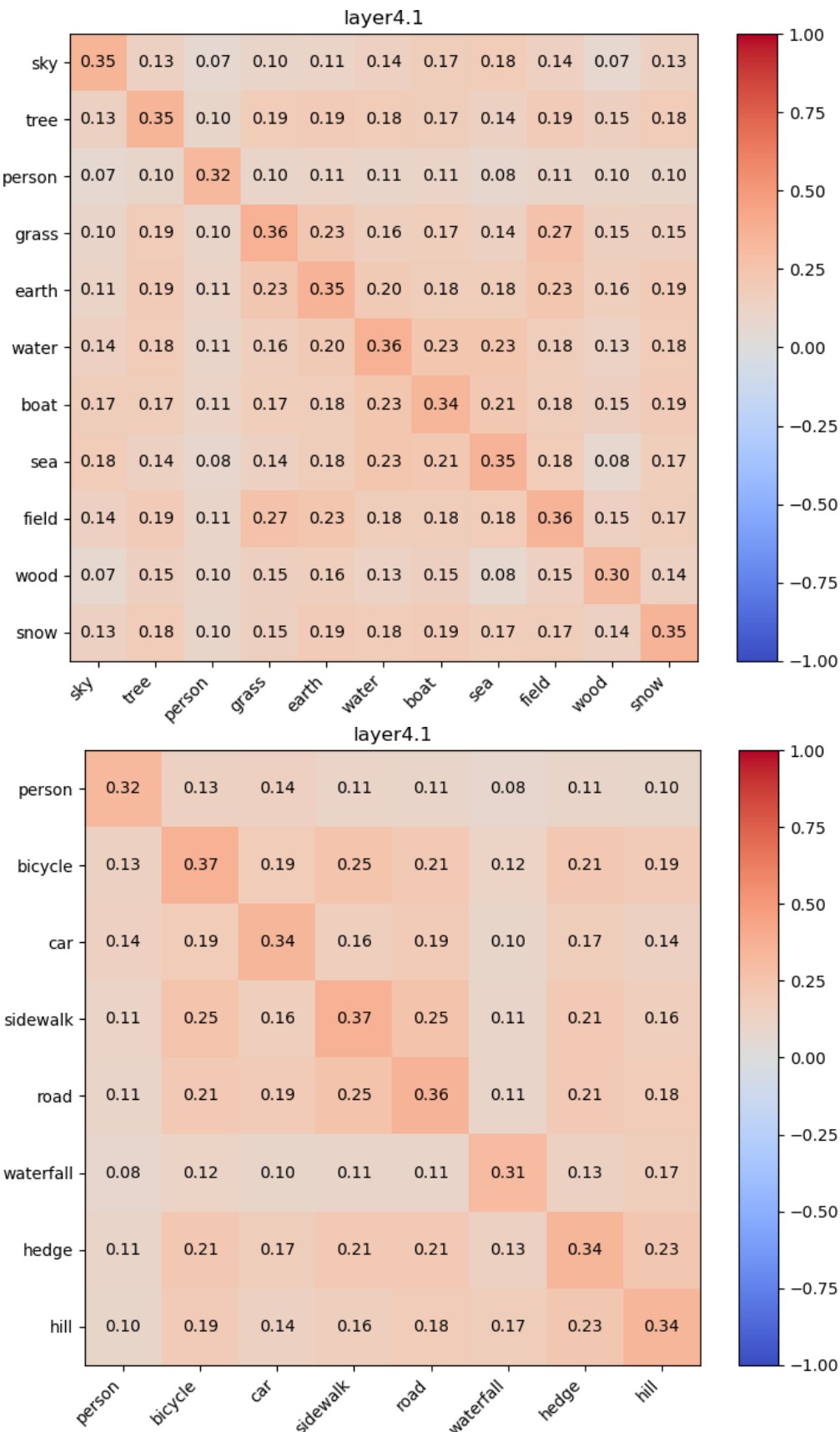

Figure 28: Mean pairwise cosine similarities between 30 CAVs for different concepts from ImageNet.

### 13.2 Polysemanticity

In Figures 4 and 27 we find that the vector representations of mutually exclusive concepts are anti-correlated with each other. Each concept vector does not just mean, for example, `red`. It means `red`, `not green` and `not blue`.

Elhage et al. (2022) also find evidence for polysemantic representations. However, they found individual neurons which were polysemantic, whereas here vectors, i.e., groups of neurons, are polysemantic. In addition, the reasoning is different. The polysemanticity discussed in Elhage et al. (2022) is caused by sparse features being compressed into fewer neurons than there are features. Here, we do not have sparse features and have more neurons than features. Instead, the polysemanticity is caused by associations between the concepts and the optimisation process favouring negatively correlated representations for mutually exclusive concepts.

### 13.3 Dot product distributions

The definition of entangled concepts in eq. (6) uses the dot products of a CAV trained on concept $c_1$ with the the activations of a probe dataset for a different concept $c_2$. Figure fig. 29 shows the distribution of dot products for a selection of CAVs, model training datasets and concept probe datasets. We use the same set of Elements datasets as in section 6.2: $\mathbb{E}_1$, $\mathbb{E}_2$, $\mathbb{E}_3$, where $\mathbb{E}_1$ has no association between `red` and `triangle` and $\mathbb{E}_2$ and $\mathbb{E}_3$ have successively stronger associations between the concepts. This changing association is apparent in the dot product distributions. For $\mathbb{E}_1$, the dot products of the test probe datasets differing from the CAV concept do not differ significantly from the dot products for random images (the negative probe dataset). Whereas, for $\mathbb{E}_2$ and $\mathbb{E}_3$, the random distribution is shifted lower than for either CAV, even if the CAV is labelled differently from the test dataset. This shows that the `red` and `triangle` CAVs are entangled for these datasets/models.

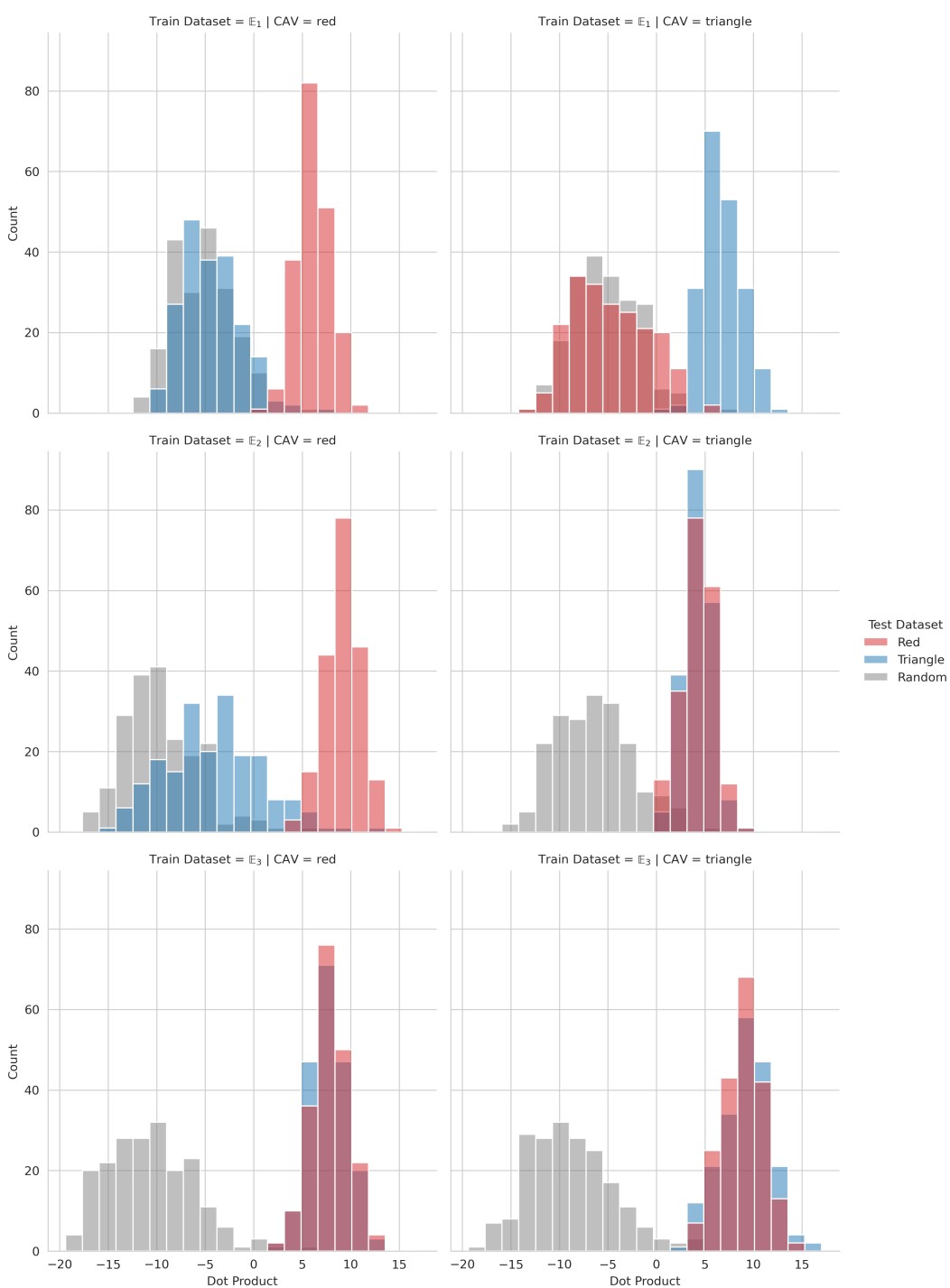

Figure 29: Distribution of dot products $(\boldsymbol{v}_{c_1,l} \cdot \boldsymbol{a}_{c_2,l})$ for the three versions of Elements with increasing association between `red` and `triangle` ($\mathbb{E}_1$, $\mathbb{E}_2$ and $\mathbb{E}_3$). The distribution of dot products are displayed for three different test sets containing in-distribution images for `red`, `triangle` and `random` images.

# 14 Spatial Dependency Experiment Details

## 14.1 Spatially Dependent Probe Datasets

For Elements, the probe datasets contained elements that only appear in specified locations – an example is shown in fig. 2. For ImageNet, we do not have direct control of where objects can appear. Therefore, we greyed out different regions of the image. For example, we created oppositely dependent concepts by either greying out the middle of the image, or the edges - see Appendix 14.1 for examples.

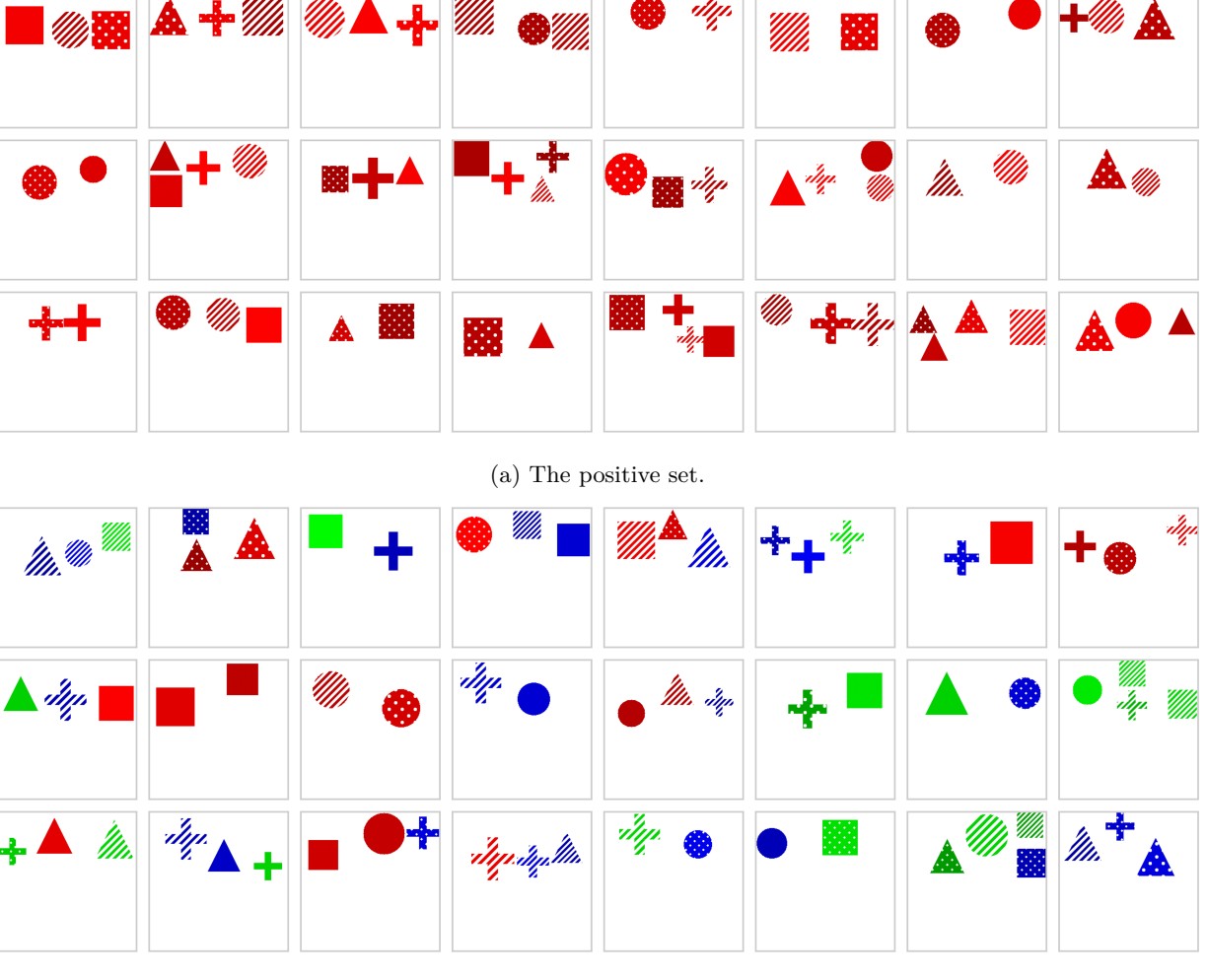

(a) The positive set.

(b) The negative set.

Figure 30: Example images for the positive and negative sets of the probe dataset for the `red top` in the simple elements dataset.

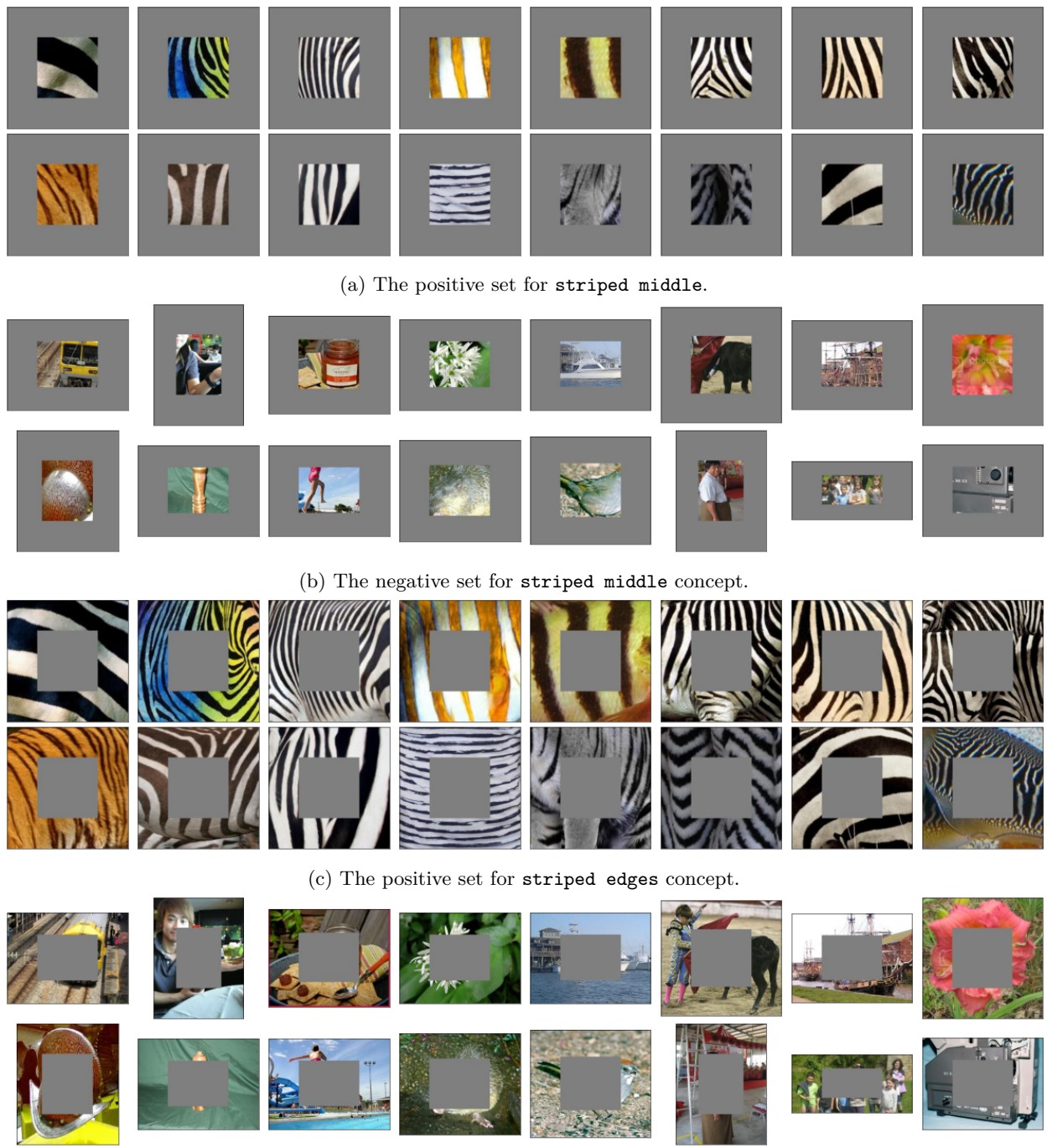

(a) The positive set for `striped middle`.

(b) The negative set for `striped middle` concept.

(c) The positive set for `striped edges` concept.

(d) The negative set for `striped edges`.

Figure 31: Example images from spatially dependent probe datasets for ImageNet.

## 14.2 Spatial Norms Details

When finding CAVs, we use the activations of layer $l$ in a convolutional neural network, which has shape $H \times W \times D$, where $H$, $W$ and $D$ are the height, width and number of channels, respectively. When finding the CAV, these activations are flattened to be vectors of length $m = H \times W \times D$. The value of each element in the activations depends on its specific height, width and depth indices. As a result, each corresponding

element of the resultant CAV is also index-dependent. We are interested in spatial dependence, so to visualise how a CAV varies across the width and depth dimensions we calculate the L2 norm of each depth-wise slice – the CAV's spatial norms.

### 14.3 Individual Spatial Norms

In fig. 32, we present the spatial norms for `striped` in a ResNet trained on the ImageNet dataset. Each heatmap is for a different random probe dataset, denoted by $r$. The different patterns in the heatmaps show that each individual $\boldsymbol{v}_{c,l}^r$ has a spatial dependency which differs across $r$. However, when we average the norms across multiple CAVs, $\sum_{r=1}^R \mathbf{S}_{c,l}^r / R$, we obtain a uniform distribution across the spatial dimensions. This uniformity suggests that the spatial dependencies of each individual CAV cancel out across multiple seeds, as depicted in the top rows of fig. 6 in the main text.

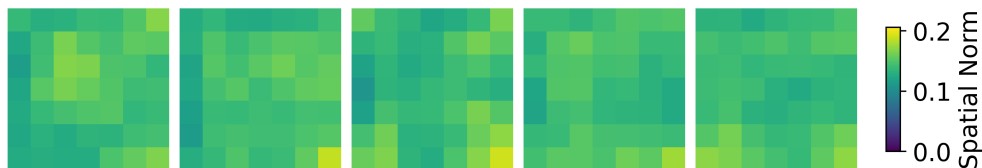

Figure 32: Individual spatial norms for `striped`, where each CAV was trained on a different negative probe set, for layer4.1 of a ResNet trained on ImageNet.

### 14.4 Additional Spatial Norms

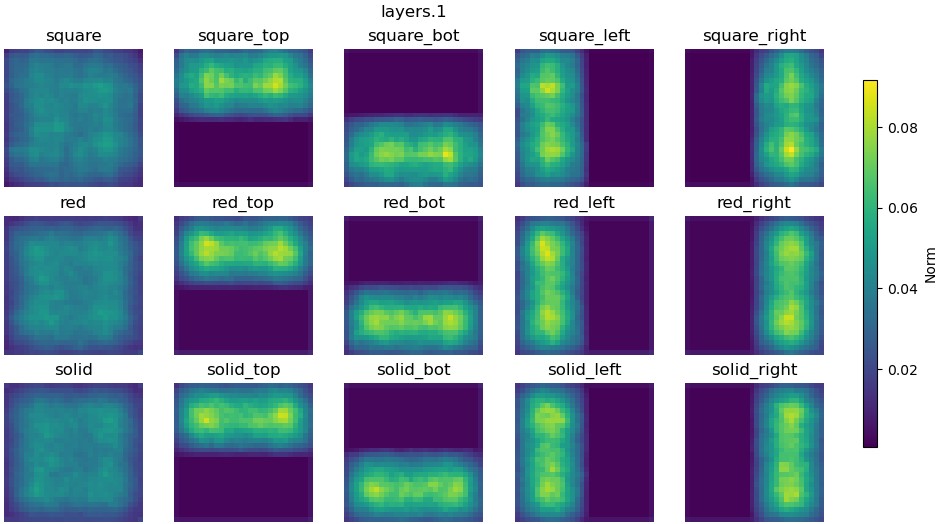

Figure 33: Mean CAV spatial norms across 30 CAVs for a selection of concepts in the Element dataset for the second convolutional layer.

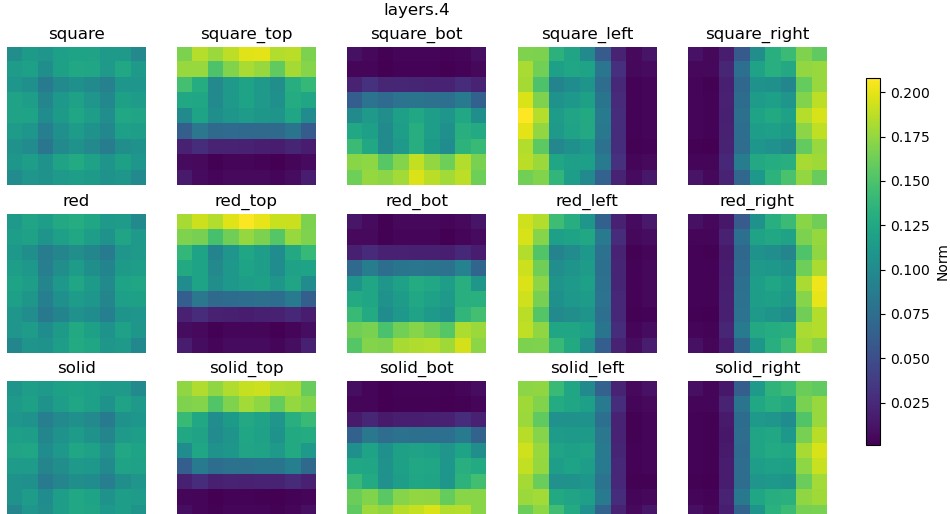

Figure 34: Mean CAV spatial norms across 30 CAVs for a selection of concepts in the Element dataset for the fifth convolutional layer.

## 14.5 Spatial Means

Instead of visualising the norm of each depth-wise slice, we can visualise the mean. We default to showing the norm because you could have a spatial mean of zero in a region of activation space which has a large effect on the directional derivative. This could occur, for example, if half the elements of the CAV are large and positive and the corresponding gradients are positive, and half the elements of the CAV are large and negative and the corresponding gradients are also negative. This would lead to that spatial region having a large contribution to the directional derivative, but the spatial mean would be close to zero. The spatial norm, however, would be large for this region. This makes the norm a better measure of the effect of each region, but the mean can still be useful to show the direction of that effect.

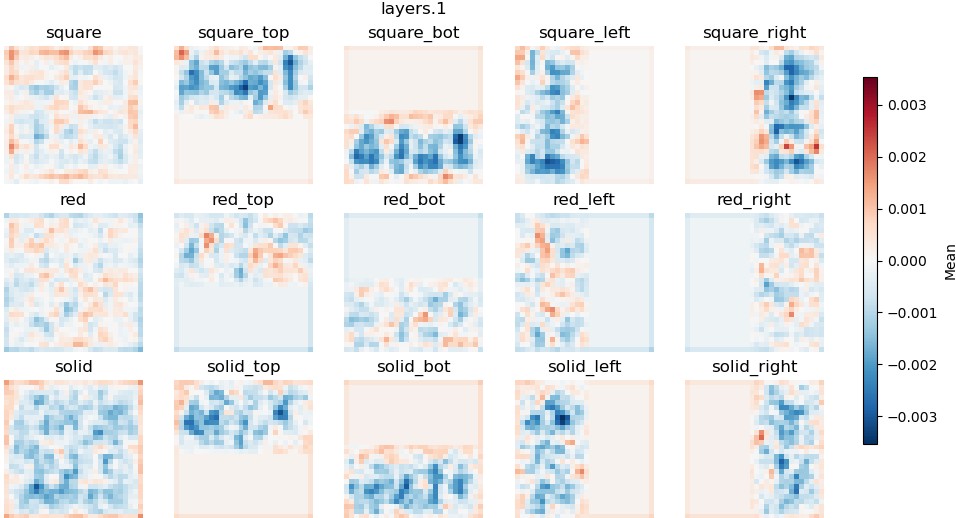

Figure 35: Mean CAV spatial means across 30 CAVs for a selection of concepts in the Element dataset for the second convolutional layer.

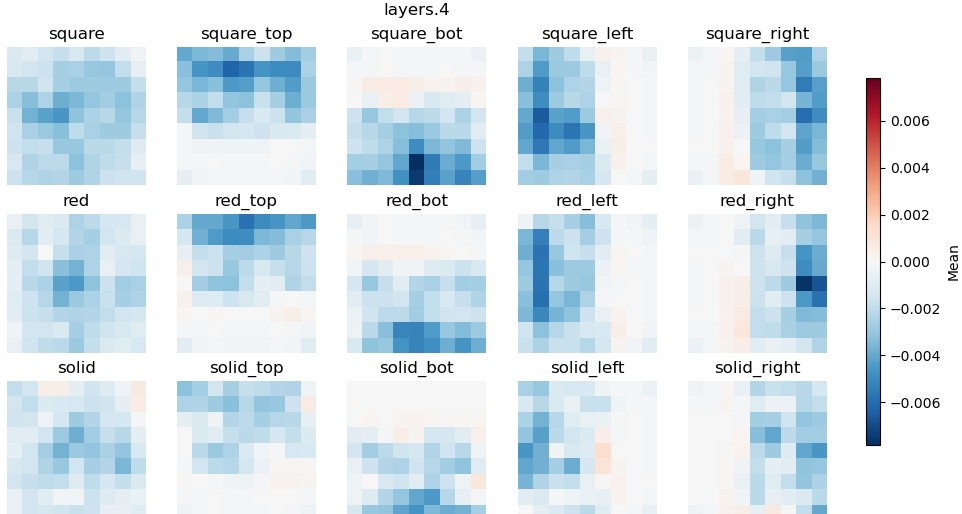

Figure 36: Mean CAV spatial means across 30 CAVs for a selection of concepts in the Element dataset for the fifth convolutional layer.

## 14.6    Spatially Dependent TCAV Scores

In this section we provide example TCAV scores which differ across complementary spatially dependent CAVs, , for at least one concept, the TCAV score is the opposite side of the null for the `edges` version of a concept compared to the `middle` version (or the `left/right` and `top/bottom` versions).

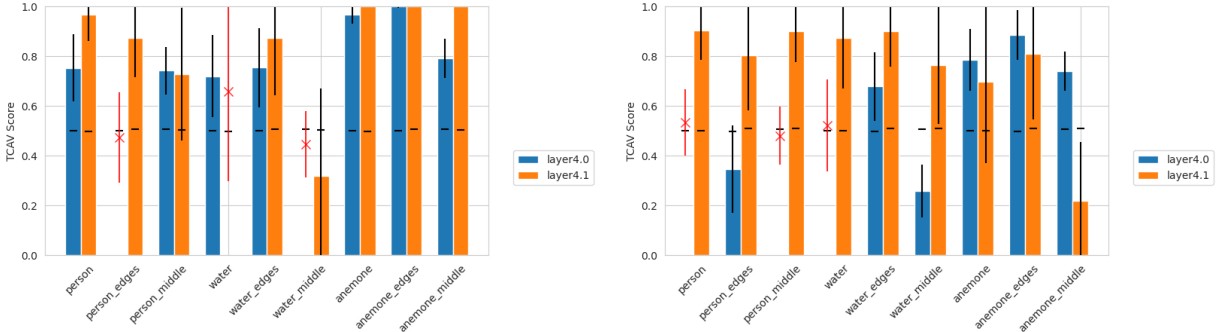

(a) TCAV scores for a selection of concepts for the 'anemone fish' class in ImageNet.

(b) TCAV scores for a selection of concepts for the 'spiny lobster' class in ImageNet.

Figure 37: Examples of spatially dependent TCAV scores in ImageNet. Each subfigure is a separate class. The standard deviation is shown in black for significant results and red for insignificant results. The mean TCAV score for random CAVs are shown as horizontal black lines.

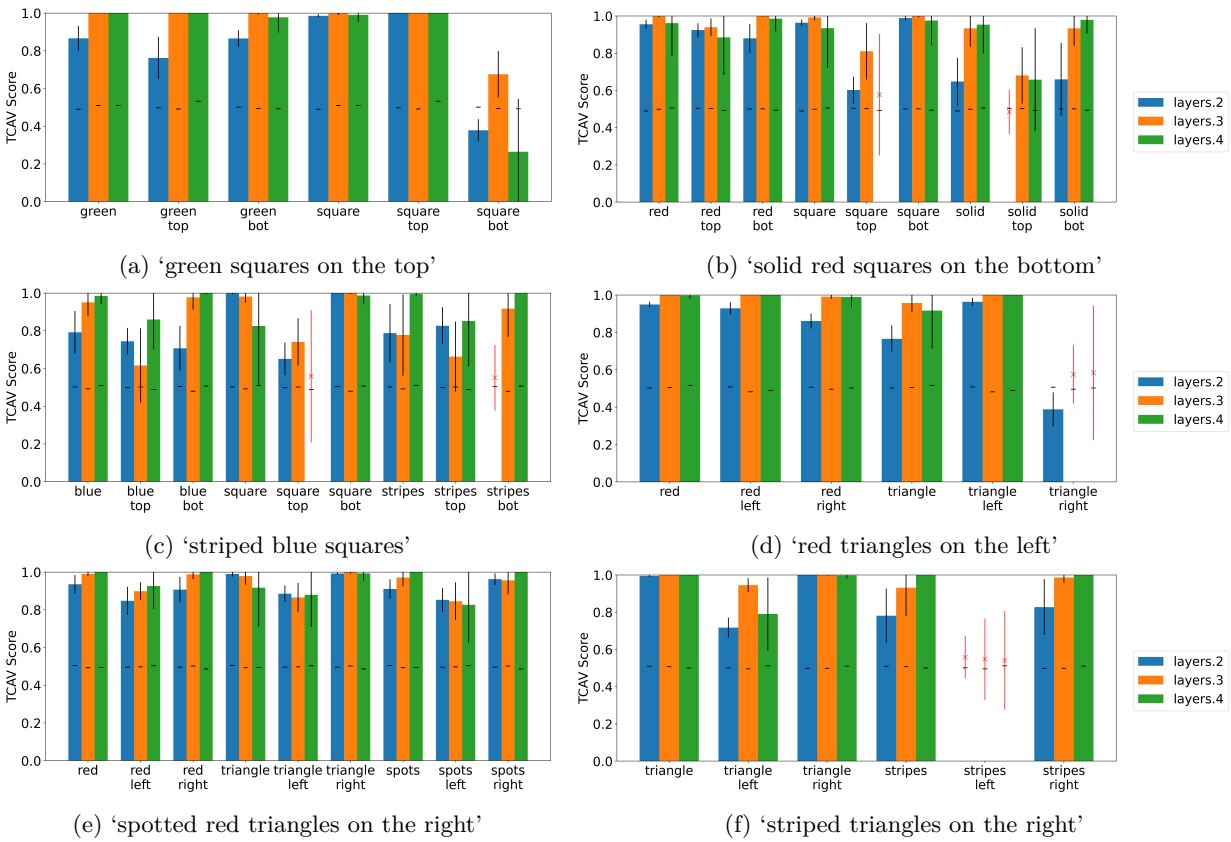

Figure 38: Examples of spatially dependent TCAV scores in the spatially dependent version of Elements for a ResNet-50. Each subfigure is a separate class. The standard deviation is shown in black for significant results and red for insignificant results. The mean TCAV score for random CAVs are shown as horizontal black lines.

## 14.7 Spatially Dependent CAVs in ViTs

Even though we cannot use spatial norms to visualise the spatial dependence of CAVs in a transformer based architecture, in this section we have some preliminary results suggesting that that spatially dependent CAVs can still be created for a ViT-B16 (Dosovitskiy et al., 2021). As for the ResNet-50, we test the model on the spatially dependent Elements dataset and found TCAV scores that vary by location.

We finetuned a ViT-B16 model pretrained on ImageNet for 50 epochs on the spatially dependent version of Elements (i.e. there are some classes which depend on the location of the objects as well as which concepts are present). We used an exponentially decaying learning rate with initial learning rate of 0.0001 and a $\gamma$ of 0.95. Similarly to the previous section, we demonstrate that we obtain different TCAV scores for the `left` versions of a concept than the `right`, although there is substantially less variation in TCAV score than for the ResNet-50 (section 14.6). Future work should perform more extensive experiments to determine how the properties analysed in this paper are affected by architecture.

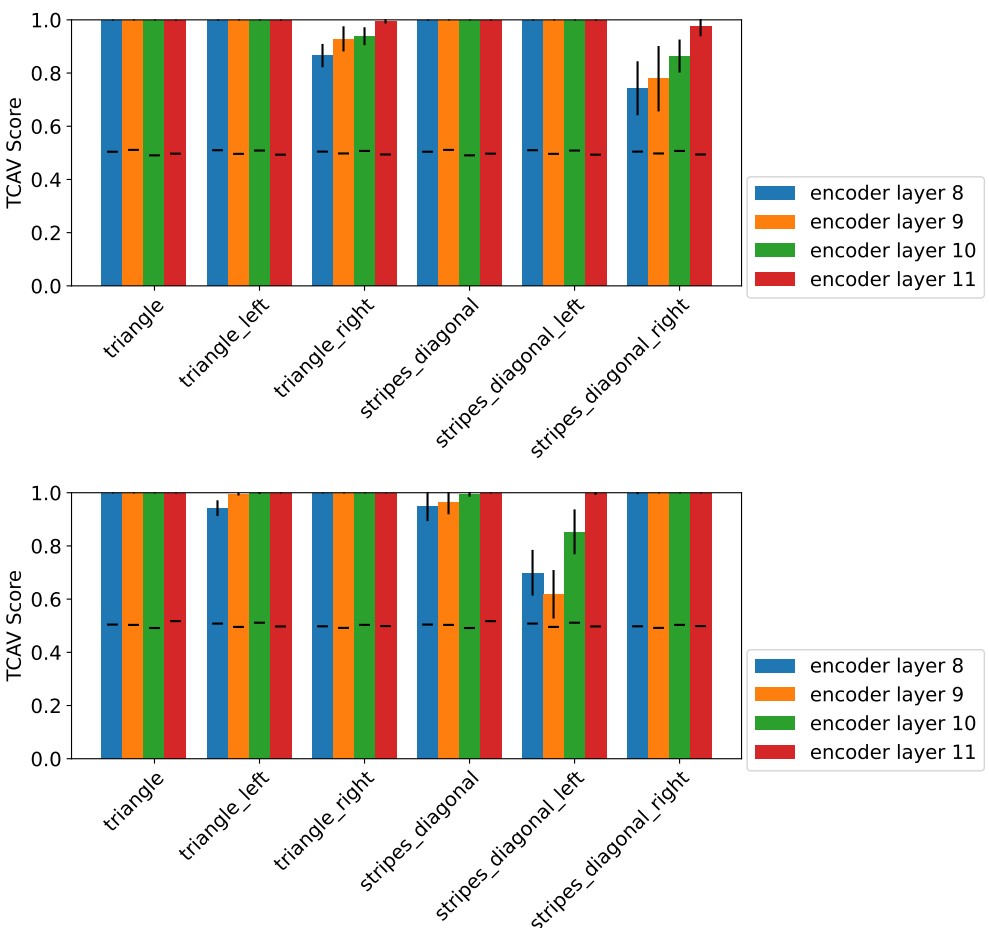

Figure 39: Spatially dependent TCAV scores in the spatially dependent version of Elements for a ViT-B16. The TCAV scores for the classes "striped triangles on the left" (top) and "striped triangles on the right" (bottom) are shown. The standard deviation is shown in black for significant results and red for insignificant results. The mean TCAV score for random CAVs are shown as horizontal black lines.

### 14.8 Dot product distributions

The definition of concept vector spatial dependence in eq. (8) compares a CAV, $\boldsymbol{v}_{c,l}$, with the activations of two positive probe datasets with different spatial dependencies, $\boldsymbol{a}^+_{c,l,\mu_1}$ and $\boldsymbol{a}^+_{c,l,\mu_2}$, by taking the dot product between them $\boldsymbol{a}^+_{c,l,\mu_x} \cdot \boldsymbol{v}_{c,l}$. In fig. 40, we show the distribution of dot products for three concepts and three test probe datasets in the spatially dependent version of Elements. The separation between the distributions for the `stripes left` and `stripes right` probe datasets (blue and green bars, respectively) for both the `stripes left` and `stripes right` CAVs (left and right plots, respectively) demonstrate that these CAVs are spatially dependent.

## 15 Further Related Work

Recent work highlighted problems with concept-based explanation methods. Ramaswamy et al. (2022a) showed that using different probe datasets to interpret the same model can lead to different explanations for the same concept. Similarly, Soni et al. (2020) showed these methods to be sensitive to the random seed used to sample images for the negative set. Our work complements this research by investigating the underlying properties of concept vectors and how they may cause problems when interpreting concept-based explanations.

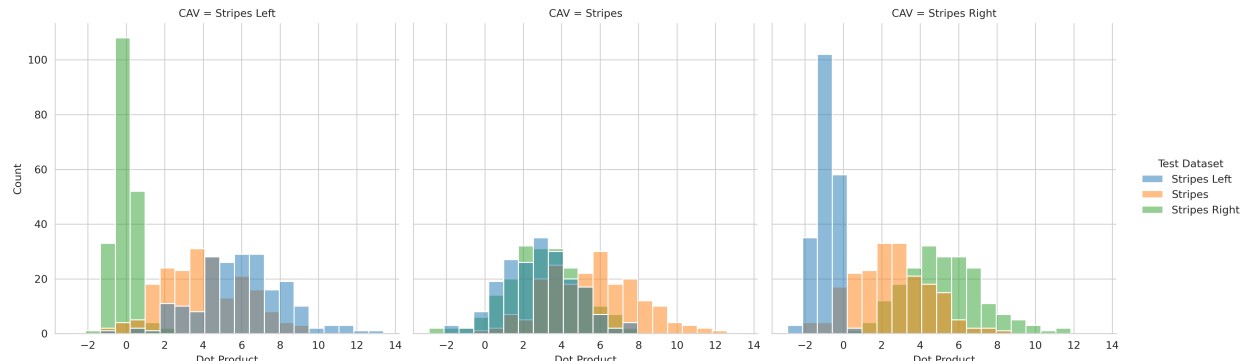

Figure 40: Distribution of dot products between spatially dependent CAVs and image activations ($\boldsymbol{a}^+_{c,l,\mu} \cdot \boldsymbol{v}_{c,l}$) for the spatially dependent Elements dataset. Each column is for different CAVs. From left to right these are: `stripes left`, `stripes`, `stripes right`. For each CAV we show the distribution for three positive probe datasets: `stripes left` (blue), `stripes` (orange), `stripes right` (green).

Extensions to the original TCAV have been suggested, attempting to improve aspects of the original method. For instance, Ghorbani et al. (2019) automate concept discovery by using super-pixels and clustering, removing the need to handcraft a probe dataset. Zhang et al. (2020) and Schrouff et al. (2021) change how CAVs are created to produce local and global explanations. However, these methods still use vectors to represent concepts (Alain & Bengio, 2017). As such, our work is still applicable to each of the extensions.

The properties analysed in this paper are generally applicable. To give insight into when the various properties may be relevant, we performed a review of papers which use CAVs in medical imaging and computer vision research. In each case, we checked if the authors had done any checks related to layer consistency, entanglement or spatial dependence of CAVs. Almost no papers evaluated the effect of these properties on their results, and when they did it was by creating CAVs in multiple layers, providing some robustness to layer inconsistency. In Table 3, we provide a list of these papers, detailing any use-cases where our recommendations could have helped check the impact of CAV properties on results.

Table 3: Examples in computer vision and medical imaging research, where consistency, entanglement and spatial dependence may impact analyses. We use the following abbreviations: skin cancer (SC), skin lesions (SL), breast cancer (BC), histology (H) CIFAR-10/100 (CF) (Krizhevsky, 2009), COCO (CO) (Lin et al., 2014), CUB (CB) (Wah et al., 2011), Places365 (Pl) (Zhou et al., 2017), Waterbirds (WB) (Sagawa et al., 2020), ImageNet (Im) (Deng et al., 2009)

| Property | Medicine | | | | CV Research | | | | | | Papers |
|---|---|---|---|---|---|---|---|---|---|---|---|
| | SC | SL | BC | H | CF | CO | CB | Pl | Wb | Im | |
| Consistency | ✓ | ✓ | | | ✓ | ✓ | ✓ | ✓ | | | Yan et al. (2023); Ramaswamy et al. (2022a); Fürböck et al. (2022); Ghosh et al. (2023); Lucieri et al. (2020); Yuksekgonul et al. (2023) |
| Entanglement | ✓ | ✓ | ✓ | ✓ | ✓ | ✓ | ✓ | ✓ | ✓ | ✓ | Yan et al. (2023); Ramaswamy et al. (2022a); Fürböck et al. (2022); Ghosh et al. (2023); Graziani et al. (2020); Lucieri et al. (2020); Pfau et al. (2020); Yuksekgonul et al. (2023) |
| Spatial Dependence | ✓ | ✓ | ✓ | ✓ | ✓ | ✓ | ✓ | ✓ | ✓ | ✓ | Yan et al. (2023); Ramaswamy et al. (2022a); Fürböck et al. (2022); Ghosh et al. (2023); Lucieri et al. (2020); Pfau et al. (2020); Yuksekgonul et al. (2023) |

