# OpenReview forum: "Explaining Explainability: Recommendations for Effective Use of Concept Activation Vectors"
_TMLR — Accepted by TMLR_

### Review · Reviewer_VsKT · 2024-11-16

**Summary Of Contributions:**

This paper examines three fundamental properties of Concept Activation Vectors (CAVs), a widely used post-hoc XAI approach that is the backbone of several modern interpretable architectures. In particular, this work studies whether CAVs are (1) consistent across layers, (2) entangled across related concepts, and (3) faithful to known spatial dependencies. This paper's main contributions are therefore:  (i) it provides a definition for CAV consistency, together with a metric to measure it, and shows that CAVs are rarely consistent and should therefore be observed from multiple layers within the model; (ii) it provides a formal definition for CAV entanglement and shows how cross-concept entanglement may lead to mistaken conclusions derived from T-CAV scores; (iii) it shows that CAVs can be spatially local (i.e., they may be localized to specific areas of the input) and that this locality could be detected via norm maps; and (iv) it introduces a new simple yet flexible concept-annotated dataset called Elements. These contributions are backed by empirical evidence on this work’s proposed dataset (Elements) and ImageNet. Moreover, using a real-world case study, this work showcases how all the considerations discussed in this paper can manifest themselves in a single practical scenario. Overall, this paper highlights some key limitations of concept-based post-hoc explanations while suggesting pragmatic considerations when deploying these methods.

**Audience:**

Yes

**Broader Impact Concerns:**

I do not believe there are any broader ethical implications from this work that would merit a Broader Impact Statement.

**Claims And Evidence:**

No

**Requested Changes:**

Below, I list some potential changes for each section and their importance in securing my recommendation for acceptance. If time is limited, please focus on all **critical** requests, followed by all **major** requests. If I misunderstood something at any point, just let me know, as this is always a possibility (apologies in advance if that’s the case!).

### General Requested Changes

- **(Major, Wording, Nitpicking)** When I read that one of the contributions was a “visualisation tool”, it might’ve accidentally biased me towards the wrong idea of the referenced contribution. I understand how visualising similarity matrices between CAVs can be seen as a "visualisation tool", however I would argue against using that term for it as it is potentially misleading and enlarges the extent of this particular contribution (visualising the cosine similarity matrices between vectors is a very common approach for analysis in ML, data science, DL, etc rather than an actual visualisation tool, in the sense of what “tool” is usually used for within these sorts of papers, e.g., a GUI).
- **(Minor, Feel free to ignore)** Given how a significant contribution of this work is to provide guidelines on how to properly use T-CAV, it would be extremely helpful (and potentially reach-increasing) to make sure the title of this paper reflects this.
- **(Minor, Nitpicking)** I would recommend ordering the appendices in the same order that they are referenced in the text and numbering their sections from a “conventional” starting point (e.g., §A.1 for the first appendix section rather than §9).

### Section 1 (Introduction)

- **(Major, Referencing)** The introduction makes several strong claims, all without any citations. At the very least, "saliency" should be defined for people not familiar with that term, and proper citations should be made to previous works arguing for the need for explainability/interpretability.
- **(Major, Clarity)** The enumeration of properties in this section jumps directly to stating conclusions (e.g., "They cannot be consistent across layers") rather than stating first the research questions this paper is exploring. This leads to an enumeration that does not naturally follow the preamble ("... we focus on understanding three properties of concept vectors:") and that rather quickly jumps to conclusions before a problem has been presented.
- (**Very Minor, Potential Typo)** In the caption of Figure 1, should “The top panel illustrates each …” be “The top *panels illustrate* each …” instead? This would also be consistent with the sentence that follows this one.
- **(Very Minor, Nitpicking, Word Choice)** Should “… a substantial impact on the meaning of a CAV” be “… a substantial impact on the *interpretation* of a CAV” instead?

### Section 2 (Background)

- **(Major, Notation)** The use of $\mathbb{X}_k$ and $\mathbb{X}_c$ to mean very different sets (one partitions a set based on task labels, the other partitions a set based on a concept) opens up the room for confusing notation (consider $\mathbb{X}_1$: is this the set representing all samples with label $y=1$ or with concept $c_1$?). I would suggest using different set names to represent these two different subsets.
- **(Minor, Potential Typo)** “is” is missing before “defined as” before defining $S_{c, k, l}$.
- **(Minor, Potential Typo)** “… output for class $k$ to the activation” should probably be “… output for class $k$ *with respect* to the activation”.
- **(Very Minor, Nitpicking)** “NN” is used before it is defined as a "Neural Network".

### Section 3 (CAV Hypotheses)

- **(Critical, Clarification, Concern)** In definition 1 there is an implicit choice to make the definition a function of a **unit shift** in the direction of the CAV (i.e., we want $ f(\mathbf{a}\_{l\_1} + \mathbf{v}\_{c, l\_1}) = \mathbf{a}\_{l\_2} + \mathbf{v}\_{c, l\_2} $ rather than the more general formulation $f(\mathbf{a}\_{l\_1} + \lambda \mathbf{v}\_{c, l\_1}) = \mathbf{a}\_{l\_2} + \lambda \mathbf{v}_{c, l\_2}, \forall \lambda \in \mathbb{R}$). Could you please clarify why this was done this way? And what is the argument to be made that a unit (or for that matter a $\lambda$-scaled) shift in the direction of the CAV in layer $l\_1$ should correspond to a shift of the same proportion in the CAV of layer 2? Intuitively, I can definitely see situations where a $\lambda\_1$-scaled shift in the direction of $\mathbf{v}\_{c, l\_1}$ in layer $l_1$ (i.e., $\mathbf{a}\_{l\_1} + \lambda\_1 \mathbf{v}\_{c, l\_1}$) may correspond to a $\lambda\_2$-scaled shift in the direction of $\mathbf{v}\_{c, l\_2}$ in layer $l\_2$ (i.e., $\mathbf{a}\_{l\_2} + \lambda\_2 \mathbf{v}\_{c, l\_2}$), where $\lambda\_2$ is a function of $\lambda\_1$. That seems to me to be a fairly reasonable definition of consistency. Can you please clarify why the definition you use, where the shift is expected and assumed always to be a unit shift, is the appropriate definition for consistency?
- **(Critical, Clarification, Concern)** After Definition 2, it is stated that “If concepts are entangled, it is not possible to separate the model’s sensitivity to one concept from its sensitivity to related concepts.” I can see the general intuition behind this; however, I am not entirely sure this is undesirable or unavoidable in real-world setups. Concepts are highly entangled as they are highly correlated in real-world situations. There are no assumptions or need for assumptions, stating that concept probes for CAVs should be trained on datasets containing completely independent concepts (if anything, part of their usefulness is that they allow this exact behaviour). Considering this, why should we ever expect that if CAVs satisfy definition 2, then it must not be true that “A CAV represents only the concept corresponding to the concept label of its probe dataset” as stated by NH2? Couldn’t it be that two CAVs are highly entangled just because the underlying concepts themselves are highly correlated, yet they still faithfully represent the concept they are intended to represent? In other words, **how can one claim that satisfying Definition 2 immediately leads to a violation of NH2?**
- **(Critical, Clarification, Concern)** Definition 3 seems a bit too hand-wavy and ill-defined. This severely complicates understanding its usefulness and its implications. I would ask the authors to please clarify what the “location” of a concept $\mu_i$ really means. How is this defined for concepts that are a function of multiple pixels (e.g., the example of $\texttt{stripes}$ itself can be seen not at a specific pixel location but at a set of pixels whose locations can vary)? How is this defined for high-level concepts that may not be easily mapped to specific features (e.g., a bird’s $\texttt{size}$ is a useful concept but one that does not correspond to a specific subset of pixels when working with bird images)? And how is this related to similar notions found in the literature (e.g., [9])? I hope these questions exemplify my concerns with the usefulness of this definition.
- **(Very Minor, Potential Typo)** “… an infinitesimal change of the activations …” should probably be “… infinitesimal change *on* the activations …”.

### Section 4 (Elements)

- **(Major, Literature Relevance)** Would it possible to describe in this section what the proposed *Elements* dataset, which is claimed as a contribution in the introduction and abstract, offers that existing similar datasets such as dSprites [1] and 3D Shapes [2], also used in the concept XAI literature (and the disentanglement literature), do not already offer? I think this may help build a case for how one should use the newly proposed dataset over already existing ones.
- **(Minor, Clarification)** Related to my question above I see that a similar comparison to the one requested above is made with CLEVR in Section 5. I can somewhat understand the efficiency argument made there. Yet, wouldn’t this be a one-time cost that is negligible overall, as once the dataset is generated, it doesn’t need to be generated again?

### Section 5 (Related Work)

- **(Minor, Missing Relevant Works)** **[Concept Correlation and Entanglement]** There are a couple of relevant works that could be helpful within this related works discussion. These works include those discussing the effect of concept correlations on concept representations [3, 4] and whether inter-concept relationships (which may imply correlations too) are captured by concept-based models (including T-CAV) [5]. I do believe there is a clear distinction between this paper and these other previous works, as this work focuses on T-CAV specifically, while the examples I included here (and others, e.g. [6, 7]) focus on studying the effect of concept correlation/entanglement on inherently interpretable concept-based architectures (e.g., CBMs).
- **(Minor, Missing Relevant Works)** **[Spatial Dependence]** There are some highly relevant works that may be worth discussing here as they study the spatial locality of concepts. These include one of the first works casting doubt on the validity of a CBM’s explanations [8] and more recent works explicitly studying whether the explicit locality of concepts is captured by inherently interpretable concept-based models [9]. Again, this is a minor change request as these works relate more to inherently interpretable architectures than post-hoc methods such as T-CAV. As such, I do not believe they hamper the novelty of this work.

### Section 6.1 (Consistent CAVs)

- **(Very Critical, Potential Soundness Mistake)** I am not very convinced of the main theoretical result of this section. If I understood it correctly (based on §6.1 and Appendix 9), the crux of the impossibility for consistent CAVs relies on the claim that if $f: \mathbb{R}^{m\_{l\_1}} \rightarrow \mathbb{R}^{m_{l\_2}}$ is not a linear function, then there does not exist $\mathbf{u} \in \mathbb{R}^{m\_{l\_1}}$ and $\mathbf{v} \in \mathbb{R}^{m\_{l\_2}}$ such that $f(\mathbf{x} + \mathbf{u}) = f(\mathbf{x}) + \mathbf{v}$ for all $\mathbf{x}$ in the domain of $f$. If this is indeed the main claim needed, then **I strongly believe this is not true**. As a pedantic yet trivial example, consider the case where $m\_{l\_1} = m\_{l\_2}$ and $\mathbf{v} = \mathbf{u} = \mathbf{0}$. In that instance, it is always true that $f(\mathbf{x} + \mathbf{u}) = f(\mathbf{x}) = f(\mathbf{x}) + \mathbf{v}$ for all $\mathbf{x}$.

As a more complex counterexample, we can consider periodic functions or piece-wise functions (as all you care about is that the difference between $f(x + u)$ and $f(x)$ is constant for all $x$). For example, if $f(x) = \text{sin}(x) + cx$ and $u = 2\pi$, $v = 2\pi c$ (for some $c \in \mathbb{R}$), then we have that $f(x + u) = f(x + 2\pi) = \text{sin}(x + 2\pi) + c(x + 2\pi) = (\text{sin}(x) + cx) + c 2\pi = f(x) + c 2\pi = f(x) + v$ for all $x$. This shows that such $u$ and $v$ can indeed exist, **contrary to the claim made in this work**. I grant that this is a trickier example. However, my whole point is that claiming that a function does not exist just because it may be a function of another variable is not necessarily a valid or sound proof, and it is a potentially flawed argument (as these counterexamples show). Before moving onwards, especially since this is a key contribution of this work, I believe this result should be properly fixed, adjusted, and formally proved so that it reflects a true statement (this may likely involve making some stronger assumptions on the form $f$, $u$, and $v$ may have).
- **(Very Critical, Potential Soundness Mistake)** Similarly, I have some concerns/hesitations about the ReLU and sigmoidal proofs (Appendix 9.1 and 9.2). First, when working with NN, the nonlinear function (e.g., sigmoid or ReLU) is applied to (usually) a linear transformation of the current layer’s input. Therefore, it will rarely be the case that the output of layer $l_2$ is just the relu or sigmoidal activation of the output of layer $l_1$ (this is the underlying assumption regarding those proofs). If the case that you are trying to build is that even in this simple instance, consistency is not possibly preserved (I have some doubts on this too tho), then you need to show that this implies that consistency is not possible anywhere downstream of layer $l_2$ (these statements by themselves say nothing as to whether consistency may be “recovered” down the line). More importantly, however, as in the proof in section 9, the key conclusion for both the ReLU and sigmoid proofs are derived from very hand-wavy arguments that may not necessarily be true (of a similar nature to the one I provided a counterexample above). As mentioned above, I would like to ask the authors to please make sure to properly formalize these statements and provide a sound proof once they have been formalized. All these proofs seem to depend on the false claim that for “a single $v$ to exist which is consistent for all $a_{l_1}$ it cannot depend on $a_{l_1}$”, something my counterexample above shows is not necessarily true. Because of this, I believe they require quite a lot of work before these theoretical results can be considered sound.
- **(Major, Clarity)** I would appreciate it if any theoretical results (e.g., proofs) could be formally written and expressed by enumerating their assumptions and conclusions in the form of a theorem. Otherwise, capturing the essence of each proof is hard as it requires going over many separate pieces where assumptions and conclusions are stated throughout.
- **(Major, Nitpicking)** Why is the conclusion “the projected CAVs have a nonzero error, indicating that vector addition is not preserved” unexpected when one considers that DNNs are by design non-linear?

### Section 6.2 (Entanglement)

- **(Very Minor, Potential Typo)** “… we investigate the effect of entangled concept vectors on TCAV score” should probably be “… we investigate the effect of entangled concept vectors on TCAV *scores*”.

### Section 6.3 (Spatial Dependence)

- **(Very Minor, Potential Typo)** “2. That each depth-wise slice…” should probably be “2. *Each* depth-wise slice…”.

### Section 7.2 (Results)

- **(Critical, Clarification, Potential Mistake)** At the beginning of Section 7.2, it is claimed that “as shown in fig. 7, initial results with random CAVs gave poor performance for the medical concepts, so we used the ‘absent’ category for each class as the negative set. This gave better performances …”. However, if one looks at Figure 7 and its caption, the top left pane (which the caption says are the CAVs trained with a random negative set, shows significantly more accurate CAVs than those in the top right pane (which the caption says are CAVs trained with “absent” samples in the negative set). These two appear to contradict each other directly. Am I misunderstanding something here? Or is this a potential labelling or plotting mistake?
- **(Very Minor, Potential Typo)** In the “Entanglement” paragraph, the closing quotation for “absent” is backwards.
- **(Major, Clarification)** In the “Entanglement” section, it is claimed that “We hypothesise that this is because the two concepts share the same negative set.” Is this something you could easily verify by changing the set and observing whether the similarity is still there? I am a bit puzzled by this result, as it appears to contradict things observed/discussed earlier in this paper when studying NH2.
- **(Minor, Figure)** Some of the error bars on the figures are cut (e.g., Figures 5 and 8). I recommend extending the figure limits so that these bars are not cut.
- **(Major, Clarification w.r.t. competing work)** This is roughly mentioned at a high level in the conclusion, but do you have a sense as to how the properties discussed in this paper may manifest or may be mitigated when getting rid of the linear-separator assumption of CAVs as Concept Activation Regions (CARs) [10] do?

## References

- [1] Matthey, Loic, et al. "dSprites: Disentanglement testing sprites dataset." May 2017.
- [2] Kim et al. "Disentangling by factorising." ICML (2018).
- [3] Heidemann et al. "Concept correlation and its effects on concept-based models." WACV (2023).
- [4] Espinosa Zarlenga et al. "Towards robust metrics for concept representation evaluation." AAAI (2023).
- [5] Raman et al. "Understanding Inter-Concept Relationships in Concept-Based Models."  ICML (2024).
- [6] Huang et al. "On the Concept Trustworthiness in Concept Bottleneck Models." AAAI (2024)*.*
- [7] Furby et al. "Towards a Deeper Understanding of Concept Bottleneck Models Through End-to-End Explanation." arXiv preprint arXiv:2302.03578 (2023).
- [8] Margeloiu et al. "Do concept bottleneck models learn as intended?." ICLR Workshop on Responsible AI (2021).
- [9] Raman, Naveen, et al. "Do Concept Bottleneck Models Obey Locality?." NeurIPS Workshop “XAI in Action: Past, Present, and Future Applications” *(*2023)
- [10] Crabbé et al. "Concept activation regions: A generalized framework for concept-based explanations." NeurIPS (2022).

**Strengths And Weaknesses:**

Thank you so much for submitting this work! I enjoyed reading this paper and appreciate the time taken to write it up in a very straightforward and clear manner. Below are what I believe are this paper’s main strengths, followed by what I believe are some of its weaknesses:

### Strengths

1. **[Significance] (Major)** I think this paper does a great job of introducing initial formalisms that capture several sorts of issues/problems that have been discussed for CAVs across the literature. In particular, I think this work’s take on studying CAVs is an excellent way to get a good practical understanding of how CAVs work and how you could use them. As such, I believe this paper has good potential to be useful/significant, particularly for practitioners (e.g., the skin cancer example is a good primer for using CAVs and analysing their scores).
2. **[Significance] (Major)**  This work provides several contributions: analysis and formalisms for three key properties of CAVs, a new synthetic dataset (Elements), a set of useful, practical guidelines for using T-CAV, and a set of examples showcasing failure modes of T-CAV. I am marking this strength as “major” rather than “minor” because although all of these amount to a good set of contributions, independently, they may be seen as small contributions on top of previous existing/related works. See below for specific examples/questions about this.
3. **[Quality] (Minor)** The plots, figures, and overall presentation are easy to follow and understand.
4. **[Clarity] (Major)** The paper is well-written and easy to read. There are a few smaller typos (see below), but they are all minor, so the overall flow is unaffected.

### Weaknesses

In contrast, I believe the following are some of this work’s limitations:

1. **[Soundness] (Critical)** I have some serious concerns/doubts about the theoretical results introduced in this paper. They lack formalizing and may be **potentially wrong/unsound**. Please refer to my requested changes in Section 6.1 for specific comments on this matter.
2. **[Significance/Originality] (Major)** Most of the results and ideas discussed in this work are not particularly counter-intuitive and are highly related to observations from previous works (some of which are, rightly so, cited by this paper as well). In particular, although I like how the null hypotheses are defined in Section 3, I would argue that all of these hypotheses are intuitively expected to be false from the way CAVs are defined. This does not mean this work is not useful (intuition, after all, can be very wrong), but it does diminish the potential impact and significance of the results discussed and introduced in this work. This is especially true since there are not a lot of other kinds of results (e.g., sound theoretical results) that could shine some light on the properties studied by this paper.
3. **[Quality] (Major)** Some of the definitions and assumptions made by this work (particularly on the theoretical formalisms) could benefit from better justification (at least at an intuitive level and at best with a theoretical backing). See below for specific examples of elements exemplifying this concern.
4. **[Quality/Clarity] (Minor)** A good part of the experimental section is spent making conjectures or claims about potential reasons for the observed results. While insightful, it is challenging from the reader’s point of view to understand which conclusions could be victims of confirmation bias unless more evidence is provided to back them up. This is particularly true in the case study, where there is a lot of subjective interpretation of CAV scores.

---

> ### Author Response · Authors · 2024-12-22
> **Response to reviewer VsKT (Part 1/2)**
>
> Thank you for your detailed review. We have ordered our responses by importance and summarise and/or quote which comment we are responding to in each case. Due to character limitations we have had to split our response into two parts. For any comments regarding the consistency proofs, please see our separate response.
>
> >**(Critical, Clarification, Concern)** After Definition 2, it is stated that “If concepts are entangled, it is not possible to separate the model’s sensitivity to one concept from its sensitivity to related concepts.” I can see the general intuition behind this; however, I am not entirely sure this is undesirable or unavoidable in real-world setups.
>
> Definition 2 simply defines entanglement, it does not say whether we expect CAVs to be entangled or not. We agree with you that we should not expect concepts to be completely disentangled and we state this at the beginning of section 6.2. However, in our work we demonstrate that when concepts are associated it can lead to misleading explanations. Our recommendations suggest visualising the similarities between concepts and checking that the associations are plausible, not to make sure that there are no associations between any concepts. We have edited the introduction to make it clearer that some of these properties are expected but that our aim is to formally define them, demonstrate their presence, and examine the implications on the interpretation of TCAV scores.
>
> > **(Critical, Clarification, Concern)** The spatial dependence definition (Definition 3) seems a bit too hand-wavy and ill-defined.
>
> Spatial dependence, as described in this paper, is about the location of a concept relative to the frame of the image. We have edited section 3.3 to make this clearer. In our experiments, we use simple binary labels referring to whether the concept is in the top/bottom of the image or the middle/outside of the image but more complex representations could be used. For the bird size example, a spatially dependent CAV for bird size would mean that the CAV has high similarity to the image only when the bird is in a specific region of the image (e.g. in the middle or near the edge of the image) .
>
> Regarding [9], this paper examines the locality of concept bottleneck models where they find that changing the pixels of parts of the image not related to the object of interest can affect the concept predictions about the object, i.e. the CBM is not making concept predictions using purely local features around the object. This is interesting, but a very different definition to our exploration of spatial dependence. For activation spatial dependence we simply want to know if the activations vary depending on where the concept is present in the image (which is an inherent part of CNNs) and for concept vector spatial dependence we want to know if a CAV can be trained which has a different similarity to the activations of an image depending on where a concept is present in the image.
>
> >**(Critical, Clarification, Potential Mistake)** Figure 7 Caption not matching the text in Section 7.2.
>
> Thank you for picking up on this. There is a typo in the caption of Figure 7. We have fixed this in the current version of the paper.

---

> ### Author Response · Authors · 2024-12-22
> **Response to reviewer VsKT (Part 2/2)**
>
> > **(Major, Wording, Nitpicking)** Using the term “visualisation tool” is misleading.
>
> We have rephrased this in the updated manuscript.
>
> > **(Major, Referencing)** Missing citations in the introduction. "saliency" is not defined.
>
> We have added an explanation of saliency and some additional citations to the introduction.
>
> > **(Major, Clarity)** The enumeration of properties in the introduction jumps directly to stating conclusions (e.g., "They cannot be consistent across layers") rather than stating first the research questions this paper is exploring.
>
> We think succinctly stating our conclusions is a clear way to communicate the findings of our work. The remaining paper can go into more detail, but we favour keeping the introduction brief.
>
> > **(Major, Literature Relevance)** What is the benefit of the Elements dataset over existing similar datasets such as dSprites [1] and 3D Shapes [2]?
>
> Both dSprites and 3D Shapes are very relevant datasets. Thank you for bringing them to our attention. We have added a paragraph in our Related Work section explaining the difference between these datasets and Elements.
>
> > (Minor, Clarification) A comparison has already been made to CLEVR in Section 5. I can somewhat understand the efficiency argument made there. Yet, wouldn’t this be a one-time cost that is negligible overall, as once the dataset is generated, it doesn’t need to be generated again?
>
> The dataset would be overly large if we pre-generated all possible images. Instead, we generate the data as/when we need it.
>
> > **(Major, Nitpicking)** Why is the conclusion “the projected CAVs have a nonzero error, indicating that vector addition is not preserved” unexpected when one considers that DNNs are by design non-linear?
>
> You are correct. It is expected that NNs have a non-zero consistency error. We have updated the text to make it clear we are not surprised by this result.
>
> > **(Major, Clarification)** In the “Entanglement” section of section 7.2, it is claimed that “We hypothesise that this is because the two concepts share the same negative set.” Is this something you could easily verify by changing the set and observing whether the similarity is still there?
>
> We do test for this using exactly the experiment you suggest! Two sentences later we state “In addition, if compared to the similarities between CAVs trained using random images as the negative set (the left of fig. 9), we see that this pattern disappears.“
>
> > **(Major, Clarification w.r.t. competing work)** This is roughly mentioned at a high level in the conclusion, but do you have a sense as to how the properties discussed in this paper may manifest or may be mitigated when getting rid of the linear-separator assumption of CAVs as Concept Activation Regions (CARs) [10] do?
>
> We do not have a good intuition as to how these properties might relate to other forms of concept representation. We think it likely that each property would still be present for CARs as many of the properties are at least partially caused by the contents of the probe datasets, rather than the vector nature of CAVs, however without further experimentation or analysis that is pure speculation on our part so we do not wish to make further statements in the paper.
>
> > **(Minor, Feel free to ignore)** An important contribution of the work is the recommendations provided to practitioners but this is not reflected in the title of the paper
>
> We have changed the title of the paper to: “Explaining Explainability: Recommendations for Effective Use of Concept Activation Vectors”
>
> > **(Minor, potential typo)**
>
> Thank you for finding various typos in the text which we have now fixed.
>
> > **(Minor, Missing Relevant Works)**
>
> Thank you for bringing our attention to these works. We have edited the related work section for both concept entanglement and spatial dependence accordingly.

---

### Review · Reviewer_rcyS · 2024-11-17

**Summary Of Contributions:**

The paper investigates three key properties of Concept Activation Vectors (CAVs): inconsistency across layers, entanglement with other concepts, and spatial dependency. The authors explore each property and apply their findings to a practical example involving a model fine-tuned on a medical dataset. Authors provide recommendations for practitioners on how to mitigate the impact of these properties when using CAVs for model interpretation.

**Audience:**

Yes

**Claims And Evidence:**

No

**Requested Changes:**

### Critical to securing the recommendation for acceptance:
Addressing the Weaknesses discussed in the Strengths And Weaknesses section.

### Would strengthen the paper:
1. Authors state research questions as Null Hypothesis, that naturally invoke the expectations that some statistical test would be employed to test the hypothesises -- I would propose to rename this for the Research Questions instead
2. Hypothesis 3 is "Concept activation vectors cannot be spatially dependent" -- I would propose for a bit of a clarity to rename this to "CAVs are spatially independent"
3. Before the Equation 9 there is an new line inserted could be deleted to save space and make the style consistent with other equations
4. Missing discussion on the general limitations of the CAV approach -- the concepts are mapped *to* the model, therefore we can only find something we already know. Other methods in concept-based and mechanistic interpretability aim to explain features in the models, therefore in general being able to find some novel, previously unknown concepts.

**Strengths And Weaknesses:**

### Strengths
- **Clarity and Presentation**: The paper is well-written with a clear presentation.
- **Insightful Investigation**: It explores interesting problem of practical limitations of CAVs.
- **Practical Application**: Interesting application to the model trained on the ISIC dataset.
- **Recommendations for Practitioners**: Clear recommendations for the practitioners who to mitigate described limitations of CAVs.

### Weaknesses

- **Dependence on Underlying Data**: The CAV approach heavily depends on the dataset employed for probing. The paper focuses on the CAVs directly but approaches the stated problems/limitations only from the model's perspective, not considering that the problem might lie in the data, that is used for probing. It is possible that the entanglement or spatial dependency of CAVs arises from the data used to compute these concepts, and not in the model representations.

- **Lack of Rigor in Hypotheses**:
  - **Null Hypothesis 2 (NH2)**: The formulation lacks rigor and depends on the datasets employed. In the case of "stripe" and "red" concepts, the entanglement might arise if the "stripes" dataset includes images with red stripes, indicating a dataset issue rather than a problem with CAVs.
  - **Null Hypothesis 3 (NH3)**: The discussion on spatial dependency lacks depth regarding what constitutes a "concept." For example, this research question could be perceived not as discussion on spatial dependency of CAVs, but rather as discussion about that probing datasets used for CAVs do not have all the concepts, that model could have learned (i.e. that concept "stripes" should be not one, but more concepts "stripes on the left/right/up/down".

- **Section 6.1**: - The impossibility of consistency is demonstrated only for functions $f$ that are a single ReLU or Sigmoid layer. The analysis does not consider more complex functions that involve multiple nonlinearities or non-linear mappings, that are more realistic examples, given that $f$ is introduced as a mapping between different layers of Deep Neural Network. For vectors $A \in \mathbb{R}^N$ and $B \in \mathbb{R}^K$, there exists a function $f: \mathbb{R}^N \rightarrow \mathbb{R}^K$ that does not conserve vector addition (i.e.,  $f(a + b) \neq f(a) + f(b)$ ) but satisfies $f(x + A) = f(x) + B$ for all $x \in \mathbb{R}^N$. Such function could be build as follows: assume $A = MB$, where $ M \in \mathbb{R}^{K \times N}$.  $f$ could be defined as $f(x) = Mx + H(x)$, where $H: \mathbb{R}^N \rightarrow \mathbb{R}^K $ s a periodic function with period $A$ (i.e., $H(x + A) = H(x)$). Then, $\forall x \in \mathbb{R}^N: f(x + A) = M(x + A) + H(x + A) = Mx + MA + H(x) = f(x) + B$.
This suggests that theoretically non-linear mappings can produce consistent CAVs.
The paper does not address why, in practice, the mapping between layers cannot approximate such functions.
- **Limited Quantitative Experiments**: The investigation significantly relies on qualitative demonstrations, such as the "striped" concept in Figure 3. Assessing the hypotheses across various models and a broader set of concepts would strengthen the conclusions. Without extensive quantitative analysis, the results may be anecdotal and not indicative of general trends. As a solution, a broader spectre of models and datasets could be employed, and results could be shown not on individual concepts, but rather on some qualitative general metrics.

### Minor Weaknesses
 - Definition 1: in description layers are defined as $l_1, l_2$ but later actors write that $l_1 < l_2$ -- while conceptually it is clear what is meant there, I believe the inequality between layers is not appropriate way to state that one layer is preceding another
- Definition 3: the  \phi is not defined

---

> ### Author Response · Authors · 2024-12-22
> **Response to reviewer rcyS**
>
> Thank you for your feedback on our work. Below, we cover each of your comments and clarify where we have made changes in the paper relating to each one.
>
> Both **Dependence on Underlying Data** and **Lack of Rigor in Hypotheses** relate to the properties we discuss in the paper being about the choice of probe dataset, rather than the CAVs themselves.  We completely agree that the choice in probe dataset has a substantial impact on both spatial dependency and entanglement, and we have shown this in several of our experiments. However, just because the probe dataset can cause the property to be present, does not indicate the property is not about the CAVs. In our updated manuscript, we have emphasised the effect of the probe dataset and how the properties can affect the interpretation of CAV-based explanations. Below, we address specific sections of your review:
>
> > It is possible that the entanglement or spatial dependency of CAVs arises from the data used to compute these concepts, and not in the model representations
>
> CAVs can be entangled because either the probe data is entangled or the model has entangled representations of the concepts (or both). However, our results show that regardless of the mechanism, entangled CAVs can cause misleading explanations.
>
> > In the case of "stripe" and "red" concepts, the entanglement might arise if the "stripes" dataset includes images with red stripes, indicating a dataset issue rather than a problem with CAVs.
>
> As discussed in the last point, we agree that CAVs can be entangled because of the probe dataset. However, this does not mean the entangled CAVs cannot lead to misleading explanations.
>
> > this research question could be perceived not as discussion on spatial dependency of CAVs, but rather as discussion about that probing datasets used for CAVs do not have all the concepts, that model could have learned
>
> Once again, this relates to whether the property is an issue with CAVs or with the probe dataset used to create the CAV. Similarly to our previous response, even if the root cause is because of a spatially dependent probe dataset, it can lead to a spatially dependent CAV and hence a spatially dependent TCAV score.
>
> **Section 6.1 (Consistency proof)**
>
> Please see our separate response
>
> **Limited Quantitative Experiments**
>
> We agree that we use a lot of individual examples in our paper, however this is necessary to show the effect of the properties on the interpretation of individual explanations. We believe we have already demonstrated our results on a broad array of concepts and data (Elements, ImageNet and ISIC-2019) in both the main paper and our extensive appendix. However, you are correct to point out that we do not have summary statistics across multiple concepts/layers/datasets. This is broadly because there is not a good justification to combine the results of separate layers/concepts into a single number as the TCAV scores of different layers should each be compared to the null in that layer, rather than to each other. However, thanks to your feedback, we have included a metric related to the prevalence of consistent TCAV scores in each dataset in order to demonstrate how often inconsistent TCAV scores occur. We name the metric TCAV layer consistency score. Please see the end of section 6.1 where we define the metric and discuss some results on ImageNet and Elements. Appendix 12.5 contains some further detail on these experiments.

---

### Review · Reviewer_yta8 · 2024-12-08

**Summary Of Contributions:**

In this paper, the authors analyze a popular post-hoc interpretability method - Testing using Concept Activation Vectors (TCAV).

Cnt1. Particularly,  this work analyzes 3 properties of CAVs - consistency across layers, the entanglement of CAVs among each other and that CAVs are spatially dependent. The authors propose experiments that demonstrate how the three aforementioned properties are satisfied/not satisfied.

Cnt2. The datasets utilized are both synthetic (Elements) and real-world (Lesions, ImageNet, etc.) making the analysis comprehensive.

Cnt3. Practical recommendations are given by the authors on how to best utilize TCAV for a real-world problem.

**Audience:**

Yes

**Claims And Evidence:**

Yes

**Requested Changes:**

(major) C1. Incorporating ViT and similar backbones for a more thorough analysis (as mentioned in W4) \
(major) C2. Improve experiment design for entangled concept vectors (as mentioned in W2) \
(major) C3. The Elements dataset is too simplistic to test properties on spatial dependence. Please consider adding more complex datasets in the analysis. \

(minor) C4. Formulation of the experiment around entangled concept vectors: As mentioned in W2, please justify why the hypothesis in Eq 2 holds \
(minor) C5. Section 12.3 is unclear. I do not understand why DeepDream-type visualizations help in CAVs. Visualizations like these are not strictly related to CAVs and can be done using any sampling techniques (activations, PCA, etc.)

**Strengths And Weaknesses:**

## Strengths:

S1. Good work discussing the properties of TCAV which indeed has not been well explored by related works.
S2. Range of datasets considered is good and a good mix of real and synthetic data has been used to test out the proposed properties.
S3. Appendix section is strong and exhaustive with results on a real-world ImageNet dataset included.
S4. Synthetic Dataset Contribution: "Elements" dataset is a notable contribution and can be used further for controlled experiments to understand CAV properties better and provides a resource for the community to explore these issues further.


## Weakness:

W1. Lackluster properties discussion: The properties introduced and tested provide a good overview of CAVs, however in my opinion are not the most important considerations for TCAVs. Practitioners using such explainability methods are often more interested in learning about the robustness of the method itself [1], its behavior under distribution shift and adversarial attacks [2], etc. This weakens the motivation for the paper. In addition, it is not surprising that CAVs for different layers are not similar (information encoded in every layer is different) making the analysis simple and less useful. Properties 2 (entanglement) and 3 (spatial dependence) are more interesting for TCAV and unfortunately, the discussion on those is also not satisfactory (See W2 and W3).

W2. Experiment design for entangled concept vectors: Analysis based on Equation-6 does not seem correct to me as it assumes that there is sufficient semantic gap between concepts. There can be multiple instances where $a^+ . v_{c,l}$ can be high if a significant portion of two concepts are related (for example vertical and horizontal stripes). Even though concepts are easy to isolate in Elements dataset, it might not be possible in more complex data.

W3. Spatial dependence is not a 'concept-property': Even though Elements dataset captures the location of the objects pretty well and the concepts can be seen to be spatially correlated in the heatmaps, the choice of the dataset makes the property analysis too simplistic. As concepts are usually more abstract and cover multiple parts of the objects in the image themselves, it is not clear how this property will be satisfied for more complex, distributed concepts. For example, consider 'stripes' and the object in question is a Zebra, where stripes are all over the body and cannot be localized as in the Elements dataset (example from TCAV paper), this analysis does not make sense. I hope the authors will think deeply about such an example and perform experiments validating this.

W4. TCAV for more vision backbones: In W1, I mentioned that spatial dependence is an interesting property. I would like to see how this changes when the backbone of the classifier itself is changed to architectures with different inductive biases than CNNs (like ViTs). If spatial dependence results are still valid, we can be sure of the property itself.

[1] Fooling Explanations in Text Classifiers, Ivanky et al.
[2] Interpretation of Neural Networks is Fragile, Ghorbani et al.

---

> ### Author Response · Authors · 2024-12-22
> **Response to reviewer yta8**
>
> Thank you for your review. We cover each of your comments below and specify where we have changed the paper in each case. Broadly, we have laid out our response in how it relates to each weakness, rather than to each requested change.
>
> > W1. Lackluster properties discussion
>
> We disagree with the statement that other properties of CAVs may be more interesting. Each of the properties we have analysed can affect the interpretation of TCAV scores, and each property can lead to misleading explanations. We clearly demonstrate this using a practical example in section 7. Regardless, it is difficult to claim what topics practitioners are more/less interested in without providing evidence.
>
> > it is not surprising that CAVs for different layers are not similar
>
> We agree! But we don’t believe anyone has done any experiments to check how similar they are once projected into the same space or, more importantly, discussed the implications on CAV-based explanations, e.g. TCAV scores. Namely, that TCAV provides layer sensitivity, not model sensitivity, so when different layers give different TCAV scores it is difficult to interpret the overall model sensitivity to the concept. We have edited the introduction and the discussion in our results around consistency (section 6.1) to make it clearer that it is not a surprising result but that the implications are underexplored and are important.
>
> > W2. Experiment design for entangled concept vectors
>
> Definition 2 simply defines entanglement, it does not say whether we expect CAVs to be entangled or not. We agree that we should not expect concepts to be completely disentangled. For example, the sky is often blue, so we might expect a sky CAV to be associated with a blue CAV. However, just because it is expected does not mean it can’t cause misleading explanations. For example, let’s say that we only test for the sky concept. We might find a TCAV score close to one, indicating that the model is sensitive to the presence of the sky, when in fact it is simply sensitive to whether there is blue in the image. We discuss this at the beginning of section 6.2 and we have edited the introduction to make our position more clear. By exploring the associations between different CAVs we can check for associations such as this. Our recommendations suggest visualising the similarities between concepts and checking that the associations are plausible, not to make sure that there are no associations between any concepts.
>
> > W3. Spatial dependence is not a 'concept-property':
>
> - Spatial dependence, as described in this paper, is about the location of a concept relative to the frame of the image. We have edited section 3.3 to make this clearer. For example, we use the top/bottom of the image or the middle/outside of the image because location (in this context) is relative to the frame of the image, not to any objects within the image. Our discussion around spatial dependence is not around creating a CAV which means “stripes on a zebra” or “stripes not on a zebra” but it is on creating CAVs meaning “stripes on the left of the image” . It may be possible to create such CAVs by carefully selecting the probe dataset, but that is beyond what we explored in this paper and would likely involve careful selection of the probe dataset.
> - In regards to performing some of the spatial dependence experiments on more complex data than Elements, we already demonstrate some of the same properties for ImageNet in both the main paper (Section 6.3) and the appendix (Section 14). In addition, in Section 7.2, we demonstrate how an understanding of spatial dependence led to CAVs which better represent confounding concepts for the ISIC 2019 dataset.
>
> > W4. TCAV for more vision backbones
>
> - The visualisations we use in the spatial dependence experiments (with the spatial norms) are not possible for a ViT as the activation space does not have a HxWxD structure for us to calculate norms over. We have added this limitation to section 6.3.
> - In terms of whether you can have activation or concept vector spatial dependence for a ViT, we have added some preliminary results in Appendix 14.7 indicating that you can create spatially dependent CAVs for a ViT-B16 but we leave more extensive exploration of this question for future work.
>
> > (minor) C5. DeepDream
>
> - This was an interesting additional experiment we performed to highlight that CAVs for the same concept but in different layers represent different aspects of the same concept. We found the visualisations a more intuitive way of viewing the phenomena, rather than the numerical results in the rest of the paper.
> - You are correct that we don’t need CAVs to perform DeepDream-type visualisations. In our work we are simply using DeepDream to visualise the CAVs

---

### Author Response · Authors · 2024-12-22
**General Response**

We thank the reviewers for their valuable feedback, which has been incorporated into the paper and undoubtedly improved our work. Overall, reviewers found the paper to be an interesting analysis of the properties that can affect CAV-based explanations. In particular, reviewers liked our practitioner recommendations and the practical application of them on the ISIC melanoma dataset and the reviewers each agreed that the writing and presentation of our work was clear.

**Reviewer yta8** agrees the properties of TCAV have not been well explored in previous literature and appreciates our extensive results/appendices demonstrating these properties on a wide range of data.

**Reviewer rcyS** highlights our recommendations to practitioners and example use-case as a particularly strong contribution and we thank the reviewer for highlighting the dependence on probe data as a significant factor in CAV properties. We completely agree and have made this more clear in the discussion of the properties, particularly when each is introduced.

**Reviewer VsKT** responded with a particularly thorough review and we thank them for their time and effort spent to improve our work. They agree that our provided recommendations will be useful in practice and that we highlight some of the key limitations of concept-based post-hoc explanations.

For most points we respond to each reviewer individually, however two of the reviewers highlighted issues with our consistency proofs so we respond to these separately. In the revised manuscript we have addressed many of the comments and to ease the response from reviewers we highlight changes in the main text in blue.

---

### Author Response · Authors · 2024-12-22
**Theoretical Results**

Two reviewers (rcyS and VsKT) brought up issues with the definition of layer consistency and the proofs relating to it. We thank the reviewers for bringing these to our attention and have made substantial edits to the proof to account for them. We have moved most of the proof to the appendix where we can go through it in more detail and simply outline the approach and key result in the main text. Below we detail some of the issues the reviewers had and our relevant changes:

1. It is not clear what the assumptions are and the proof seems to rely on “hand-wavy” arguments

We have rewritten (and changed) the proof, making it clearer what the main assumptions are and going through a more formal proof. This makes it much clearer when the proof holds

2. The proof does not hold for some examples

Both the periodic function and $u=v=0$ counterexamples are not relevant because of $||v|| \neq 0$ and  $||u|| \neq 0$ (see updated proof in the appendix for an explanation). This assumption is valid because CAVs are directions in activation space so, by definition, they cannot have zero norm.

3. The individual proofs are only demonstrated for simple examples (ReLU and Sigmoid)

The main proof now holds in the general case. We provide more detail for the ReLU and sigmoid functions as an example and because they are commonly used activation functions.

4. Why should CAVs in different layers be consistent to a perturbation by a unit vector, instead of a more general change of $\lambda_1 u$ and $\lambda_2 v$?

This is correct, there is no reason to expect a change by a unit vector in one layer should be the same as a change by a unit vector in another. The proof handles this by not assuming that the vectors are unit vectors and simply that they are vectors of non-zero norm. In practice we have to decide how to scale the two perturbations. We do this by scaling the unit CAV vectors by some small lambda (typically 0.01) and the norm of an example set of activations in that layer.  We provide detail on this choice of scaling in appendix 12.1 and have added a longer explanation and more results demonstrating that the scaling is suitable

---

### Decision · Action_Editor_x37L · 2025-01-22

**Recommendation:** Accept as is

**Comment:**

This submission investigates concept activation vectors (CAVs), which are used for neural network interpretability. Reviewers praised the clear and novel framing and the soundness of the investigations. The work even goes beyond negative results around CAVs to recommend mitigation strategies for undesirable properties of CAVs. One reviewer expressed remaining concerns surrounding the demonstration of the spatial dependence property, but did not elaborate. I assess that the authors have sufficiently responded to concerns raised in the initial review around this property. I recommend acceptance to TMLR.

**Audience:**

Yes.

**Claims And Evidence:**

Yes.